# Dual-frequency spectral radar retrieval of snowfall microphysics: a physics-driven deep learning approach

Anne-Claire Billault-Roux[1], Gionata Ghiggi[1], Louis Jaffeux[2], Audrey Martini[3], Nicolas Viltard[3], and Alexis Berne[1]

[1]Environmental Remote Sensing Laboratory, École Polytechnique Fédérale de Lausanne, Lausanne, Switzerland
[2]Laboratoire de Météorologie Physique, Aubière, France
[3]Laboratoire Atmosphère, Milieux et Observations Spatiales, IPSL, UVSQ Université Paris-Saclay, Sorbonne Université, CNRS, Guyancourt, France

**Correspondence:** alexis.berne@epfl.ch

**Abstract.** The use of meteorological radars to study snowfall microphysical properties and processes is well established, in particular via a few distinct techniques: the use of radar polarimetry, of multi-frequency radar measurements and of the radar Doppler spectra. We propose a novel approach to retrieve snowfall properties by combining the latter two techniques, while relaxing some assumptions on, e.g., beam alignment and non-turbulent atmosphere.

The method relies on a two-step deep-learning framework inspired from data compression techniques: an encoder model maps a high-dimensional signal to a lower-dimensional "latent" space, while the decoder reconstructs the original signal from this latent space. Here, Doppler spectrograms at two frequencies constitute the high-dimensional input, while the latent features are constrained to represent the snowfall properties of interest. The decoder network is first trained to emulate Doppler spectra from a set of microphysical variables, using simulations from the radiative transfer model PAMTRA as training data. In a second step, the encoder network learns the inverse mapping, from real measured dual-frequency spectrograms to the microphysical latent space; doing so, it leverages with a convolutional structure the spatial consistency of the measurements to mitigate the ill-posedness of the problem.

The method was implemented on X- and W-band data from the ICE GENESIS campaign that took place in the Swiss Jura in January 2021. An in-depth assessment of the retrieval accuracy was performed through comparisons with colocated aircraft in-situ measurements collected during 3 precipitation events. The agreement is overall good and opens up possibilities for acute characterization of snowfall microphysics on larger datasets. A discussion of the sensitivity and limitations of the method is also conducted.

The main contribution of this work is on the one hand the theoretical framework itself, which can be applied to other remote sensing retrieval applications and is thus possibly of interest to a broad audience across atmospheric sciences. On the other hand, the retrieved seven microphysical descriptors provide relevant insights into snowfall processes.

# 1 Introduction

Solid precipitation is a phenomenon of extraordinary complexity, whose better understanding and modeling remains a key challenge in atmospheric science. A more accurate representation of snowfall microphysical processes is not only crucial to improve weather forecast models and precipitation quantification (e.g. Khain et al., 2015; Morrison et al., 2020), but also to reduce current uncertainties in cloud radiative properties, with in turn sizeable impacts on climate-oriented research (e.g. Curry et al., 1996; Matus and L'Ecuyer, 2017). From a different perspective, snowfall microphysics is also relevant to a wide range of socio-economical fields, including the aviation industry, for which ensuring flight safety in snowfall conditions is critical (Rasmussen et al., 2000; Cao et al., 2018; Taszarek et al., 2020).

However, the quantification of snowfall properties, such as particle size, mass, bulk density and geometry, is not a straightforward task. In-situ snow particle measurements, whether ground-based or airborne, are highly valuable but are typically sparse and often insufficient to capture the complex spatio-temporal evolution of the particles. Besides, certain quantities like particle mass are particularly difficult to measure and usually available only for small sets of particles, although recent technical and methodological developments open up the possibility for automatized estimations (Leinonen et al., 2021; Rees et al., 2021).

Alternatively, remote sensing instruments, such as meteorological radars, provide measurements related to the scattering of an electromagnetic signal by an ensemble of hydrometeors, over a vertical column of the atmosphere for profiling radars or full 3D regions for scanning ones. Yet, such measurements are indirect and there is no known analytical expression to derive snowfall microphysical descriptors directly from radar measurements. Due to the large variability of snow crystal geometrical and scattering properties, too strong simplifications of the radiative calculations may yield erroneous results (Leinonen et al., 2012). Recent research efforts in this direction have brought about significant improvements in scattering models (e.g. Kuo et al., 2016; Lu et al., 2016; Hogan et al., 2017; Ori et al., 2021). In spite of this progress, the estimation of microphysical properties from radar measurements often remains an ill-posed problem, and is further hindered by measurement uncertainty, for example related to instrument miscalibration or attenuation along the radar path.

Radar retrievals and analyses of snowfall microphysics have been successfully conducted using distinct approaches. On the one hand, the use of multi-frequency measurements has become quite popular: this approach relies on the fact that large snow particles transition to non-Rayleigh scattering regimes at millimeter wavelengths, while they remain Rayleigh scatterers at larger wavelengths. Combining measurements from a shorter and a longer wavelength radar (e.g. W- and X-band), the dual-frequency ratio of radar equivalent reflectivity ($DFR = Ze_X - Ze_W$, in dB) can thus be used to identify populations of snow particles with a larger size, or with a higher degree of riming (e.g., Matrosov et al., 1992; Matrosov, 1998; Szyrmer and Zawadzki, 2014; Liao et al., 2016; Battaglia et al., 2020), thus indicating regions of enhanced snowfall growth. With three well-chosen radar frequencies, studies were able to identify distinct signatures for riming and aggregation mechanisms, and even retrieve estimates of fractal dimension during some parts of snowfall events (e.g. Kneifel et al., 2011; Kulie et al., 2014; Leinonen et al., 2018a), which were later confirmed through comparison with in-situ airborne data (Nguyen et al., 2022). Similar retrievals, focusing on snow density, were achieved using only two frequencies, but leveraging information contained in the mean Doppler velocity in addition to radar reflectivity (Mason et al., 2018). Bringing this a step further, Mróz

et al. (2021b) were recently able to retrieve from triple-frequency radar reflectivity and Doppler velocity accurate estimates of ice water content, snow particle characteristic size as well as an estimate of riming degree. In the case of scanning radars, additional polarimetric information can be included, which opens up possibilities for the geometrical characterization of ice-phase hydrometeors (e.g., Bukovčić et al. (2018); Matrosov et al. (2020) or Tetoni et al. (2021); Oue et al. (2021) where polarimetric and multi-frequency measurements are combined). Most of the cited studies, however, rely on certain hypotheses, for example on the mass-size relation, with assumptions ranging from the use of a strict parameterization to more flexible yet still constraining models like the "filling-in" hypothesis (Mróz et al., 2021b).

On the other hand, more qualitative studies have been conducted relying not solely on radar moments (e.g. reflectivity $Z_e$ or mean Doppler velocity $MDV$) but rather on the full Doppler spectrum, which allows separating the contribution of slow-falling —typically small— vs. fast-falling —typically large or dense— particles to the total reflectivity. Indeed, the full Doppler spectrum encloses more information on microphysical properties and the particle size distribution (PSD) than scalar moments like $Z_e$ or $MDV$. By observing wider, more skewed, or even multi-modal spectra, signatures of specific microphysical processes can be identified such as riming or aggregation (e.g. Shupe et al., 2004; Kalesse et al., 2016).

Combining multi-frequency and Doppler spectral techniques appears like a promising way to go, possibly allowing to reduce the number of required assumptions for a microphysical retrieval, as investigated by Kneifel et al. (2016) and Barrett et al. (2019). The transition of the scattering regime at higher frequencies is visible in dual-frequency Doppler spectra with the following signature: slow-falling particles are typically Rayleigh scatterers and contribute to similar reflectivity at both wavelengths; while larger, fast-falling particles are no longer Rayleigh scatterers for the higher frequency with thus smaller spectral reflectivity than for the lower frequency. This means that the Doppler spectra at both frequencies should "match" on the low-velocity side, and diverge for large velocities. However, using this principle to perform a direct inversion like Barrett et al. (2019) is only rarely possible. Difficulties related to imperfect measurements are substantial: not only should the different radars be well cross-calibrated in reflectivity, they should also be well aligned vertically to avoid contamination by horizontal wind. The additional issue of non-uniform beam filling is all the more problematic when the radars have different beam widths or range resolution: this would hinder the retrieval, especially when turbulent broadening is observed, or when the particle populations in the sampled volumes are too heterogeneous. Differential attenuation of the two frequencies is yet another significant challenge, for which some workarounds were proposed (e.g. Li and Moisseev, 2019), but are not always possible to implement. In such cases, a direct computation of the dual-frequency spectral ratio is difficult to interpret and may be dominated by these artifacts.

In this work, we propose an approach to retrieve snowfall microphysics from dual-frequency Doppler spectra, while partly relaxing these constraints on turbulence or beam alignment, as well as reducing the number of prior assumptions on snowfall microphysical properties. Whereas many retrievals in atmospheric sciences rely on classical Bayesian frameworks (e.g. Rodgers, 2000), we opt here for an alternative machine-learning based method: some cutting-edge developments achieved in the past decade have outlined the strong potential of such statistical methods in atmospheric science (Bauer et al., 2021; Chantry et al., 2021) and weather radar applications (Geng et al., 2021), especially to tackle retrieval problems (Vogl et al., 2022; Chase et al., 2021).

Exploiting recent advances in deep learning research, we introduce a physics-driven inversion framework, which is partly inspired from auto-encoder models. The auto-encoder is a neural network architecture originally designed for dimension reduction purposes, and sometimes referred to as a non-linear principal component analysis variant (Kramer, 1991; Hinton and Salakhutdinov, 2006): an *encoder* neural network maps a high-dimensional signal to a low-dimensional *latent* or *feature* space, while the *decoder* neural network learns to recover the original signal from this latent space. In our case, dual-frequency Doppler spectrograms constitute the high-dimensional signal, while the dimensions of the latent space are implicitly constrained to represent the snowfall properties which we seek to retrieve. In a first step, the decoder is trained to emulate a radiative transfer model, i.e. to reconstruct dual-frequency Doppler spectrograms given (latent) snowfall descriptors. The encoder is trained in a second step: it consists of an advanced deep learning architecture, that ingests the radar data (dual-frequency Doppler spectrograms), and is optimized to retrieve the latent snowfall properties which, when passed through the decoder, minimize the reconstruction error with respect to the input data. An important peculiarity of the encoder's architecture is its ability to leverage the spatial consistency of the radar variables, which reduces the ill-posedness of the inversion problem. By training not only one but several deep learning models with different random initializations, we gain additional insight into the uncertainty of the retrieval.

The proposed framework is implemented on data from the ICE GENESIS campaign that took place in the Swiss Jura in January 2021. The set-up included in particular X- and W-band Doppler spectral profilers at the ground, complemented with overpasses of a scientific aircraft equipped with microphysical probes. This offers the possibility to validate the retrieval against in-situ measurements.

A general overview of the retrieval framework and its theoretical foundation is presented in Sect. 2. Section 3 is dedicated to the presentation of the synthetic and real data sets used to train and evaluate the inversion model. In Sect. 4, we detail the technical implementation of the framework. Results are then presented in Sect. 5, with a particular focus on the comparison of the retrieval outputs to in-situ aircraft measurements. The discussion of the results is taken a step further in Sect. 6, with a focus on the current limitations of the method and its sensitivity to certain key assumptions.

## 2 Theoretical framework

This section introduces the theoretical components required to understand the proposed retrieval framework, and provides an overview of its general structure.

### 2.1 Doppler spectra: forward model

Radar Doppler spectra, computed through the Fourier transform of the radar return signal (Doviak and Zrnic, 1993), feature the reflectivity-weighted distribution of the targets' Doppler velocity in a given radar volume. From here on, *"Doppler spectrum (pl: spectra)"* refers to the measurement at a given time and range gate, and the vertical stack of spectra at a certain time is denoted as the *"Doppler spectrogram"*. Note that this name convention is used here for clarity, but it may not be universal.

In the case of a vertically-pointing profiler, the shape of the Doppler spectrum in snowfall results from a combination of several

factors (e.g. Doviak and Zrnic, 1993; Kollias et al., 2002; Luke and Kollias, 2013; Kneifel et al., 2016). It is primarily defined by the snowfall PSD and the microphysical properties of the snow particles (e.g. bulk density, geometry, etc) which determine their backscattering cross-section and terminal velocity. In reality, this purely microphysical spectrum is affected by atmospheric dynamic conditions —turbulence, horizontal and vertical wind— in a way that depends on the settings and parameters of the radar itself —sensitivity, beam width. The actual measured spectrum is additionally perturbed by instrument noise, the effect of which is mitigated through temporal averaging of the spectra, at the risk of smearing out underlying microphysical signatures (e.g Acquistapace et al., 2017). Understanding how those parameters (microphysical, environmental, instrumental) translate into a measured Doppler spectrum is delicate: it involves complex radiative transfer models to compute the radar backscatter of snow particles, and it also requires an understanding of snowfall aerodynamic properties e.g. for the parameterization of the particle velocity-size relations.

Efforts have been devoted to the construction of increasingly accurate forward models, for instance through computationally costly discrete dipole approximation (DDA) calculations (e.g. Draine and Flatau, 1994; Liu, 2004; Kuo et al., 2016; Lu et al., 2016), and through simulations based on the self-similar Rayleigh-Gans approximation (SSRGA), which are tuned to represent accurately the scattering of various particle types (Hogan and Westbrook, 2014; Hogan et al., 2017; Ori et al., 2021). In this work, we use the radiative transfer code PAMTRA (Mech et al., 2020) as a forward model, as it is particularly suited to simulate full Doppler spectra and provides an implementation of several scattering models. Details on how PAMTRA is used and parameterized in this study are presented in Section 3.1.1.

## 2.2 Approach to the inverse problem

Assuming a forward model, noted $\mathbf{f}$, is known —which, given a set of properties $\boldsymbol{x}$, outputs realistic Doppler spectra $\boldsymbol{y}$— the aim of the retrieval is to solve the following *inverse problem*: from real observed spectra $\boldsymbol{y_r}$, estimate the underlying microphysical properties $\boldsymbol{x_r}$ (see e.g. Maahn et al., 2020, for a discussion on inverse problems). Here the subscript $r$ denotes real values as opposed to synthetic or modeled quantities.

In a mathematical language, this means estimating $\mathbf{g} = \mathbf{f}^{-1}$. This is in general not possible, as $\mathbf{f}$ is usually not an invertible mapping. Workarounds can be developed in certain cases, for example through look-up tables (e.g Leinonen et al., 2018b). Alternatively, one can seek $\boldsymbol{x_r}$ as the minimizing argument of a cost function (e.g. $||\boldsymbol{y_r} - \mathbf{f}(\boldsymbol{x})||^2$), which can also include a regularization term (e.g. Mason et al., 2018); this minimization problem can then be solved iteratively with for instance a gradient descent algorithm. From a Bayesian perspective and under additional assumptions (Gaussian probability distributions), this corresponds to the popular *Optimal Estimation* (OE) framework (Rodgers, 2000; Maahn et al., 2020), which is widely used across atmospheric science to solve moderately linear inverse problems. Although this alleviates some requirements on $\mathbf{f}$, it can only be implemented if $\mathbf{f}$ is differentiable, and if the computation of its gradient is tractable, either analytically or numerically. This "classical Bayesian" approach faces some limitations, which include but are not limited to, the need to linearize the forward operator in order to compute its Jacobian, or to explicitly assume prior values for $\boldsymbol{x}$.

Machine learning techniques offer the possibility to tackle inverse problems in a different way, with a statistical rather than an analytical approach. Note that, as pointed out by Geer (2021), the overarching framework in both cases ultimately remains that of Bayesian probabilities, viewed through different prisms. The typical machine learning route to solve an inverse problem (e.g. Chase et al., 2021) has the following structure: the available forward model is first used to create a large synthetic dataset $\{(\boldsymbol{x_s^k}, \boldsymbol{y_s^k} = \mathbf{f}(\boldsymbol{x_s^k})), k = 1..N\}$; the $s$ subscript denoting synthetic values and $N$ the size of the dataset. Then, a machine learning model is trained on this dataset to learn a statistical relation between $\boldsymbol{y}$ and $\boldsymbol{x}$, i.e. an approximation of the inverse mapping $\tilde{\mathbf{g}}$. Ultimately, this produces a gate-to-gate inversion of the problem which can be implemented on real data. This approach has been successfully used for atmospheric retrievals, for example by Piontek et al. (2021) to detect volcanic ash clouds, Vogl et al. (2022) to estimate riming occurrences from radar measurements, or Chase et al. (2021) to retrieve snowfall properties from airborne or satellite radars.

One major limitation of this "direct" gate-to-gate method is when the problem itself is ill-posed, e.g., when several values of $\boldsymbol{x}$ may yield similar outputs $\boldsymbol{y}$ ($\mathbf{f}$ is not injective): in such cases, the retrieval may yield arbitrary outputs.

The proposed approach, illustrated in Fig. 1, can mitigate this issue. It is inspired from auto-encoder architectures (Kramer, 1991; Hinton and Salakhutdinov, 2006), which use neural networks to perform powerful non-linear dimension reduction: an *encoder* network maps a high-dimension signal to a low-dimension latent space, while the *decoder* network learns to reconstruct an approximation of the original signal from the latent space. Such tools are relevant for atmospheric sciences and in particular in the context of climate studies, which handle complex, high-dimensional signals (Behrens et al., 2022). In our case, the aim is to constrain the dimensions of the latent space to contain microphysical descriptors of snowfall: the originality of the approach presented here is thus that it incorporates physical knowledge by using a physics-informed decoder.

  – In a first step, a neural network is trained on a synthetic dataset of $(\boldsymbol{x_s}, \boldsymbol{y_s} = \mathbf{f}(\boldsymbol{x_s}))$. Instead of learning an inverse mapping, it simply learns to emulate the forward model: taking microphysical and atmospheric (i.e. *latent*) variables ($\boldsymbol{x_s}$ as input, it outputs Doppler spectra ($\boldsymbol{y_s}$). This model, which we hereafter refer to as the *"decoder"* and denote $\tilde{\mathbf{f}}$, is thus a differentiable emulator of PAMTRA. When applied not to a single set of microphysical descriptors, but to a stack (of multiple range gates) at once, it is denoted with upper-case $\tilde{\mathbf{F}}$. The synthetic dataset should include a wide range of realistic parameters, to not induce bias in the further steps.

  – In a second step, we shift our attention to a real (i.e. not synthetic) dataset of full dual-frequency Doppler spectrograms $\boldsymbol{Y_r}$; the aim is to retrieve the underlying profiles of latent variables $\boldsymbol{X_r}$. The capital letter denotes that e.g., $\boldsymbol{Y_r}$ is a vertical stack of $\boldsymbol{y_r}$. A second neural network, the encoder $\tilde{\mathbf{G}}$, is trained on this real data set: it takes as input the spectrograms $\boldsymbol{Y_r}$, and its output $\boldsymbol{X} = \tilde{\mathbf{G}}(\boldsymbol{Y_r})$ has the same dimension as the number of latent features, times the number of range gates. $\boldsymbol{X}$ is passed on to the decoder $\tilde{\mathbf{F}}$, which outputs a reconstructed spectrogram $\boldsymbol{Y}$. Training is performed by optimizing $\tilde{\mathbf{G}}$ in order to minimize the reconstruction error $||\boldsymbol{Y} - \boldsymbol{Y_r}||^2$.

At the end of the training, i.e., when the pipeline has converged, $\tilde{\mathbf{F}} \circ \tilde{\mathbf{G}}(\boldsymbol{Y_r}) \approx \boldsymbol{Y_r}$ and the output of the encoder $\hat{\mathbf{X}}_{\mathbf{r}} = \tilde{\mathbf{G}}(\boldsymbol{Y_r})$ should be close to the true profile of microphysical descriptors $\boldsymbol{X_r}$.

The architecture of the decoder and encoder will be detailed further on (Sect. 4), but one key property should already be underlined. The retrieval operates on the full dual-frequency Doppler spectrograms at once, rather than on each gate independently: the idea is to synergistically make use of the spatial structure of the measurements to reduce the ill-posedness of the inverse problem. By "spatial structure", or "spatial consistency", we refer to the fact that the spectrogram might be continuous, smooth (i.e. spectra at nearby ranges are similar), or on the contrary have some abrupt changes (e.g. in the case of high shear, where neighboring spectra might be very different). By constraining the retrieval to output a profile of microphysical variables with a similar spatial structure, we restrain the number of degrees of freedom.

In practice, this is handled by the architecture of the encoder network, which contains convolution kernels: thanks to this feature, the model can capture the vertical structure of the Doppler spectrograms, and propagate this information in a way that the output profiles are themselves spatially consistent. Note that while the issue of ill-posedness is mitigated, it is not entirely resolved, as there may remain an intrinsic under-determination. Nevertheless, we believe that this implicit use of the measurements' spatial features in the retrieval is a key contribution of this work. To support this, a brief discussion of alternative methods is proposed further on (Sect. 6.5).

To conclude this overview of the framework, we highlight that while it was presented for the specific case of Doppler spectrograms and snowfall microphysics, its structure is generic and could potentially be applied to other retrieval problems with similar properties: a complex forward model that is not directly invertible, with slightly ill-posed features that hamper pointwise retrievals. One intrinsic limitation which should be highlighted is that the method is trained directly on the data of interest and cannot be directly used on any given measurements.

## 3   Data

### 3.1   Synthetic data set

As mentioned above (Sect. 2 2), the first step of the framework consists in training a neural network on a synthetic dataset containing sets of microphysical and atmospheric variables and the corresponding spectra. The focus of this section is the generation of this dataset.

### 3.1.1   Forward model assumptions

To generate this synthetic dataset, PAMTRA is run by prescribing snowfall microphysics through several parameters. These parameters are the snowfall properties that the algorithm will then learn to retrieve.

The definitions of the microphysical parameters are summarized in Table 1. The PSD is assumed to be a negative exponential ($N(D) = N_0 \exp(-D/D_0)$, e.g. Straka, 2009), whose size parameter $D_0$ is prescribed; here and further, the size or diameter of a particle is defined as its maximum dimension: $D = D_{max}$. For an exponential PSD, $D_0$ is equal to the number-concentration-weighted mean diameter (shortened as "mean diameter"); the effective diameter (ratio of the third to second moment of the

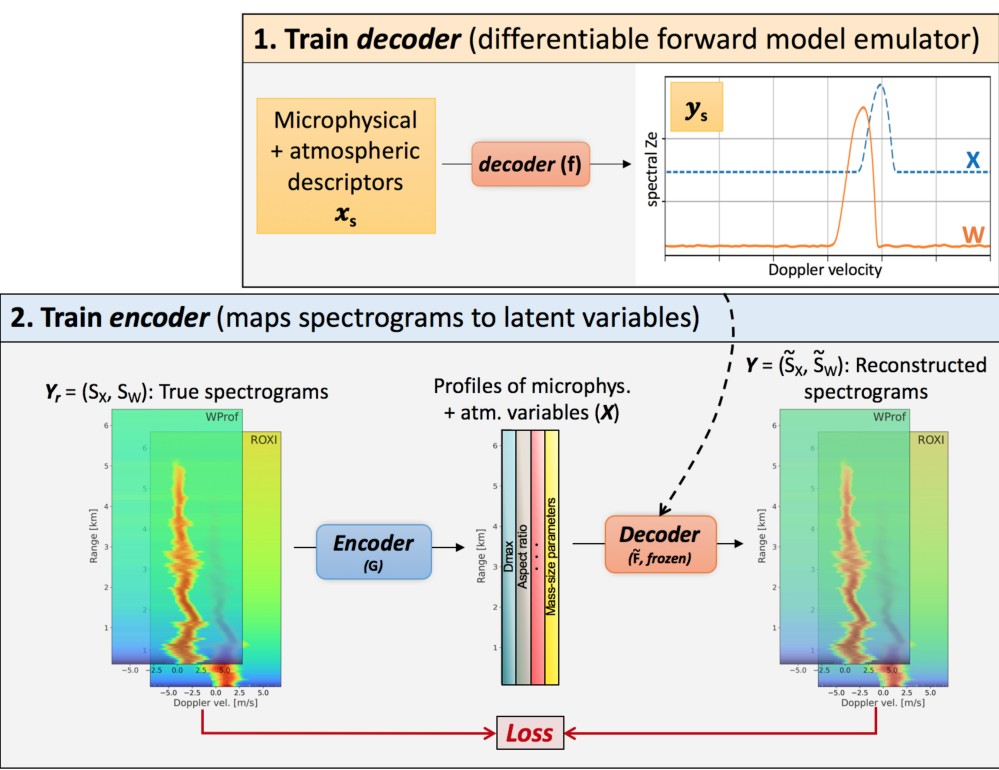

**Figure 1.** Schematic illustration of the method. The notations are those of Sect. 2.2. The upper right box illustrates that the decoder NN is trained to emulate a forward radiative transfer model. The lower box shows the full pipeline where the pretrained decoder is used to reconstruct full spectrograms based on the microphysical properties output by the encoder NN.

PSD), often relevant for radiative transfer models, is $D_{eff} = 3D_0$. The choice of an exponential shape for the PSD was made to limit the degrees of freedom of the retrieval and keep the computational expense tractable; it is nonetheless a strong underlying hypothesis of the framework in its current version (discussed in Sect. 6.4). Mass-size and area-size relations are considered to be power laws, whose prefactors and exponents are prescribed ($m = a_m D^{b_m}$, $A = \alpha_a D^{\beta_a}$). The aspect ratio $A_r$ is then specified, defined here as equal to the particle's dimension along the direction of radar beam (here, vertical) divided by maximum dimension (Ori et al., 2021), which implies $A_r \leq 1$. The particles are considered to be oriented with their maximum dimension in the horizontal plane. The ice water content (IWC) is finally assigned. Note that the particle number concentration is implicitly prescribed through the definition of IWC, $D_0$ and $a_m$, $b_m$. In addition, the noise level is specified, since it is required to simulate realistic Doppler spectra; in practice, it only depends on the range and on the radar properties, and is not related to other microphysical or atmospheric quantities. The velocity-size relation is the one proposed by Heymsfield and Westbrook (2010), and relies on the aforementioned mass-size and area-size relations.

Individual spectra are simulated through PAMTRA for an altitude of 1000 masl, using a standard (PAMTRA default) atmospheric profile with a temperature randomly chosen in [-20 °C, 1°C]. Spectra are then simulated at X- and W-band indepen-

dently. The radar settings for these simulations (frequency, beamwidth, velocity resolution, velocity range, sensitivity) should have the same values as those of the radars on which the retrieval is implemented (cf Sect. 3.2 and Table 2). In the current version of the algorithm, attenuation is not taken into account in the PAMTRA simulations.

Scattering calculations are performed using the SSRGA, with coefficients from Ori et al. (2021); more detail on this is provided in Appendix B2. These assumptions on scattering properties are not flawless, and constitute a bottleneck in our method, as in virtually any attempt at radar-based retrievals. In particular, the current implementation of PAMTRA (28/03/2022) allows for the parameterization of only two coefficients of the SSRGA ($\kappa_{SSRG}$, $\beta_{SSRG}$), while current literature suggests that more coefficients should be used ($\gamma_{SSRG}$ and $\zeta_{SSRG}$, see Hogan et al., 2017 for detail on the coefficients); furthermore, the

variability of the scattering parameters shown for instance in Ori et al. (2021) or Leinonen et al. (2018a), Fig. 5, was neglected when the parameters were sampled (cf. B2). It should also be kept in mind that the SSRGA would fail to represent large graupel particles, although Ori et al. (2021) suggests that its validity extends to particles with a relatively high riming degree. These assumptions can thus naturally be questioned. We, however, believe that it was reasonable to use the simplest possible parameterization for the initial development of the method —and in particular, use a common SSRGA framework for all

scattering calculations—, leaving possible improvements of the forward model to future work.

### 3.1.2   Forward model inputs

When generating this training set, a trade-off has to be defined: if the dataset is too narrow, that is, if it doesn't cover a large enough range of values and combinations for the microphysical descriptors, this will cause a bias in the retrieval; conversely, if the range of values is much too large, this will hinder the training process, for it will include non-realistic values. It was therefore

chosen to parameterize PAMTRA by sampling the microphysical properties using a large observational dataset collected using the Multi-Angle Snowflake Camera during 10 field deployments. This data was organized into a database in Grazioli et al. (2022). We follow the method presented in this study (Grazioli et al., 2022, Sect. "Technical Validation") to derive from the database the microphysical parameters required in the forward model (Sect. 3.1, Table 1). Four categories of particles are used: aggregates, planar crystals, columnar crystals and graupel. For each type of particle and for each parameter, a distribution is

fitted to the empirical histogram calculated from the database. We refer to Appendix B1 for more detail. When generating the training set, parameters are then randomly sampled from those distributions. It is worth highlighting that all parameters are sampled independently, with the exception of $a_m$ and $\alpha_a$. Indeed, as pointed out in Grazioli et al. (2022), a strong correlation exists between $a$ and $b$, and between $\alpha_a$ and $\beta_a$; thus, empirical fits are used from which $a_m$ (resp. $\alpha_a$) is sampled for a given $b_m$ (resp. $\beta_a$), with the addition of randomness using the mean squared error of the fit. The definition of aspect ratio

$A_r$ is slightly different between the MASC dataset and the SSRGA parameterization: in the former, it is equal to the ratio of minor axis length to major axis length (Garrett et al., 2015), while in the latter —as mentioned above— it is equal to the particle's dimension along the direction of the radar beam, divided by maximum dimension. After some empirical exploration, it was decided to use nonetheless the histograms from MASCDB, given that the distributions were quite broad, indicating that this difference in definition should not bias the retrieval. Ice water content is the only parameter for which MASCDB does

not provide estimates; therefore, it was empirically decided based on literature (Noh et al., 2013) and preliminary analyses

of aircraft in-situ measurements during the ICE GENESIS campaign (cf Sect. 3.2) to sample it from a negative exponential distribution with a mean of $0.5 \, \mathrm{g \, m^{-3}}$.

**Table 1.** Microphysical, atmospheric and radar parameters

| Name | Description |
|---|---|
| IWC | Ice water content |
| $D_0$ | Mean diameter (assuming exponential PSD) |
| $b_m$ | Exponent of the mass-size power law |
| $a_m$ | Pre-factor of the mass-size power law |
| $\beta_a$ | Exponent of the area-size power law |
| $\alpha_a$ | Pre-factor of the area-size power law |
| $A_r$ | Aspect ratio (cf. Sect. 3.1.3) |
| $TurbX$ | Broadening X-band |
| $TurbW$ | Broadening W-band |
| $WindX$ | Radial wind X-band |
| $TurbW$ | Radial wind W-band |
| $LnoiseX$ | Noise level at X-band |
| $LnoiseW$ | Noise level at W-band |

### 3.1.3 Generation of the training set

Each item of the dataset is generated through the following procedure.

1. A particle type is randomly sampled among the four aforementioned types. Given the large variety that exists within the aggregate category, as observed in MASCDB, it is given more weight in the sampling procedure (aggregates: 40% - planar crystals: 20% - graupel: 20% - columnar crystals: 20 %).

2. Microphysical descriptors are randomly sampled using the MASC-based distributions.

3. PAMTRA is run on these descriptors, under the previously stated assumptions. The corresponding Doppler spectra are computed for 9.48 and 94 GHz (frequency of the radars used in this study, cf. Sect. 3.2), with 512 bins and a Nyquist velocity of $6.92 \, \mathrm{m \, s^{-1}}$.

4. Then, turbulent broadening and spectrum shift due to radial wind are added, with randomly sampled values, different for X- and for W-band. Including these variables will allow the retrieval to handle possible velocity offsets in the X- and W-band spectra caused by beam misalignment, or differences in spectral broadening due to the different beam widths of the radars. While these could be computed directly in PAMTRA, it was more computationally efficient to implement them in post-processing in a vectorized way.

– The radial wind parameter includes the velocity shift of the spectrum that could be caused by vertical wind, beam misalignment, etc. It is randomly sampled within [-2, +2] $\mathrm{ms}^{-1}$ i.e. in the typical range of vertical wind in non-convective precipitation.

– The broadening parameter is the size of the Gaussian broadening kernel and includes the effect of turbulent eddies but also accounts for all other possible broadening causes (e.g. horizontal wind, beam width). It is computed by randomly sampling a value of atmospheric turbulence, represented by the eddy dissipation rate (sampled from a negative exponential with $10^{-3}$ $\mathrm{m}^2\mathrm{s}^{-3}$ mean, consistent with some literature standards, e.g. Sharman et al., 2014). The resulting broadening is derived following Shupe et al. (2008, Eq. 4); the radar settings (e.g. beam width) used in these equations are those of the W- and X-band radars used in this study, described in Sect. 3.2.

5. Finally, for computational reasons, X- and W-band spectra are both reduced to 256 points through bin averaging.

Ultimately, each item of the synthetic dataset contains an input vector with 13 dimensions (see Table 1) and the corresponding simulated Doppler spectra (X- and W-band) with each 256 bins. We underline that the synthetic data set contains information only at the scale of the radar sampling volume at a given range gate, i.e., not a full spectrogram.

## 3.2  X- and W-band Doppler spectrograms

In this section, we present the experimental dataset used for the implementation of the second part of the pipeline. Measurements were collected during the 2021 ICE GENESIS campaign, a joint ground-based and airborne field experiment that was conducted in the Swiss Jura in January 2021, and is fully described in Billault-Roux et al. 2022 (in press). Data from X- and W-band vertically-pointing Doppler radars are used, which were located at les Éplatures airport. The X-band radar, further on referred to as ROXI (Viltard et al., 2019), is a high-sensitivity 9.48 GHz Doppler spectral profiler with 1.8° 3-dB beam width. It was deployed next to a dual-polarization W-band 94 GHz Doppler spectral profiler (WProf, Küchler et al., 2017) with 0.53° beam width. The properties and settings of both radars are summarized in Table 2. As pre-processing steps, the radars are cross-calibrated and an attenuation correction is implemented at W-band (similarly to Kneifel et al., 2015), as detailed in Appendix A.

Then, the spectrograms of both radars are remapped to a common grid, by averaging in time (with a resolution of $20\,\mathrm{s}$), interpolating in range (resolution of $50\,\mathrm{m}$), and average-binning the velocity to the same bins as the synthetic dataset, i.e. with 256 bins and a velocity cutoff $v_{Nyq} = 6.92\,\mathrm{ms}^{-1}$. Only time frames with detectable signal in both frequencies are used, leading to a total of $\sim 9000$ profiles corresponding to around 50 hours of measurements, collected between January 16th and January 28th.

**Table 2.** Parameters of ROXI and WProf radars during the ICE GENESIS deployment. WProf range and time resolution (as well as Nyquist velocity) are defined in three chirps. The lower chirp ranges from 100 m to 900 m, the second one from 900 m to 3900 m and the higher one from 3900 m to 9000 m

| Radar properties | ROXI | WProf | | |
| --- | --- | --- | --- | --- |
| | | *chirp 0* | *chirp 1* | *chirp 2* |
| Frequency (GHz) | 9.48 | — | 94 | — |
| Beamwidth (°) | 1.8 | — | 0.53 | — |
| Time resolution (s) | 3.5 | — | 5 | — |
| Range resolution (m) | 50 | 7.5 | 16 | 32 |
| Velocity resolution (m/s) | 0.1 | 0.02 | 0.014 | 0.013 |
| Nyquist velocity (m/s) | 11 | 10.8 | 6.92 | 3.3 |
| Sensitivity (dBZ) [at range (km)] | -19 [2] | -45 [0.5] | -41 [2] | -39 [5] |

### 3.3 Data for model evaluation

#### 3.3.1 Polarimetric radar

MXPol (Schneebeli et al., 2013) is a polarimetric X-band scanning radar that was deployed 4.8 km away from the main site and performed routine range-height indicator (RHI) scans in direction of the X- and W-band profilers during precipitation. Hydrometeor classification with demixing (Besic et al., 2016, 2018) was performed on this data to estimate from the polarimetric variables the proportions of hydrometeor types in the sampled volume. From the RHIs, remapped to a Cartesian grid, profiles are extracted over the main site with a horizontal $\delta x = \pm 500$ m, using only elevation angles below 45 degrees. The time series of hydrometeor classification extracted in this manner will be used qualitatively as an independent verification tool to assess the performance of our microphysical retrieval.

#### 3.3.2 Aircraft in-situ measurements

In addition to the ground-based measurements, the ICE GENESIS campaign included scientific aircraft overpasses with remote-sensing and in-situ instruments. The airborne in-situ data is particularly valuable for the quantitative evaluation of the microphysical retrieval, presented in Sect. 5. Airborne measurements used in this work were collected during three flights of the Safire ATR-42 (Jan. 22, Jan. 23, Jan. 27) as the aircraft was performing overpasses over the ground site, with data from several probes.

First, the Counterflow Virtual Impactor (CVI, Anderson et al., 1994; Schwarzenboeck et al., 2000) provides a measurement of total water content (TWC), and the Cloud Droplet Probe (CDP) of liquid water content (LWC); from those measurements, an estimate of IWC can be obtained as $IWC = TWC - LWC$. Two imaging probes are also used, the 2D-Stero probe (2D-S) and the Precipitation Imaging Probe (PIP) sampling respectively 10 $\mu$m to 1.28 mm and 100 $\mu$m to 6.4 mm (Baumgardner

et al., 2017; McFarquhar et al., 2017, for a complete reference). From the images of these probes, the method of Leroy et al.
(2016) is used to compute the PSD and derive the following microphysical descriptors: mean aspect ratio, mass-size power law coefficients, area-size power law coefficients. In order to estimate the $D_0$ parameter, an exponential distribution is fitted to the PSD (leaving out small particles with $D_{max}$< 800μm, since it was empirically noted that these did not follow this exponential behavior); instances, when the assumption of an exponential tail is invalid, can be identified by filtering on the correlation coefficient of this fit. This approach was chosen rather than computing moments from the in-situ PSDs, as those could potentially be affected by the size cut-off at 6.4 mm, while the slope of the distribution is expected to be a more robust indicator. For the mass-size parameters, in addition to the CVI closure method proposed by Leroy et al. (2016), another method is used for particle-by-particle mass reconstruction based on Lawson and Baker (2006); the methods are respectively denoted with the $CVI$ and $BL$ subscripts. All aircraft-based microphysical descriptors are computed using 5-second running averages of the measurements.

### 3.3.3 Airborne radar retrieval

The aircraft was also equipped with an upward-looking W-band radar (RASTA, Plana-Fattori et al., 2010) from which values of IWC were derived (Delanoë et al., 2007), noted with the $RASTA$ subscript. In order to compare $IWC_{RASTA}$ to the in-situ measurements, the closest valid radar gates are used, which correspond to a vertical distance of 150 to 250 meters above the aircraft. For a fair comparison between airborne RASTA retrievals and our inversion model, only time steps when the aircraft overpasses the ground site are used, when the aircraft is within a 1 km horizontal distance to the ground site (distance chosen to allow a sufficient number of points for the comparison).

## 4 Deep learning inversion framework

This section addresses the detail of the implementation of the two-step framework outlined in Sect. 2. For designing and training both the decoder and the encoder part of the model, the pytorch library is used (Paszke et al., 2019).

### 4.1 The "decoder": a differentiable emulator of PAMTRA

The first part of the framework consists in developing a differentiable emulator of PAMTRA, by designing a deep learning model and training it on the synthetic dataset (Sect. 3.1). If viewed in perspective with the technique of auto-encoders, this consists in learning the "decoder", which maps the latent space —containing the physical variables— to the high-dimensional measurement space —the spectra. It was chosen to train separately the X-band and W-band decoders rather than use a single algorithm emulating both simultaneously; indeed, the two frequencies may have slightly different smoothness or amplitude features, which justifies the use of distinct architectures. Each decoder takes as input a vector of dimension 10 containing $IWC$, $D_0$, $b_m$, $a_m$, $\beta_a$, $\alpha_a$, $A_r$, $Turb_F$, $Wind_F$ and $Lnoise_F$, where $F$ is either X or W. They output Doppler spectra with 256 points.

### 4.1.1 Decoder architecture

The model, whose architecture is illustrated in Fig. 2 a), is designed as a neural network (NN) with a first part composed of fully-connected layers, and a second part composed of convolutional layers. More precisely, since the aim of this decoder is to increase dimensionality (from 10 to 256), we use one-dimensional "transposed convolutions" which are well suited for this purpose (Zeiler et al., 2010). Since using only transposed convolutions can create artifacts, they are combined with another type of layer that ensures a smooth output, a linear upsampling layer followed by a standard convolution. In order to improve the training of the model, *residual blocks* (He et al., 2015) are used: these blocks contain skip connections and batch normalization steps (Ioffe and Szegedy, 2015), and are quite popular in deep learning applications. In a nutshell, these techniques help mitigate issues caused by the depth of the model: they do not per se improve the expressiveness of the neural network, but they strongly facilitate the training process. For instance, the skip connections (illustrated in Fig. 2) allow to propagate information from earlier layers to further stages of the neural network, and this reduces the risk of gradients vanishing to zero during training (Balduzzi et al., 2017).

### 4.1.2 Decoder training

The synthetic dataset described in Sect. 3.1 is split into training, validation and testing sets (80% - 10% - 10%). The NN is trained on the training set, while the validation set is used to tune the architecture of the NN, and the testing set for a final assessment of its performance (Sect. 5.1.1). The input is normalized using the statistics of the training dataset, with the mean and standard deviation of each variable. For certain variables, the natural logarithm is used instead of the original value, in order to have more homogeneously spread distributions: this is the case for $IWC$, $a_m$ and $\alpha_a$. The network is trained using the Adam optimizer (Kingma and Ba, 2015), with mean square error (MSE) as a loss metric, and with Xavier normal initialization of weights and biases (Glorot and Bengio, 2010); in addition, the learning rate is periodically decreased with a scheduler. The network and training hyperparameters are summarized in Table 3. It was observed that the spectra output by the NN could have a tendency to slightly underestimate the peak values, because of a "flattening" effect common in such methods that use MSE as a loss metric. Hence, we add to the main loss a secondary loss calculated as the MSE computed only on the part of the spectrum close to its peak (above 50 % of its amplitude).

## 4.2 The "encoder": retrieving a profile of latent variables

In the second part of the framework, a second deep neural network, the *encoder*, is used to learn the inverse mapping. Taking as input dual-frequency spectrograms (i.e., an array of shape $N_{rg} \times 256 \times 2$, with $N_{rg}$ the number of range gates), it outputs vectors in the latent space, of shape $N_{rg} \times 13$, 13 being the total number of latent dimensions retrieved, as detailed in Table 1. In this section, we refer to the first dimension as the "range" dimension, to the second one as the "feature" dimension and the last one as the "channel" dimension.

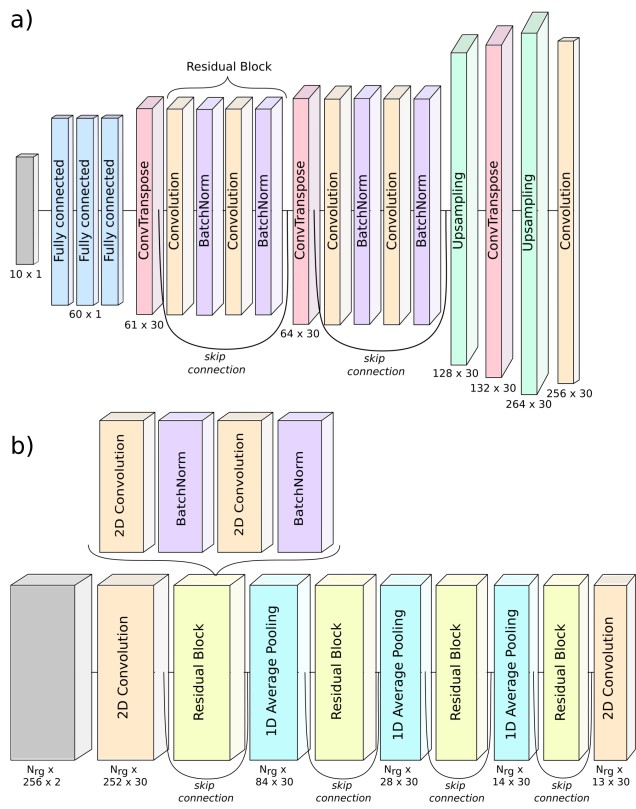

**Figure 2.** Architecture of the decoder (a) and encoder (b) neural networks. For the decoder, the architecture for W-band is shown; that of X-band is extremely similar with only slightly different kernel widths. Skip connections indicate that the output of a given layer is kept and added further on to the output of a residual block. Color coding indicates the type of each layer. The size of each layer is indicated; when not stated, the layer is the same size as the previous one. Note that for display reasons the velocity dimension is along the vertical in a) and along the horizontal direction in b).

### 4.2.1 Encoder architecture

The neural network designed for this part uses 2-dimensional convolutions, which allow to reduce the feature dimension from 256 to 13. In order to keep the range dimension constant (equal to $N_{rg}$), padding is used, which means artificially increasing the size of the array —by replicating on each side the items resp. in first and last position— before performing the convolution. Similar to the decoder architecture, residual blocks with skip connections and batch normalization steps are used. Additionally, average pooling along the feature dimension is performed after each residual block. The size of the convolution kernels along

the range dimension gives a sense of the scale at which we can expect meaningful spatial correlation —both in the latent space and in the measured spectrograms. It is, however, not directly interpretable: as the NN contains stacks of convolution layers, the field of view progressively increases with the model's depth; the output of the NN at a given range gate is influenced not

only by its closest neighbors but also by range gates which are further away. The full architecture of the encoder is displayed on Fig. 2 b).

 ### 4.2.2 Encoder training

The encoder is trained using the radar data presented in Sect. 3.2. The spectrograms are normalized using the same statistics as in the decoder part (means and standard deviations of the synthetic spectra). Rather than using the entire Doppler spectrograms ($N_{rg} = 100$) as one training item, chunks of the spectrograms are used with $N_{rg} = 25$, and are sampled in the following way: the first chunk corresponds to $i_{rg} = 0..24$, the second chunk to $i_{rg} = 5..29$, the third to $i_{rg} = 10..34$, etc.; $i_{rg}$ being the range gate index. The dataset is then randomly shuffled at each epoch during training. Rearranging the dataset in this way makes training both more tractable —thanks to the smaller size of each item— and more robust —due to the data augmentation, which helps avoid local minima during the training process. During training, the encoder output is passed as input to the X- and W-band decoders, which then output reconstructed spectrograms. The reconstruction loss is the MSE between the reconstructed ($\tilde{S}_W, \tilde{S}_X$) and the original spectrograms ($S_W, S_X$): $Loss = (\tilde{S}_X - S_X)^2 + (\tilde{S}_W - S_W)^2$. The encoder parameters are then updated at each step to minimize this loss, here with the Adam optimizer. Training parameters are reviewed in Table 3. It is important to note that the decoders' parameters are frozen at this step: only the encoder is being learned; this differs from classical autoencoder models, for which the decoder and encoder are trained simultaneously.

Estimates of the latent features are then obtained from the encoder's output, after inverse normalization and exponential transform for those variables whose logarithm was used as input to the decoder (see Section 4.1.2). To prevent occasional convergence toward unrealistic values in the latent space (e.g. $D_0 < 0$), an additional constraint was incorporated to the loss term to penalize latent values outside of a manually defined range: for a given feature $x$ with realistic bounds $x_{min}$ and $x_{max}$, this secondary loss reads $L_{sec}(x) = \mathbb{1}_{(x<x_{min})} \times (x-x_{min})^2 + \mathbb{1}_{(x>x_{max})} \times (x-x_{max})^2$. Here $\mathbb{1}_{(x<x_{min})}$ denotes the function which is equal to 1 when $x < x_{min}$ and to 0 otherwise.

### 4.3 Ensemble approach for uncertainty quantification

In order to estimate the uncertainty of the retrieval, an ensemble approach is used: several runs are performed for both the decoders and the encoder, each trained independently with random weight and bias initialization. In the end, a total number of 50 runs is used to compute mean values and standard deviations for each retrieved variable. This both ensures a greater robustness of the retrieved values, which are less likely to reflect local minima, and provides an uncertainty estimate for the retrieval. This is especially relevant given the under-determination of the problem: with this ensemble approach, we can illustrate the uncertainty related to the remaining intrinsic ill-posedness of the model. On the downside, this implies a lengthier process since training is a computationally demanding task that typically takes a few hours on a standard GPU.

**Table 3.** Hyperparameters of the encoder and decoder neural networks.

| Hyperparameter | Decoder | Encoder |
|---|---|---|
| Activation | ReLU | ReLU |
| Range of kernel sizes | $1 \times (2-7)$ | $3 \times (3-7)$ |
| Number of input channels | 1 | 2 |
| Number of inner channels | 30 | 30 |
| Number of linear layers | 3 | 0 |
| Number of neurons in linear layers | 60 | - |
| Total number of parameters | 35000 | 150000 |
| Padding mode | *replicate* | *replicate* |
| Loss | MSE | MSE |
| Optimizer | Adam | Adam |
| Batch size | 250 | 15 |
| Number of epochs | 3 | 200 |
| Learning rate (initial) | 1e-3 | 1e-3 |
| Optimizer epsilon | 1e-8 | 1e-6 |
| Scheduler step / rate | 0.6 / 0.2 | 90 / 0.5 |
| Parameter initialization | Xavier | Xavier |

## 5 Results

### 5.1 Training convergence and accuracy

This section is dedicated to the evaluation of the pipeline and the verification of its convergence, which is a necessary step before examining the retrieved latent variables themselves.

#### 5.1.1 Decoder

The training of the decoder networks was successful, with the loss function decreasing with the number of epochs until it plateaus. It was verified that increasing the training set size did not result in a change of this plateau value, meaning that the training dataset was large enough for the chosen neural network's complexity. Examples of model outputs on the synthetic testing set are shown in Fig. 3.

Since the loss function (MSE on normalized spectra), is not easily interpretable to assess the model's performance, the following metric was defined to quantify the overlap of two spectra: $O(S, \tilde{S}) = 0.5 \left[ \frac{\int min(S^*, \tilde{S}^*)}{\int S^*} + \frac{\int min(S^*, \tilde{S}^*)}{\int \tilde{S}^*} \right]$ where $S^* = S - min(S)$ and $\tilde{S}^* = \tilde{S} - min(S)$; $O(S, \tilde{S})$ is equal to 1 (or 100 %) when the spectra are perfectly identical, and to 0 if they are disjoint. A detailed illustration of this metric is provided in Appendix D.

On the synthetic testing set, the overlap for X-band (resp. W-band) is of $90.7\%$ (resp. $94.8\%$), as an average over 5 different runs with random initialization. This reflects a good, although not perfect, performance of the algorithm. Looking at a few examples of individual spectra, it comes across that the model has a slight tendency to underestimate the peak of the spec-
445 trum (see for example the X-band spectrum chosen in Fig. 3), despite the secondary loss that was used; however, the rather good overall agreement between target and output spectra suggests this is not a critical issue. Additionally, we observe that performance is slightly worse for X than for W-band spectra, in spite of some efforts to adjust the neural network architectures independently to improve each model's accuracy. This could be related to the fact that W-band spectra have a lower noise level, meaning the actual signal —the peak— occupies a larger part of the spectrum than for X-band, which could in turn facilitate
learning.

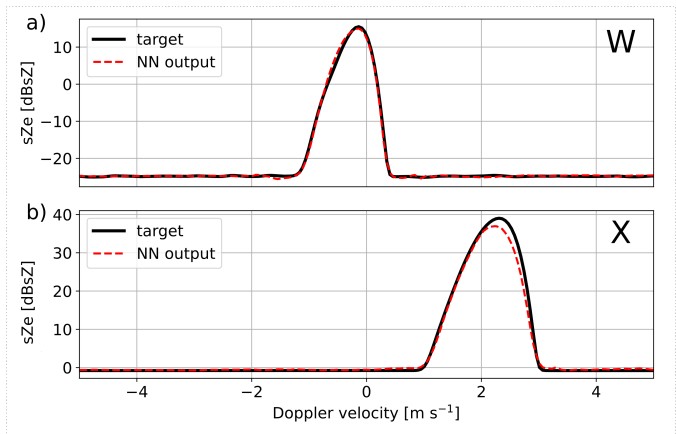

**Figure 3.** Examples of results on the synthetic testing set of the decoders, showing the decoder output (dashed red) and the target PAMTRA-generated spectrum (black line) at a) W- and b) X-band. The examples were chosen to reflect some of the typical behaviors and possible artifacts that were observed; the X- and W-band examples do not correspond to the same microphysical properties. Units of spectral reflectivity are used, i.e. $1\,\mathrm{dBsZ} = 10\log_{10}(1\,\mathrm{mm}^6\mathrm{m}^{-3}(\mathrm{ms}^{-1})^{-1})$. Doppler velocity is positive for downward motion.

### 5.1.2 Encoder

Training of the encoder is also successful: the full pipeline is able to reconstruct original spectrograms in a satisfactory manner, as is visible in Figure 4. Only W-band spectrograms are shown but results are visually very similar at X-band. The overlap metrics are slightly below the ones of the decoder alone (respectively 86% and 91% for X- and W-band spectrograms); this
slight decrease can be expected for several reasons: first, the real spectrograms include high-shear regions with significant turbulent broadening (which can be visually identified as regions with suddenly much wider spectra, along with variable velocity, e.g., in Fig. 4 below 500 m), which the model cannot be expected to resolve perfectly. Then, some time steps include bimodal spectra (e.g. 2.5 km - 3.5 km) which the model in its current state is unable to replicate. When looking only at spectrograms with moderate apparent turbulence (e.g. 0.5 km - 1.5 km), and strict unimodality, the overlap metrics are similar

to the decoder. Finally, it is also empirically observed that in some cases (Fig. 4 e) ) the model can slightly underestimate the peak of the spectrograms, which is a propagation of the decoder behavior. As a safety check, it was also verified that running PAMTRA on the latent variables also led to spectrograms close to the original ones, as displayed on Fig. 4 c).

Let us point out at this stage that unlike most machine learning models, which are trained on a dedicated dataset and then implemented on independent data, the encoder is here trained directly on the data of interest. Indeed, the aim is not to create a generic model that can be used to retrieve microphysical variables from any dual-frequency spectrogram: the aim is to find the latent variables which minimize the reconstruction error on specific measurements; the encoder can learn any relevant feature from the input data to achieve this goal. In that sense, *overfitting* the data, which can be an issue in usual machine learning problems, is not a concern when training the encoder. It is however preferable to train the model on a large enough dataset rather than just a few spectrograms: this reduces the risk of converging toward local minima that would correspond to non-physical combinations of microphysical parameters.

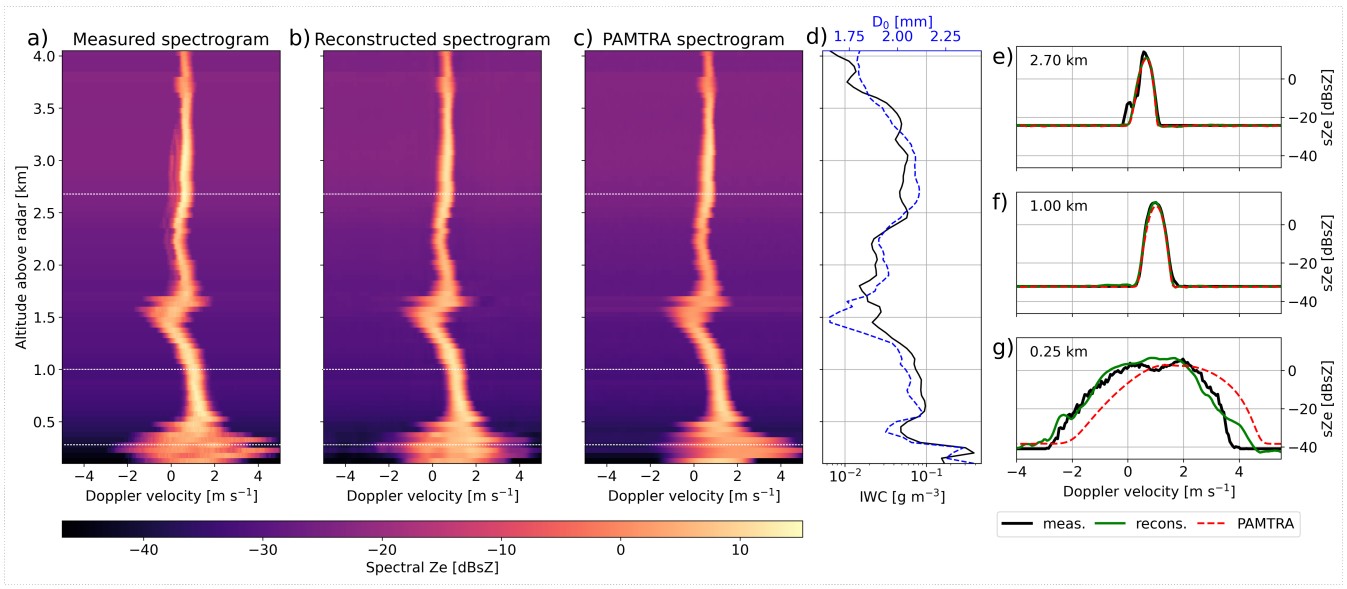

**Figure 4.** Examples of W-band spectrograms: a) measured, b) reconstructed through the pipeline, c) reconstructed in PAMTRA from the learnt latent features. Corresponding spectra from selected altitudes are displayed in panels e), f) and g); they are indicated with dashed white lines in the spectrograms. Units of spectral reflectivity are used, i.e. $1\,\mathrm{dBsZ} = 10\log_{10}(1\,\mathrm{mm}^6\mathrm{m}^{-3}(\mathrm{ms}^{-1})^{-1})$. The reconstruction of X-band spectrograms (not shown) is of similar quality. Panel d): $IWC$ and $D_0$ profiles retrieved from these spectrograms.

## 5.2 Qualitative assessment of the retrieval

### 5.2.1 Microphysical parameters

This section presents a qualitative perspective on the retrieval results, based on a snowfall event that took place on January 23, 2021. Figure 5 features timeseries of some relevant radar measurements (left column) and retrieved microphysical variables (right). The radar data include $Ze_X$, the dual-frequency reflectivity ratio $DFR_{XW}$, Doppler velocity and spectral width (at W-band), as well as the hydrometeor classification from MXPol (cf Sect. 3). Let us highlight that the latter classification, derived from polarimetric variables, is completely independent from WProf and ROXI spectrograms. The microphysical variables included are IWC, $D_0$, $b_m$, $\beta_a$ and $A_r$. The variables $a_m$ and $\alpha_a$, not shown, are highly correlated to respectively $b_m$ and $\beta_a$ (see e.g. Fig. 10 and B4).

A first general observation from the retrieval timeseries is the persistence of spatio-temporal structures visible in the radar data, like the fall streaks. While the pipeline explicity took into account the *spatial* consistency of the measurements —through the use of convolutions—, the *temporal* features are never used in the training of the model. It is thus reassuring that the full spatiotemporal features are well captured by the retrieval method.

The retrieved values are also fairly consistent with the physical interpretation that stems from the radar measurements. IWC correlates quite strongly with $Ze_X$ values, i.e. large IWC values are retrieved for strong reflectivity measurements (e.g., around 15:10 and 15:50 UTC). The size parameter $D_0$ also matches the intuition, with small diameters near cloud top, and some localized pockets with large values e.g. around 15:10 between 1 and 2 km range, which correspond to regions of large dual-frequency ratio. The $D_0$ timeseries also agrees seemingly well with the hydrometeor classification that tends to identify aggregates in regions where $D_0$ is larger (with for example the same fall streak around 15:10 UTC).

The $\beta_a$ exponent of the area-size relation features smaller values when $Ze_X$ and DFR are low, which is compatible with small non disk-like particles such as columnar crystals, while larger values could indicate aggregates or rimed snowflakes.

Somewhat more noisy are the mass-size exponent $b_m$ and the aspect ratio $A_r$, although their values and spatial trends still seem reasonable. They are rather correlated, which is not unrealistic: particles with aspect ratio near 1 are rounder and thus closer to spheres, which in turn, have a $b_m$ close to 3. Indications of riming in the hydrometeor classification (visible as yellowish-red regions) roughly correspond to regions with larger $A_r$ and $b_m$, as expected from rimed particles (e.g., 15:10 between 0.5 and 1 km, 15:50 between 0 and 1.5 km, 16:00 around 1 km). Additionally, small values of $b_m$ and $A_r$ are retrieved near cloud top, consistent with crystal-like particles, while fall streaks where high $D_0$ values point to aggregation (15:00 to 15:20) also have medium-high $b_m$ and $A_r$. A few time steps stand out with large values of $A_r$ and $b_m$, coinciding with regions where large spectral width and variable $v_{Dopp}$ suggest strong turbulence (16:00, 1 km). In such high-turbulence cases, the retrieval cannot be expected to perform perfectly since the shape of the spectra is then largely dominated by turbulent broadening.

Let us add a few words about correlations between certain variables. As mentioned before, some expected consistent behaviors are observed in the retrieval like the apparent correlation between $b_m$ and $A_r$, or between $D_0$ and $\beta_a$. This is not in any way

enforced by the pipeline, since those variables are prescribed independently when generating the training set. The correlations between $a_m$ and $b_m$ (and $\alpha_a$ and $\beta_a$), which are also expected, are slightly different: when building the training set (cf Sect. 3.1 and Appendix B1), these variables were sampled in a correlated way —with some noise included— to avoid completely unrealistic combinations, and this may therefore influence the retrieval; however, the variables are not explicitly constrained during the training of the encoder: it is therefore reassuring to see that the model output still follows the expected correlations.

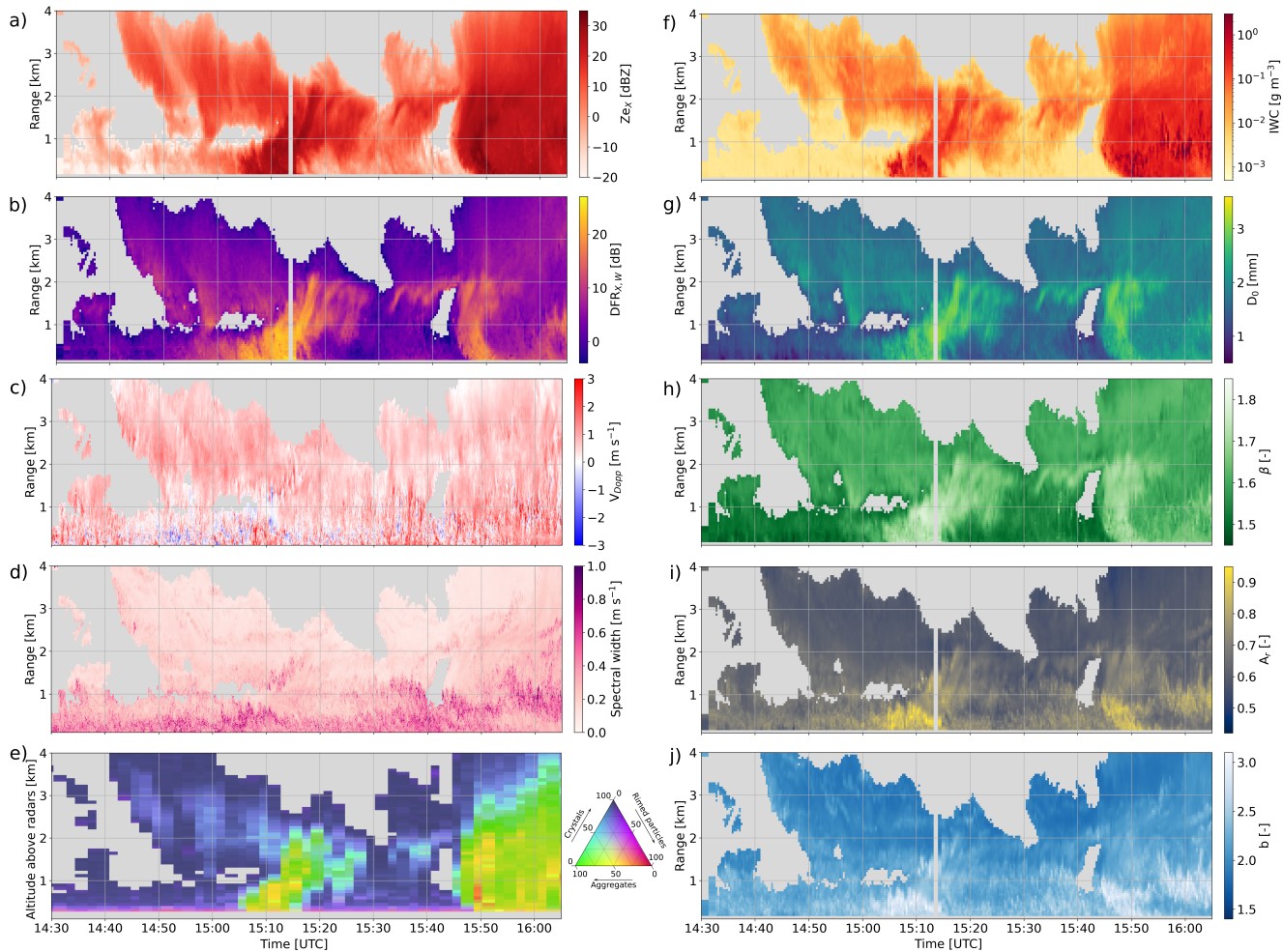

**Figure 5.** Height-time plots of radar measurements and microphysical retrievals. The left panels contain radar data: a) $Ze_X$, b) $DFR_{XW}$, c) W-band mean Doppler velocity, d) W-band spectral width, e) Timeseries of hydrometeor classification with demixing showing the proportion of the three main particle types identified: here aggregates, rimed particles and crystals (cf. Section 3.3). The right panels feature microphysical retrievals: f) ice water content, g) size parameter $D_0$, h) area-size exponent $\beta_a$, i) aspect ratio $A_r$, j) mass-size exponent $b$.

### 5.2.2 Other retrieved variables

In addition to the microphysical descriptors, the latent features comprise other quantities which are required by the pipeline in order to reconstruct the spectrograms (cf. Table 1). These are not designed to serve a proper physical interpretation, but their behavior should still be assessed.

The noise level is only related to instrument properties and range gate, and in no way to microphysical or atmospheric processes. As visible in panels a) and b) of Fig. 6, the noise level estimates are exactly what could be expected and reflect the evolution of the radars' sensitivity with range. At W-band, the abrupt change of sensitivity around 900 m range is due to the change of chirp table. Some artifacts are also visible (panel b), 16:00 at 1 km), in the same regions of Fig. 5 where the other retrieved values visually also appeared less reliable. This is possibly related to the presence of strong turbulence in these areas —as suggested by enhanced spectral width and varying mean Doppler velocity—, which can indeed be expected to affect the retrieval accuracy.

The radial wind estimates serve to artificially correct for shifts of the Doppler spectra caused either by vertical wind (up- or downdrafts), contamination by horizontal wind in the case of imperfectly vertical radar beams, or by biases in the velocity-size relation of the forward model. Their interpretation as a physical atmospheric quantity should thus be avoided. However, it is rather reassuring to see in panels c) and d) of Fig. 6 that the X- and W-band are not too far off and especially that their cofluctuation is satisfactory: the opposite would be a problem since the Doppler velocity timeseries of both radars are rather similar (not shown). Likewise, the broadening parameters are similar in X- and W-band, and also somewhat follow the spectral width (Fig. 5 d). We recall that the broadening parameters are not expressed in physical units, but as the size of the Gaussian kernel that results in the observed broadening. This is kept as such to highlight that these variables include all the broadening causes (not just turbulence, but possibly also horizontal wind, etc.) and are rather a side-product of our retrieval than descriptors of actual atmospheric dynamics. A reasonable agreement is found (not shown) when comparing these values to broadening estimates derived through classical methods (Borque et al., 2016; Shupe et al., 2008), which rely on the variability of mean Doppler velocity and on wind profiles.

### 5.3 Comparison to in-situ data

In this section, we take a step further in the evaluation of the retrieval by performing quantitative comparisons with airborne in-situ measurements.

#### 5.3.1 Ice water content

Figure 7 illustrates retrieval results of ice water content in comparison with in-situ estimates, computed as $IWC = TWC - LWC$ (cf. Sect. 3.3). In Fig. 7 a) and b) are displayed the timeseries of ice water content, first as a time-height plot to which the aircraft trajectory is added, then along the aircraft trajectory to which retrieval outputs are overlaid at timesteps of overpasses.

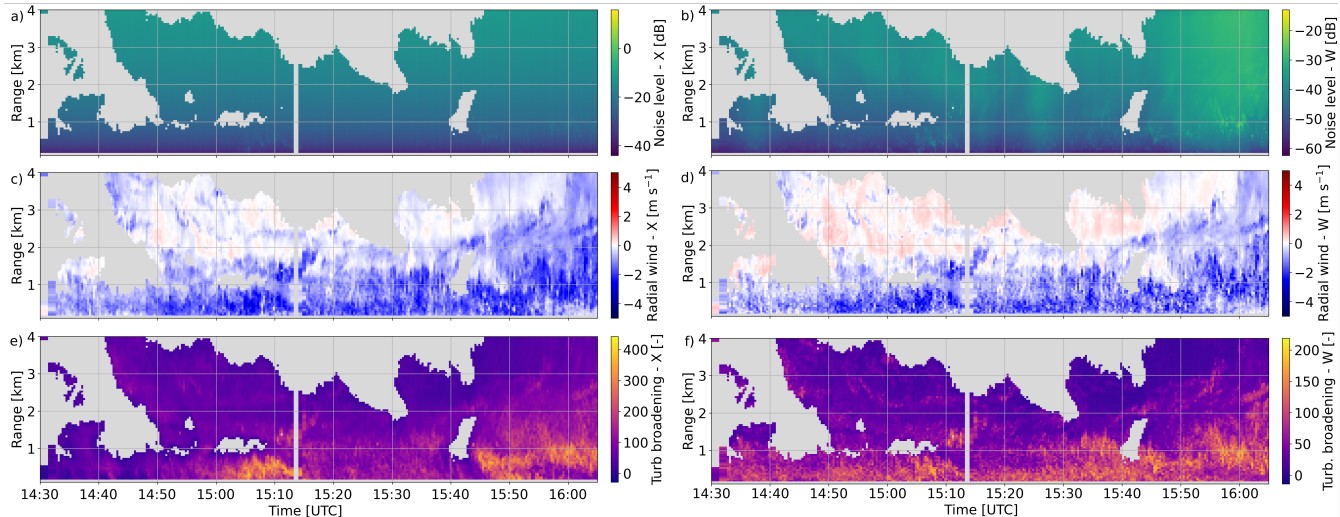

**Figure 6.** Time-height plot of additional retrieved variables: a) (resp. b)) Noise level at X- (resp W-) band; c) (resp. d)) radial wind at X- (resp. W-)band; e) Broadening at X- (resp W-)band

The comparison is overall good, with satisfactory cofluctuations as well as reasonable agreement in the values themselves. For reference is also displayed the IWC retrieved from RASTA measurements (Delanoë et al., 2007, cf. Section 3.3)), which appears slightly more variable. In Figure 8 a), the scatterplot of retrieved to measured IWC combines the results from the three flights; the points are color-coded with $Ze_X$ to illustrate that large IWC corresponds to large reflectivity, as expected and already noted in the qualitative analysis. The error bars illustrate the ensemble spread (standard deviation) of the retrieval realizations as described in Sect. 4.3 This scatterplot confirms the robustness of the retrieval results and their good correlation to the measured IWC (R = 0.87 in logarithmic scale), with, however, the existence of a slight bias toward low values (-0.19 in logarithmic scale). Surely, the spread of the values remains substantial, sometimes within orders of magnitudes, but it should be kept in mind that, even at times of overpasses, the aircraft is not perfectly colocated with the radar measurements, and that the sampled volumes are not identical; additionally, the single-frequency RASTA retrieval seems to have an even larger variability than our retrieval.

### 5.3.2 Size parameter $D_0$

Aircraft measurements do not provide a variable that can directly be compared to the $D_0$ retrieved through our method. Hence, we use the values of $D_0$ derived from the exponential fit of the in-situ PSDs (cf. 3.3, Fig. 9 a). In order to monitor the validity of this approach, the correlation coefficient of the fits are also included in the timeseries, and are typically very high (often R > 0.9). Our retrieval is superimposed to the timeseries, and compared to the in-situ values in Fig. 8 b) using all available flights. While this was not perceptible in the qualitative analysis, $D_0$ retrievals actually show a strong bias (+1.3 mm) when compared to aircraft measurements, leading to an overestimation of particle size. An investigation of possible causes for this behavior is

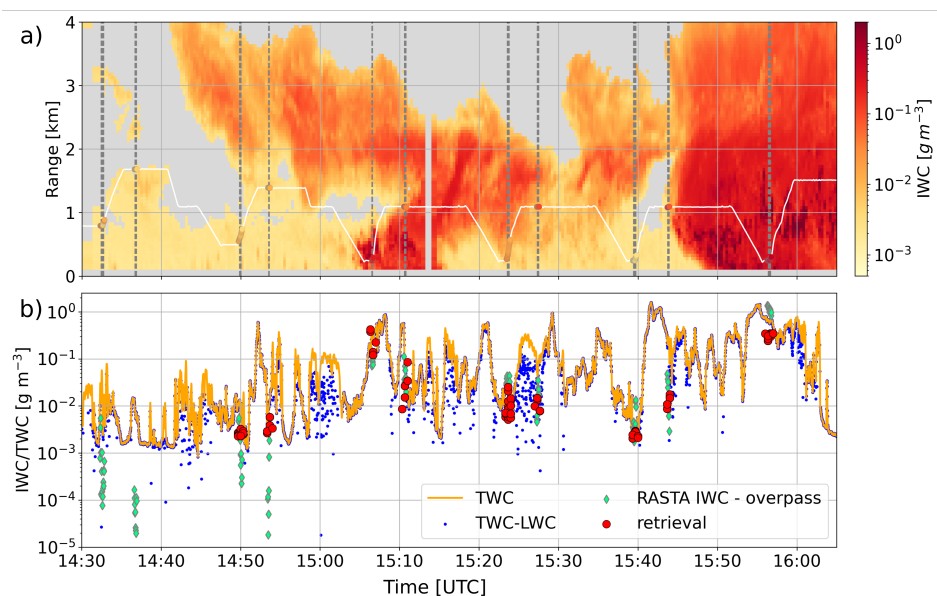

**Figure 7.** a) Time-height plot of IWC retrieval, to which the aircraft trajectory is overlaid as altitude as a function of time; aircraft IWC values at timesteps of aircraft overpasses (horizontal distance smaller than 1 km) are shown as scattered points. Dashed vertical lines indicate when the aircraft is within 500 m of horizontal distance to the radars. b) Timeseries of water content measured by the aircraft ($TWC$ and $TWC - LWC$) and overlaid radar retrieval.

proposed in Section 6. This being noted, the cofluctuation between retrieved and in-situ $D_0$ is nonetheless good (R = 0.74), which gives confidence that the retrieval is still highly relevant for process-oriented studies: there, even more than the actual values, the changes and evolution of particle size can indicate the occurrence of specific snowfall growth or decay mechanisms.

### 5.3.3    Mass- and area-size relations

Mass- and area-size power law coefficients are explicitly computed from the aircraft measurements and can therefore be com-
pared to our retrieval. However, the timeseries of these aircraft quantities are highly noisy and thus point-to-point comparisons did not appear meaningful; it was therefore preferred to perform a statistical analysis. For each flight, we compare the histogram of $b_m$ (resp. $\beta_a$) sampled by the aircraft during its entire flight (except for the part of the flight to and from the campaign location), to the histogram of retrieved $b_m$ (resp. $\beta_a$) above the radars, during the time frame of the flight and in the altitude range sample by the aircraft, which excludes for instance regions near cloud top. The histograms of $b_m$ agree rather well (Fig.
10 a)), with a similar mode around 2.2, although fewer values below 2 are retrieved. In Fig. 10 b), the histograms again have relatively close peak values (around 1.6 for the retrieval and 1.7 for the aircraft). There however, and for $b_m$ to a lesser extent, the retrieved values histogram is much narrower than the aircraft one. This is not too surprising, given how noisy the aircraft measurements are, and considering that the volume sampled by the PIP and 2DS probes is much smaller than the radar volume

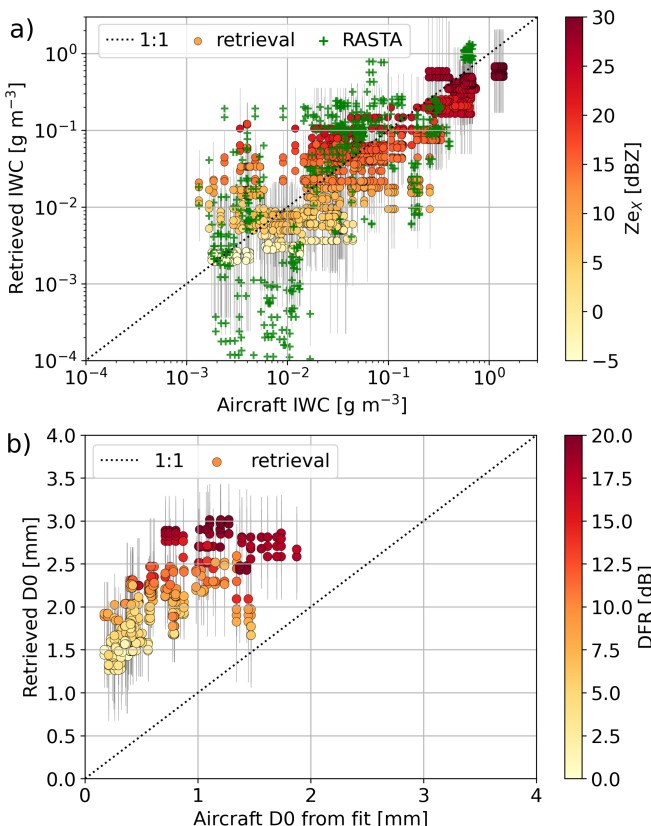

**Figure 8.** Scatter plots of retrieved vs. aircraft measurement of a) IWC and b) size parameter $D_0$. Each point corresponds to a time step when the aircraft is within 1 km horizontal distance to the radars. Three flights are used (Jan 22, Jan 23, Jan 27). Color indicates corresponding a) $Ze_X$ and b) DFR. The black vertical lines indicate the standard deviation of the retrieval.

—which automatically increases the variability and flattens the distribution. With this in mind, these histograms support a rather good consistency of the retrieval with the aircraft measurements.

In addition, we verify that the relations between $a$ and $b$ (resp. $\alpha$-$\beta$), retrieved and measured, are consistent: this is visible in Fig. 10 c) (resp. d)), where the scatterplots of $a$ vs. $b$ (resp. $\alpha$ vs. $\beta$) are overlaid. Although not perfect, the match is reasonable. Let us highlight that once again, retrieved $\beta$ values are narrower, consistent with the histograms.

### 5.3.4  Aspect ratio

The last microphysical variable for which we can perform a comparison is the mean aspect ratio: similarly to the mass-size exponent, Figure 11 a) displays the histogram of retrieved and aircraft values. A significant difference is visible in the modes, with the aircraft values around 0.45 and the retrieval mode around 0.6; this however is consistent with the difference in the definitions of aspect ratio in each case. The aspect ratio retrieved through our pipeline is $Ar_v$ defined as the ratio of

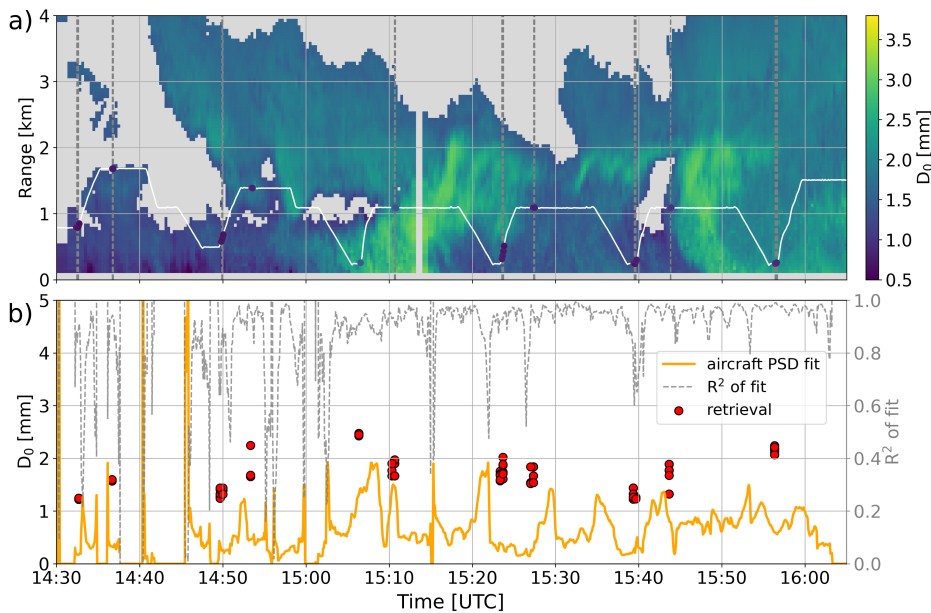

**Figure 9.** As for 7, but for $D_0$.

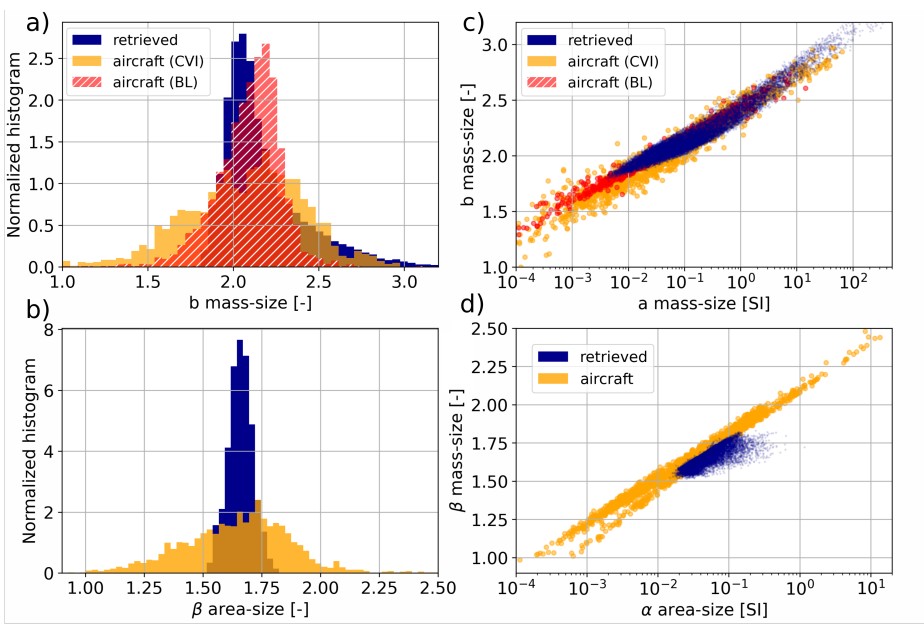

**Figure 10.** Histograms of a) mass-size and b) area-size exponents. Scatter plots of c) mass-size and d) area-size exponent to prefactor, from retrieval and aircraft measurements (cf. Sect. 3.3)

particle dimension along vertical to maximum dimension, whereas the aircraft measurement is $Ar_\perp$ is the ratio of minor axis length to maximum dimension. Relating both quantities is not directly possible without having additional information on particle orientation, but an intuition can be gained from Fig. 11 b) where the relation between $Ar_v$ and $Ar_\perp$ is shown for particles randomly oriented within a certain angle (90° corresponds to completely random orientation). Using the relations of Fig. 11 b), a transformed histogram is included into panel a) showing the equivalent aircraft $Ar_v$ assuming ellipsoidal particles with random orientation within 75°: it fits rather well with the retrieval. While this is not per se rigorous, it gives a qualitative understanding of the observed discrepancy. Note that the aspect ratio values derived from the in-situ images are themselves prone to some bias, as discussed in Jiang et al. (2017), due to the projection of 3-dimensional particles in a 2-dimensional space. This bias would however be opposite to what is observed here, and is likely not dominant in our study. Another element to consider is that aspect ratio is assumed to be the same across the particle size distribution, which may well be an oversimplification; in particular, smaller particles may have smaller aspect ratios which would affect the aircraft-derived quantity differently than the radar-based estimate.

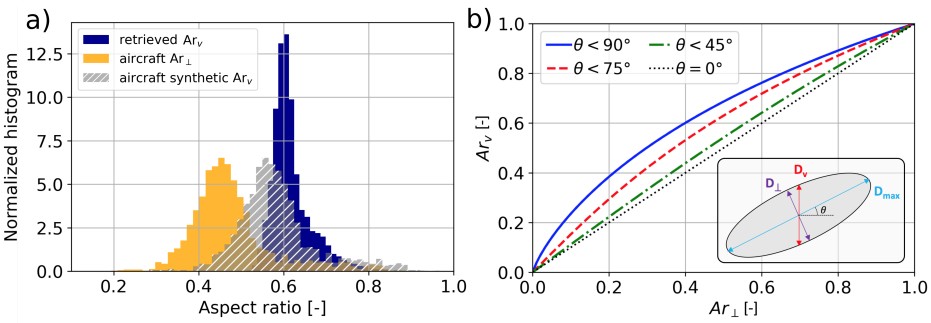

**Figure 11.** a) Histogram of retrieved and aicraft-measured aspect ratio. b) Illustration of the relation between $A_{r,\perp}$ and $A_{r,v}$ for particles with random orientation within a given angle $\theta$; the various quantities are sketched in the bottom right of panel b).

## 6  Discussion

While the results are overall encouraging, the previous section highlighted some points that call for further discussion. This section investigates the sensitivity of the pipeline to certain key hypotheses and provides some insight into possible causes for the bias in $D_0$.

### 6.1  Sensitivity to miscalibration and differential attenuation

One limitation of our framework is that it requires a good calibration of the radars —both absolute and relative— as well as an independent correction of attenuation. As detailed in Appendix A, the issue of attenuation was here tackled by implementing a correction of W-band reflectivity based on estimates of gaseous, snowfall and liquid water attenuation. This correction method is however error-prone and we cannot exclude that reflectivity biases are present in the measurement dataset. The presence

of supercooled liquid water cloud layers or wet snow can be particularly difficult to identify and diagnose, while attenuating strongly millimeter-wavelength signal (with e.g. path-integrated attenuation up to 5 dB for liquid water paths of 500 g m$^{-2}$, Kneifel et al., 2015). To assess the possible importance of inaccurate calibration or attenuation correction on the retrieval, we investigate its sensitivity to reflectivity offsets, both absolute and relative.

In Fig. 12 are displayed the mean bias of retrieved IWC and $D_0$, computed as the mean difference between retrieved values
and aircraft measurements, when a constant offset in reflectivity is added to the input X- and W-band spectrograms. The following behavior is observed, in accordance with previous qualitative observations: IWC is especially sensitive to X-band reflectivity (as illustrated in panel b and by the rather horizontal color gradient in panel a). Rather, $D_0$ is more sensitive to differential offset i.e. to changes in the DFR (as illustrated in panel d and by the rather diagonal color gradient in panel c).

While this $Ze$ calibration is undoubtedly a key factor in the uncertainty of the algorithm, it does not appear to cause extreme
divergence in the retrieval: the changes in $D_0$ and $IWC$ shown in Fig. 12, while not negligible, are also not massive. In particular, $Ze$ miscalibration could not explain solely the observed $D_0$ discrepancy: changing the $DFR$ of $-6dB$ —which is substantial— only brings down the bias from 1.3 mm to around 1.0 mm.

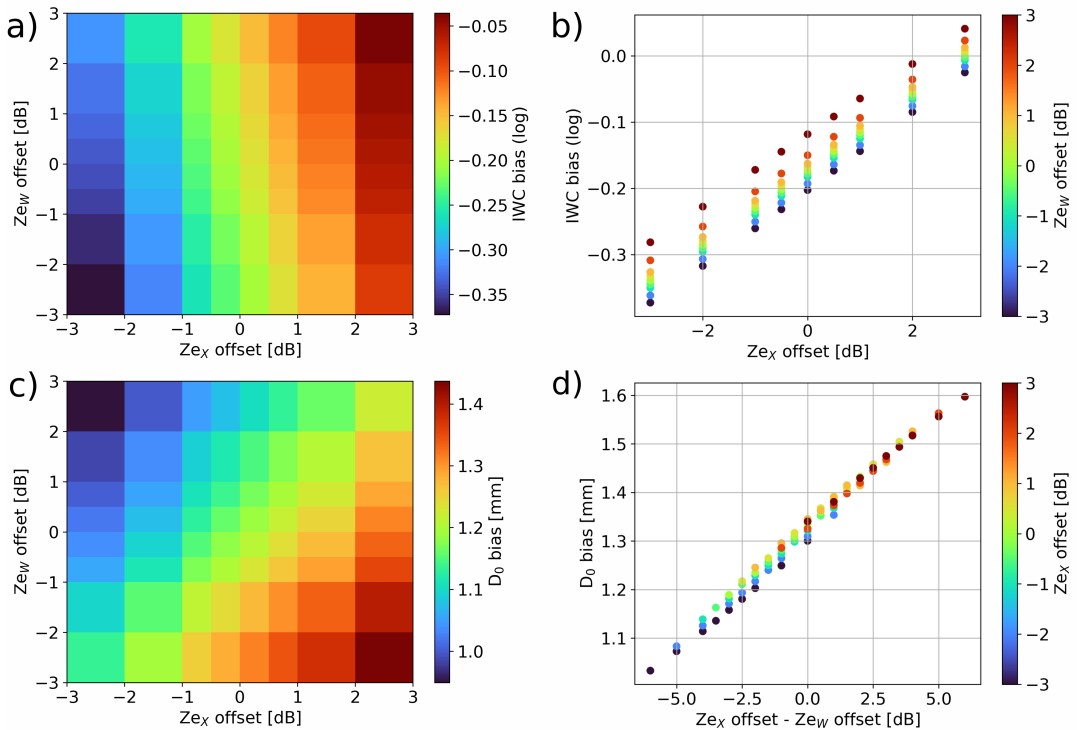

**Figure 12.** a) Heat map of IWC bias as a function of X- and W-band reflectivity offset; the bias is computed using the aircraft values as reference b) Different visualization showing IWC bias as a function of $Ze_X$ offset. c) As for a), but for $D_0$. d) As for b), but for $D_0$ and offset in dual-frequency ratio.

We note (not shown) that shifting the spectra by constant or relative velocity offsets, to mimic one of the effects of radar mispointing, only minimally affects the retrieval of microphysical properties, and mostly translates into changes in the retrieved radial wind.

## 6.2 Training set limitations

Another aspect of our framework which could cause a bias in the retrieval is if the training set is too narrow. While special attention was paid to this potential issue as the microphysical parameters were sampled from the MASC database, there is likely still room for improvement. In particular, the size cutoff for good-quality images in the MASC is quite high and very few particles with a diameter below 0.5 mm are accurately captured. For reference, Fig. 13 illustrates the histogram of $D_0$ derived from MASC measurements (Grazioli et al., 2022), and by the aircraft 2D-S and PIP probes during the ICE GENESIS campaign. The limitation is apparent: the aircraft is able to capture much smaller particles, but not beyond a certain size, while the MASC can detect large snowflakes but very few small particles. The framework would thus probably benefit from training the decoder on a larger dataset, that would include a better representation of this smaller particle range. It is yet unlikely that this would entirely resolve the size bias, for there is still an overlap between the aircraft-measured size range and that on which the model was trained.

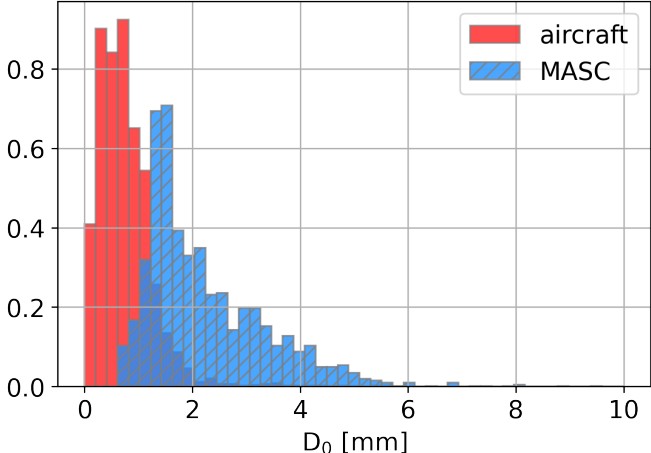

**Figure 13.** Histogram of $D_0$ from aircraft measurements (PIP and 2D-S) during ICE GENESIS (red) and from MASC measurements (MASCDB)

## 6.3 Scattering model

Surely one of the strongest hypotheses on which the pipeline was built is the parameterization of the scattering model in forward simulations. As explained in Sect. 3.1.1 and Appendix B2, the default version of PAMTRA was used which to this date (28/06/2021) assumes constant values for certain parameters of the SSRGA, and allows to change two coefficients ($\kappa$ and

$\beta$, cf. Hogan and Westbrook (2014) and online documentation (PAMTRA team, 2015)). Several studies suggest however that more parameters are needed, and moreover that their values can vary significantly (e.g. Leinonen et al., 2018a; Ori et al., 2021) depending on particle type, shape, etc.

To get an empirical sense of how this could affect the retrieval, the approach described hereafter was followed. First, a few time and range gates were randomly selected from the dataset. The corresponding retrieved values were then modified by adding a $D_0$ offset ranging from -1.5 to +2 mm, and PAMTRA simulations were run on the microphysical parameters obtained, using the same settings as in Sect. 3.1.1. Parallel to that, slight modifications of the PAMTRA code were conducted to allow for modification of the four literature coefficients of the self-similar Rayleigh-Gans approximation ($\kappa$, $\beta$, $\gamma$, $\zeta$); new simulations were run for the selected time and range gates, keeping the retrieved microphysics unchanged but randomly changing the SSRGA parameters within $\pm 10\%$ of their original values. As seen in Appendix B2, this is well within the typical variability of the coefficients calculated from simulating various types of particles (e.g. Leinonen et al., 2018a, , Fig. 5). A visual inspection suggests that changes in $D_0$ and changes in SSRGA coefficients mostly affected the amplitude of the spectra: hence, the influence of these changes were measured by the change in the scalar total reflectivity at X- and W-band, then in the dual-frequency ratio. The obtained results are illustrated in Fig. 14 (detail in the caption): they suggest that moderate changes in the SSRGA parameters could have an impact similar as varying the size parameter of approximately 1 mm (-0.6 to +1.5 mm), which is a significant change. Taking this investigation a step further, the influence of each of the four parameters can be computed independently, by following the same steps but changing only one coefficient: it appears that the output is most sensitive to $\beta$ and $\gamma$, which each cause amplitude changes corresponding to $\Delta D_0$ of at least $\pm 0.4 mm$. Obviously, this empirical analysis cannot be directly translated into a quantitative interpretation; yet it highlights that the scattering model can have a substantial influence on the retrieval. This leads us to believe that the $D_0$ bias observed when comparing our retrieval to aircraft measurements is partly caused by an inaccurate or insufficient parameterization of the radiative transfer model. In order to remedy this effect, a forward model with a more subtle parameterization is likely required when designing the decoder training set.

## 6.4 Shape of the particle size distribution

Another underlying hypothesis that was made when designing the pipeline and defining the set of microphysical descriptors was to consider only exponential particle size distributions. This choice was made to keep a minimal number of retrieved parameters at this stage. It is however known that multi-frequency signatures, on the one hand, and Doppler spectra, on the other hand, are both affected by PSD shape (e.g., resp. Mason et al., 2019; Barrett et al., 2019). Here, we conduct a similar analysis as in the previous subsection: this time, as changes in the PSD shape mostly influence the shape of the spectrum, we focus on the skewness of the W-band spectrum (instead of the DFR) as a metric to understand how changing the shape of the PSD could influence the retrieval. A gamma distribution was assumed ($N(D) = N_0 D^\mu \exp(-D/D_1)$), e.g. Petty and Huang, 2011), constraining $D_1$ by keeping the effective diameter constant ($D_{eff} = 3D_0$), and varying the shape parameter $\mu$ in the range [-2, +5] as observed in snowfall (Mason et al., 2019). Figure 15 illustrates that changes in the PSD shape may have a similar effect on W-band spectrum shape as varying $D_0$ of approximately 1 mm (-0.9 to +1.5 mm). Here again, this observation is mainly

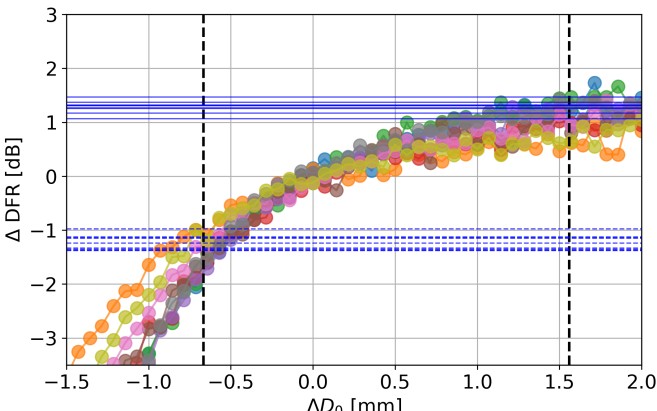

**Figure 14.** Colored lines with scattered points: $\Delta SDFR$ caused by adding a diameter offset $\Delta D_0$ on microphysical descriptors of selected (time, range) gates. Horizontal lines: for each of these (time,range) gate, maximum $\Delta DFR$ (positive and negative) caused by a modification of the SSRGA coefficients within $\pm 10\%$. For each selected (time,range) gate, the intersection of the horizontal and colored lines give a $\Delta D_0$ value which causes the same change in DFR as a change in SSRGA coefficients (worst case). Dashed vertical lines show the mean of these $\Delta D_0$ values.

qualitative and cannot be directly used to quantify the influence of PSD shape on the retrieved microphysical descriptors, but it does underline that considering more complex distributions would be necessary to further refine the framework and that the assumption of an exponential behavior may also have a role in the observed $D_0$ bias.

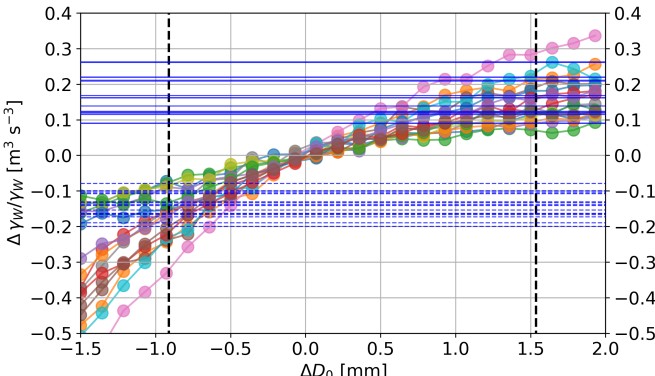

**Figure 15.** Colored lines with scattered points: relative change in W-band skewness $\gamma_W$ ($\Delta \gamma_W / \gamma_W$) caused by adding a diameter offset $\Delta D_0$ on microphysical descriptors of selected (time, range) gates, if assuming an exponential PSD. Horizontal lines: for each of these (time,range) gates, maximum relative change in W-band skewness caused by a modification of the PSD shape (assumed a gamma distribution, $\mu$ in the range [-2, +5], cf. text.). For each selected (time,range) gate, the intersection of the horizontal and colored lines gives a $\Delta D_0$ value which causes the same relative change in skewness as a change in PSD shape (worst case). Dashed vertical lines show the mean of these $\Delta D_0$ values.

In addition to these important hypotheses —SSRGA scattering model and assumption of exponential size distributions—, we recall that other modeling choices were made during the design of the synthetic dataset and the underlying physical framework, such as assumptions on particle orientation, velocity-size relation, etc. (cf Sect. 3.1.3) which are inevitably a simplification of the physical reality and may thus also influence the retrieval. Another point should be briefly mentioned regarding small particles, which are Rayleigh scatterers at both X- and W-band. This means that if a population is composed entirely of small particles, the influence of particle size and number concentration is hardly distinguishable in the spectrograms. The ill-posedness of the problem is reinforced and the retrieval could be expected to have a reduced accuracy, even if the training set and scattering model were improved.

## 6.5 Comparison to other frameworks

In this section, we discuss more broadly the pipeline that was developed, in comparison with other possible approaches. As argued in Sect. 2, we believe that the framework introduced here is a key aspect of this work.

In order to support this point, a *direct* deep learning inversion model was also designed: it essentially consists in learning the inverse of our decoder, similar to the approach of Chase et al. (2021). It is presented in Appendix C and the results show that although still respectable, this direct retrieval is noisier and less accurate than our model due to its ill-posedness. For instance, when comparing retrieved values to in-situ measurements of IWC, the correlation coefficient drops from R = 0.87 (with the proposed method) to R = 0.59 (with this direct inversion). Similar behaviors are observed for the other retrieved variables.

Let us mention an alternative approach that could be used, which lies half-way between classical OE and the proposed method. The notations used here are those of Sect. 2. Once a differentiable approximation of the forward model is known ($\tilde{\mathbf{f}}$), another way to look for $X_r$ is to find the minimizing argument of $||\tilde{\mathbf{F}}(X) - Y_r||^2$ using gradient descent; a regularizing term can be added to ensure for instance the spatial continuity of $X$ or to enforce some degree of spatio-temporal smoothness. This requires only one deep learning model instead of two, and could thus seem more appealing, but the first approach was preferred. Indeed, by actually learning an approximation of the inverse mapping $\tilde{\mathbf{G}}$, and doing so on a large dataset, the risk of reaching a local minimum in $X$ is reduced. Our method also does not require any explicit prior assumption on $X$, nor on any property of the latent space, like spatial smoothness; rather, it is constrained by the spatial structure of the observed signal itself.

## 7 Conclusion

In this work, we proposed a new method for the retrieval of seven microphysical properties of snowfall from dual-frequency Doppler radar spectrograms. To our knowledge, no previous method allowed for the joint retrieval of these descriptors, and with this high spatial and temporal resolution. Some typical challenges of Doppler spectral retrievals were overcome, like the need for perfectly vertical beam alignment, or the requirement of very low turbulence, thus allowing for microphysical

retrievals in a larger range of atmospheric conditions. The approach relies on a two-step deep learning framework: a *decoder* network serves as a differentiable gate-to-gate emulator of a known radiative transfer model, while the *encoder* network learns to map the Doppler spectrograms to full profiles of microphysical variables. The algorithm could be assessed thoroughly by confronting the retrieved quantities to in-situ aircraft measurements which were conducted during the 2021 ICE GENESIS campaign. Overall, the comparisons with in-situ data are highly encouraging and support the validity of the framework: good co-fluctuations as well as similar statistics are reported. Certain discrepancies were nonetheless observed: in particular, the retrieved values of the size parameters are affected by a bias, for which possible explanations were proposed. They point to limitations in the training set itself (in which small particles are under-represented), and to assumptions in the scattering model (which relies on the SSRGA) or the parameterization of the PSD (as an exponential distribution). These analyses open up for possible improvements of the retrieval, particularly along the line of radiative transfer modeling. Meanwhile, in spite of these limitations, the method can provide relevant insights into snowfall properties in the perspective of process-oriented studies whose focus is typically the relative spatial and temporal evolution of microphysical variables, rather than their exact numerical values.

We highlight that one drawback of the algorithm in its current state is that it relies on attenuation-corrected data. Further improvements of the method could include the retrieval of an attenuation profile, used to correct the spectrograms within the pipeline itself in a recursive way.

The approach could potentially be extended to include other variables and further alleviate the baseline microphysical assumptions. For instance, the restrictive hypothesis of exponential PSDs, whose limitations were discussed in Sect. 6.4, could be relaxed by considering gamma or modified gamma distributions and retrieving their additional shape parameter(s). A retrieval of the scattering coefficients themselves could also be considered. It should be kept in mind that the addition of new parameters increases the computational cost of the algorithm but also its ill-posedness, and that two-frequency Doppler spectrograms may not be sufficient to resolve it. Given the convincing results obtained recently with triple-frequency data (e.g. the retrieval of snowfall properties from triple-frequency radar moments proposed in Mróz et al. (2021b), or studies of Mróz et al. (2021a) and von Terzi et al. (2022) where triple-frequency spectra are used to study respectively the melting and dendritic growth layers), it is likely that our method would gain in robustness and precision with the inclusion of an additional frequency. Further extensions could include the use of spectral polarimetric variables, which could help retrieve more accurately geometrical properties of hydrometeors.

Adapting the method to study rainfall microphysics from multi-frequency radar Doppler spectra would be feasible with a minimal number of changes in the retrieved variables —for instance, some geometrical descriptors could be simplified (e.g. mass-size power law coefficients), while more care ought to be devoted to the parameterization of the size distribution and the correction of attenuation.

The theoretical pipeline itself is an important contribution of this work, for it can be implemented in other settings and for different types of inverse problems. One fundamental difficulty of such problems is often their ill-posedness: several combinations of physical parameters can yield similar observations. The proposed approach mitigates this by learning information from the spatial structure of the data thanks to convolutional neural networks.

*Code availability.* The code for generating the training set using PAMTRA, and for training the decoder and encoder models, is available at https://github.com/annecbroux/DeepSpectralRetrieval (latest version).

*Data availability.* The data of the ICE GENESIS campaign will be made available on the *Aeris* platform (https://en.aeris-data.fr/). The entire dataset will be made fully public, after an embargo period and the necessary post-processing time, at the beginning of 2023. Individual access can be granted before this time upon contact with the authors.

## Appendix A: Radar calibration and W-band attenuation correction

In this appendix section, we detail the pre-processing that was performed to ensure a proper cross-calibration of the X- and
745 W-band data used, and to correct for the attenuation correction of the W-band measurements.

### A1 X-band calibration drift

A first issue that was encountered was that ROXI's calibration was found to be time variable, for a reason that is not fully clarified yet —possibly related to a hardware artifact causing either the output power or received secondary wave trains to fluctuate; the investigation of this issue is beyond the scope of this work. In addition to these calibration fluctuations, occasional
presence of wet snow on the antenna was found to affect the measurements, despite frequent manual removal. In order to correct for this, we used reflectivity profiles of MXPol (over ROXI) as a reference. Although these are collected at a lower time resolution, they were sufficient to correct for these calibration fluctuations and wet snow antenna attenuation.

### A2 W-band attenuation correction

Once the X-band data is considered reliable enough, we focus on the correction of W-band attenuation by following the method
described in Kneifel et al. (2015):

- Gaseous attenuation: atmospheric profiles are taken from COSMO-1 analyses (Consortium for Small-scale Modeling, 2017), and the corresponding profile of gaseous attenuation is computed using PAMTRA.

- Snowfall attenuation: We use a baseline $Ze_X$-IWC relation ($IWC = 0.015\,Ze_X^0.44$, Kneifel et al. (2015); Boudala et al. (2006)) to estimate the profile of snowfall content, and the corresponding attenuation profile is obtained considering that
ice attenuates around $0.9\,\mathrm{dBkm}^{-1}(\mathrm{gm}^{-3})^{-1})$.

- Supercooled liquid water attenuation: this is the most error-prone step, considering that no measurements of the profile of liquid water content (LWC) are available. We make use of liquid water path (LWP) estimates based on radiometer measurements (Billault-Roux and Berne, 2021), and assume a uniform LWC profile in the cloud/precipitation column. The corresponding attenuation profile is then computed with PAMTRA.

The mean path-integrated attenuation for each of these categories is respectively 0.3, 0.4 and 1.7 dB, on the entire dataset. W-band reflectivity (and Doppler spectrograms) are then corrected using these attenuation profiles. In a final step, as in Dias Neto et al. (2019), the reflectivity values at X- and W-band are cross-corrected by selecting areas close to cloud top and with low reflectivity, and correcting WProf with the mean reflectivity offset in these regions (regions within 1 km of cloud top and $Ze_X < $ -3dB were used; if lower Ze values were used, not enough points would be available). This relies on the assumption
that near cloud top, small ice crystals (with low reflectivity) are Rayleigh scatterers at both frequencies, for which the DFR is 0 dB.

A further note should be added regarding one event in the data set (January 22) which featured rain at ground level during a few hours before a transition to snowfall in the evening. Here, step 3 of the method described could not be conducted because LWP retrieval is dominated by the lower-level rain. The other steps were performed similarly, and only data above the melting
layer was used for the retrieval.

Overall, this allows to mitigate attenuation-related issues but cannot eliminate them. In particular, the presence of super-cooled liquid water is difficult to assess and correct accurately. This should be considered as a limitation of our method at its current stage, and motivated the discussion in Sect. 6.1 about the sensitivity of the retrieval to reflectivity offsets.

**Appendix B: Training set**

**B1   Distributions**

Fig. B1 illustrates the distributions of microphysical parameters in the MASC database (Grazioli et al., 2022) from which the decoder training set is sampled. The particles were grouped into four large categories: aggregates, planar crystals, columnar crystals, and graupel, using the classification output of Praz et al. (2017). To generate the training set, $D_0$, $b_m$ and $\beta_a$ are sampled using skewnorm fits of these distributions; $a_m$ and $\alpha_a$ are sampled using their relation to $b_m$ and $\beta_a$, as illustrated in
Fig. B3 and Fig. B4. The histograms of other variables in the decoder training set are in Fig. B2.

**B2   Self-Similar Rayleigh-Gans Approximation**

The coefficients of the SSRGA made available by Ori et al. (2021) through the SnowScatt model were grouped by types of particles in order to match the four main categories used to sample the training set: planar crystals, columnar crystals, aggregates, graupel. The SSRGA in general and here the simulations of Ori et al. (2021) are mostly targeted on aggregates,
so it was decided to include e.g. columnar aggregates in the columnar category, with the reasoning that when the size regime is that of columnar crystals, the scattering properties would approach those of the individual particles. While this rationale could be debated, it would most likely not trigger diverging results since the SSRGA collapses to Rayleigh scattering for small particles, meaning the exact values of coefficients have little impact. After grouping the particles into the different categories, the coefficients $\kappa$ and $\beta$ are then averaged within each group, for each size bin, as shown with the black lines in Fig. B5.
In addition to the limitations already discussed in this study, mostly focused on the use of only two parameters and on the

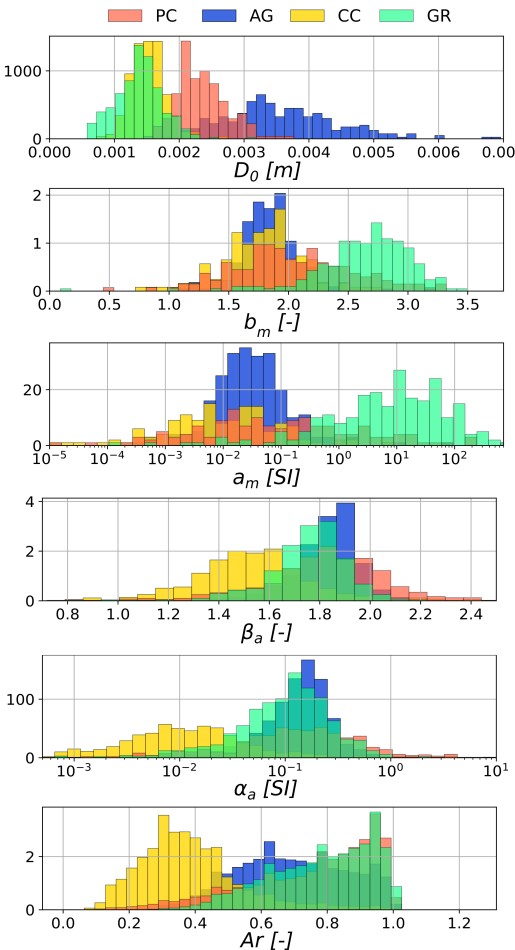

**Figure B1.** Histograms of the microphysical parameters in the MASC database. PC: planar crystals, AG: aggregates, CC: columnar crystals, GR: graupel

reduction of complexity by averaging, an additional point can be noted. It is strictly speaking not valid to assume a single set of scattering coefficients in SSRGA computations for an entire particle size distribution, since the coefficients are size-dependent. However, as underlined in Ori et al. (2021), the coefficients do not change significantly for large particles, while for small particles, it was already noted that the SSRGA simplifies to Rayleigh scattering regardless of the coefficient values, which makes this assumption altogether reasonable.

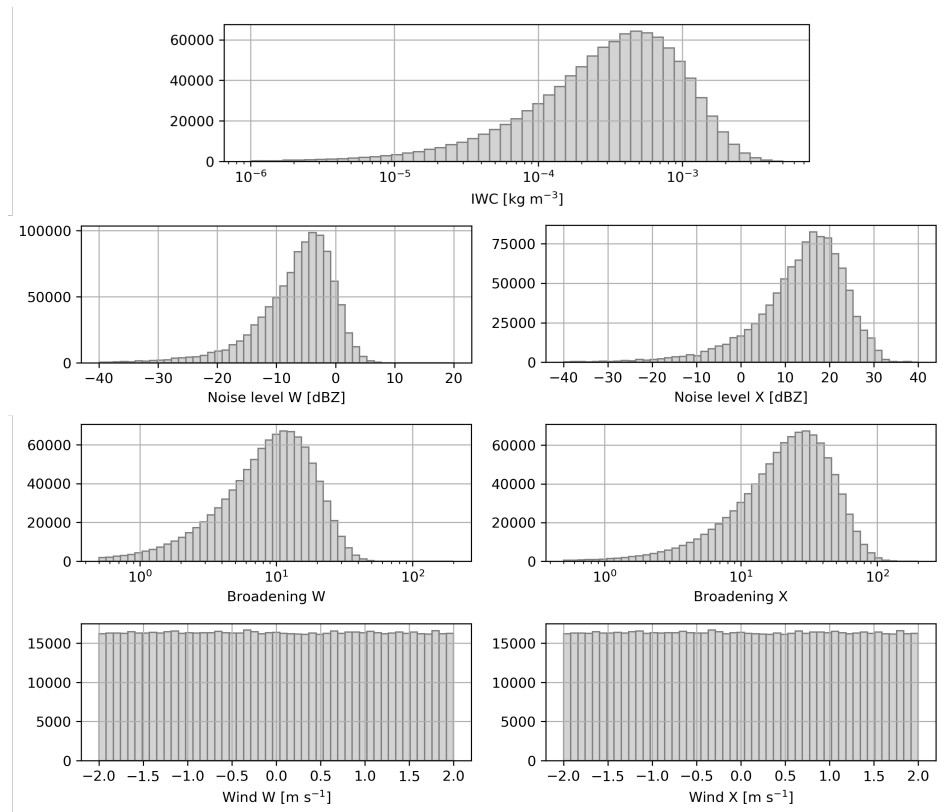

**Figure B2.** Same as Fig. B1 but for the other microphysical descriptors, not provided in MASCDB.

## Appendix C: Alternative deep learning method

One of the motivations for using the architecture proposed in this work is the ill-posedness of the problem, which is arguably an obstacle for direct inversion methods. Nonetheless, such an inversion was also implemented, through a deep learning framework trained on the same synthetic dataset as the one used to train the decoder. This time, the input consists of dual-frequency spectra, and the output is the set of microphysical and atmospheric descriptors (same as Table 1). The architecture, not detailed here, is virtually the same as the encoder (cf. Fig 2 b), except that 2-dimensional convolutions are now 1-D: the neural network is not trained on full spectrograms but on single-gate spectra, thus the range dimension is equal to 1. After training and tuning, the model is applied to the ICE GENESIS dataset. Figure C1 shows the same variables as in the left panels of Fig. 5, retrieved through this *direct* inversion. Overall, the order of magnitude of the variables is similar to that obtained with the new pipeline, and the very general spatio-temporal structure is also visible. This is reassuring since it suggests that the training dataset was appropriate and indeed captured the scope of possibly observed spectra. However, it is also apparent that the retrieved variables are substantially noisier than through our method, reflecting the ill-posedness issue. When comparing these retrieved results

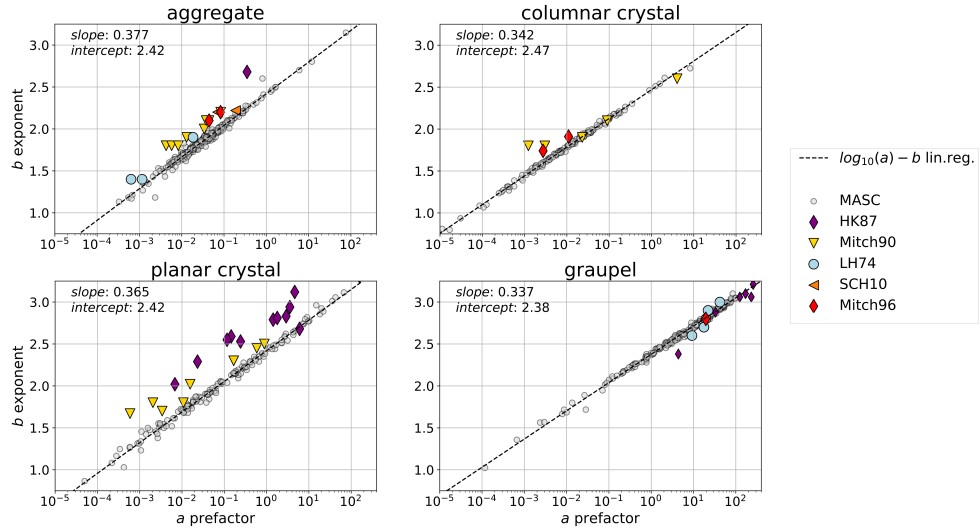

**Figure B3.** Relation between exponent and prefactor of mass-size relations for different particle types, computed using MASCDB.

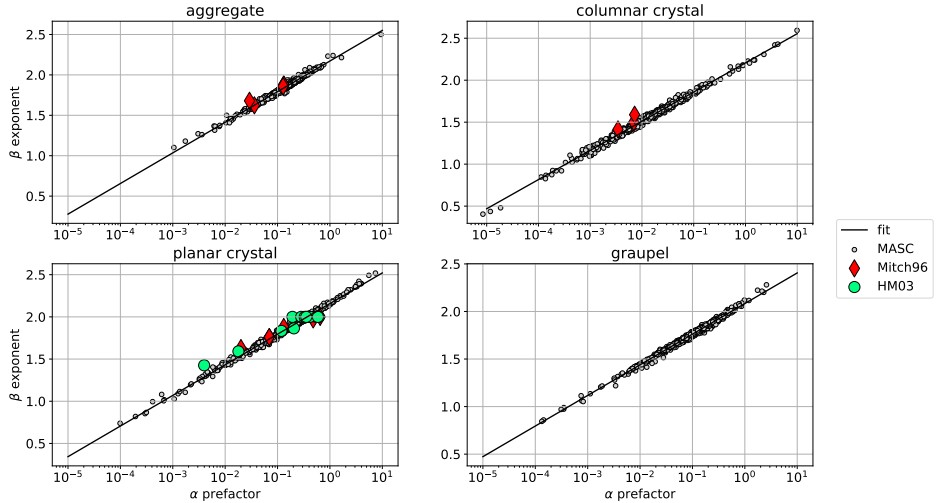

**Figure B4.** Same as B3 but for area-size relation.

with aircraft in-situ measurements, as done in Section 3.3, we obtain for example R = 0.59 for IWC (instead of R = 0.87). Some variables also reach unrealistic values, e.g. aspect ratio close to 0 or even negative values of $D_0$.

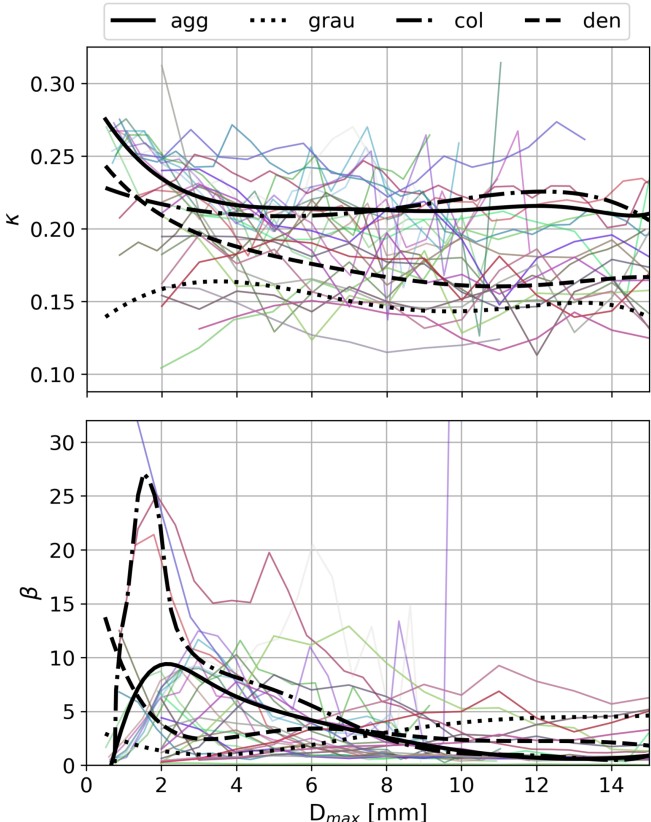

**Figure B5.** Colored lines: coefficient of the SSRGA computed in Ori et al. (2021) for a given particle type (as a function of $D_{max}$) for a) $\kappa$ and b) $\beta$. Black lines: average coefficient when grouping by particle type, used for sampling the training set.

## Appendix D: Overlap metric

We recall the definition of the overlap metric used to evaluate the reconstruction of the spectra: $O(S, \tilde{S}$ is equal to 1 (100%) when the spectra are identical, and to 0 when they are completely disjoint. In this equation, $v_{min}$ and $v_{max}$ are the negative and positive cutoff velocities of the Doppler spectra ($\pm$ 6.9 ms$^{-1}$).

$$O(S, \tilde{S}) = 0.5[\frac{\int_{v_{min}}^{v_{max}} min(S^*, \tilde{S}^*)}{\int_{v_{min}}^{v_{max}} S^*} + \frac{\int_{v_{min}}^{v_{max}} min(S^*, \tilde{S}^*)}{\int_{v_{min}}^{v_{max}} \tilde{S}^*}]$$

Fig. D1) can be helpful to understand this definition (note that the spectra are not real ones, they were drawn for a purely illustrative purpose).

1. Here, $S$ is the reference spectrum (target), and $\tilde{S}$ is the model output (whose quality we want to assess).

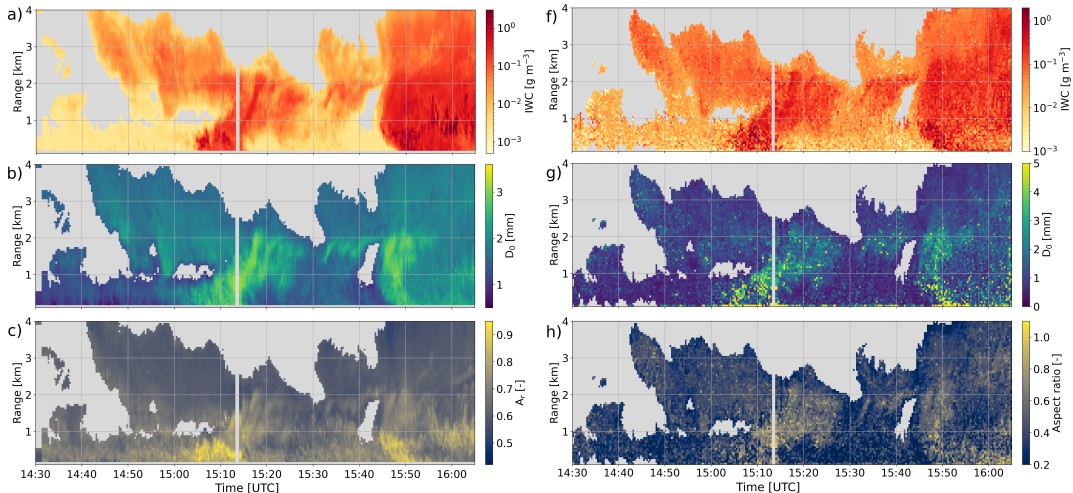

**Figure C1.** Comparison of timeseries for three examples of variables (IWC, $D_0$ and $A_r$) retrieved through the proposed framework (left panels) or a direct deep-learning retrieval (right panels). Note that the colorbars may differ (adjusted to reflect at best the variability in each field).

2. $S$ and $\tilde{S}$ are offset as $S^* = S - min(S)$ and $\tilde{S}^* = \tilde{S} - min(S)$, i.e. we substract the minimum of $S$ to both $S$ and $\tilde{S}$. $S^*$ and $\tilde{S}^*$ are introduced to bring the base level of the target spectrum to 0, otherwise the integrals would be dominated by the noise rather than the signal, as logarithmic values are used. Note that both spectra are offset with the same value ($min(S)$), which allows to identify discrepancies in the absolute reflectivity.

3. The first term of the sum in $O(S, \tilde{S})$ is the hatched area divided by the blue area (see Fig. D1); the second term is the hatched area divided by the pink area. Both terms are needed to account for cases when $\tilde{S}$ would be broader than $S$ (i.e. when $\tilde{S}$ would overlap $S$ completely), **and** when $\tilde{S}$ would be narrower than $S$ (i.e. when $\tilde{S}$ would completely overlapped by $S$).

4. In this example, the value of the overlap metric is 0.65 (65 %).

*Author contributions.* ACBR and AB designed the study, with input from GG in the conception of the retrieval framework. ACBR implemented the deep learning pipeline with contributions from GG. The radar data were prepared by AM, NV and ACBR. Aircraft in-situ measurements were processed by LJ. Comparisons of retrieved to in-situ values, and sensitivity analyses, were conducted by ACBR with input from AB. ACBR prepared the manuscript with contributions from GG and AB and supervision from AB. All authors have read and agreed to the published version of the paper.

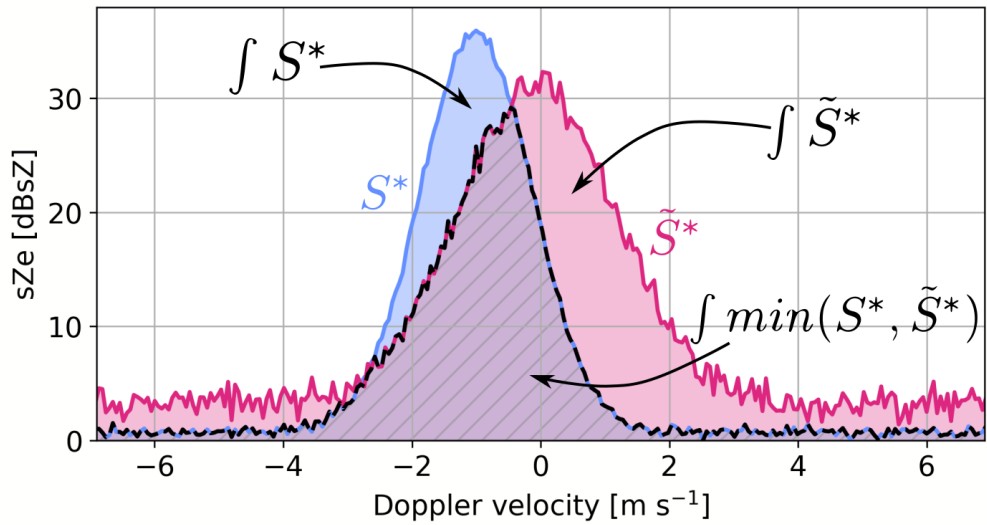

**Figure D1.** Illustration of the overlap metric. The spectra were created for illustration purposes and are not part of the dataset.

*Competing interests.* At least one of the (co-)authors is a member of the editorial board of Atmospheric Measurement Techniques.

*Acknowledgements.* This project has received support from the European Union's Horizon 2020 research and innovation program under
840 grant agreement No 824310 (ICE GENESIS project). Airborne data were obtained using the aircraft managed by Safire, the French facility for airborne research, an infrastructure of the French National Center for Scientific Research (CNRS), Météo-France and the French National Center for Space Studies (CNES). Most of the microphysical in-situ data were collected using instruments from the French Airborne Measurement Platform, a facility partially funded by CNRS/INSU and CNES. We thank Davide Ori for his help in the initial parameterization of PAMTRA, and we are grateful to Julien Delanoë and Susana Jorquera for providing the ice water content retrieved from airborne
RASTA radar measurements. Finally, we thank Stefan Kneifel and one anonymous reviewer for their constructive and helpful comments on the manuscript.

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
