# Peer review of "Dual-frequency spectral radar retrieval of snowfall microphysics: a physics-driven deep learning approach"

_Atmospheric Measurement Techniques, 2022_

## Referee Comment (RC1)

Review of "Dual-frequency spectral radar retrieval of snowfall microphysics: a physically constrained deep learning approach" by Billault-Roux et al., AMT-2022-199

The manuscript presents a unique approach to utilize dual-frequency radar Doppler spectra and a two-step deep learning framework to retrieve a number of snow microphysical parameters. The method is applied to spectra collected during a recent ground-based campaign where the retrieval results could be compared to in-situ data obtained by aircraft overpasses.

Overall I congratulate the authors to this comprehensive approach using machine learning to extract information from multi-frequency Doppler spectra. In my opinion, this will help a lot to better exploit the information about snow microphysics in Doppler spectra. Given the complexity of the method itself and the challenge of selecting appropriate input parameters for the training dataset I find the results very impressive and encouraging. As the authors discuss properly, the method has still some shortcomings as any retrieval probably has. I found the paper very well structured and clearly written which is important in order to help non-expert readers to understand the different steps of the approach. I also appreciate that the code for training the NN is openly accessible. I think this is very important given the fact that the NN has to be trained separately for any other radar combination.

My comments are mostly minor and many of them meant as suggestions to make the description even more clear and complete. I completely understand that certain compromises in the assumptions had to be made in order to keep the computational costs feasible. The references to previous literature could be a bit more extended; I provided some references in my specific comments.

I clearly suggest the manuscript to being published in AMT after my comments and questions have been addressed.

**General comments**:

Forward model assumptions, Sect. 3.1.1: I am a bit surprised to see that you used only inverse exponential PSDs for generating the synthetic Doppler spectra dataset. Several in-situ and remote sensing studies showed that a modified Gamm distribution is a much better function to fit snow PSDs. As shown by Mason et al., the PSD shape (for example mu) impacts also DWRs. My concern is that with your inverse exponential PSD, you are unable to represent for example cases of intense aggregation, where one expects the small particle number to decrease (high mu). Or imagine secondary ice processes, which can lead to super-exponential PSD by enhancing the number of small particles. If it is too much effort to implement a more general PSD in your framework I would strongly suggest to add this aspect to the discussion.

What is general a bit unclear to me is how the retrieval is able to handle attenuation effects (especially at W-band) which can be expected especially when more super-cooled liquid water is in the ice part of the cloud or if even melting layer or rain is present. From the description I find it hard to understand how attenuation was handled. PAMTRA is able to also simulate attenuated spectra, so maybe this information could even be implemented as a variable to be retrieved in the future? (I see a small comment in L. 553 on this topic)

Another aspect is still unclear to me but maybe I misunderstood it: You mention in the beginning that it is extremely challenging to match spectra of two radars due to various effects, for example slight antenna mis-pointing. It is unclear to me how your retrieval deals with those effects. For example, a mis-alignment of the two radars would cause a different velocity shift due to different vertical and

horizontal wind components in the radial velocity. This will cause "fake" sDWR simply due to the different shifting of the spectra. As such effects are probably not included in the training, I wonder what effect those artefacts might have on your retrieval?

General question: Maybe it could be worth to comment in the outlook a bit on how easily your method could be also applied to multi-frequency spectra obtained in rain. There has be quite some work done on that using mostly OE but I wonder how much of the "pre-selection" work that needs to be done for those OE retrievals can maybe be avoided using your approach.

**Minor comments:**

L. 2: I think radar polarimetry should be mentioned here as well. Admittedly, the signals are weak for snowflakes or rimed particles but overall it provides a lot of information about ice processes which are related to the generation of snow. In the future, I guess your method could also be extended to utilize also spectral polarimetry, right?

L. 56-57: I think you should somehow indicate that there are many more studies than Tetoni et al. which explored radar polarimetry for studying ice and snow processes. There is also lots of literature on improving snowfall retrievals by using polarimetric information such as for example Bukovcic et al., 2018 (https://journals.ametsoc.org/view/journals/apme/57/1/jamc-d-17-0090.1.xml).

L. 63-64: For the non-expert reader I suggest to explain more what is the underlying idea of using Doppler spectra: If the particle's terminal fall velocity differs, we can assign the backscattered signal to different populations and hence refine our retrievals. If ice, snow, rimed particles would all fall with the same velocity, the spectra would be actually quite useless.

L. 73-76: Also differential attenuation at the two frequencies is an issue. It shifts the two spectra up and down. This effect can also be used if one knows that certain parts of the spectra must match. Li and Moisseev, JGR, 2019 (https://agupubs.onlinelibrary.wiley.com/doi/full/10.1029/2019JD030316) used this effect to estimate melting layer attenuation. I think it would make sense to mention this study here.

L. 114-116: As the spectra are quite essential for this work I suggest to expand a bit on it in order to enable to reader to better understand the background. I would split the sentence and explain first that the underlying microphysics defines a "perfect" microphysical spectrum which only depends on v-D relation, backscattering properties and PSD. In order to estimate the impact of the dynamics on the spectrum, I already need to know the radar parameters (e.g. beam width). Finally, the radar itself adds noise to the spectrum. In order to reduce the noise effect, one needs to average over several spectra but this leads usually to a smear-out effect of the microphysical information. A study which explores and discusses those different effects w.r.t drizzle signatures is Acquistapace et al., AMT, 2017. Maybe also a reference to a textbook or review article explaining the various influences on Doppler spectra might be good to add.

L. 193: It's not clear to me how you assign aspect ratio if you only have mass, size and cross sectional area. I assume you apply some ellipsoidal fitting? Please specify or provide reference.

L. 197: If I am not wrong, you use Heymsfield+Westbrook to calculate the terminal velocity based on your assumed m-D and A-D relation, right? In the way you write it it sounds like another independent v-D relation is used.

L. 195-197: Sorry if I missed it but I would like to see a table (maybe in the Appendix) which all relevant radar parameters assumed in PAMTRA such as noise level, number of spectral averages,

velocity resolution, beam width, etc. I could imagine different choice of those parameters might affect your results quite a bit. I assume you have adjusted those parameters to the radars you use later for applying the retrieval? I think making the assumptions in Pamtra consistent with the "real" radar settings is quite key because you will get for example quite different spectral broadening if beam widths or number of spectral averages don't match. I would like to see these aspects clarified/discussed more deeply.

L. 198: How do you deal with attenuation? Do you use PAMTRA spectra which include attenuation?

L. 216: Did you use SSRGA also for graupel in Pamtra?

L. 243-244: Why did you not assume a range of Eddy dissipation rates? I think this would represent real conditions much better than a single EDR.

L. 260: Similar to my previous question: Wouldn't you need to do the Pamtra simulations separately for X- and W-Band simply because of the much larger beam width of the X-band? I mean, the broadening due to turbulence must be quite different, or? I think on L. 301 I found the answer. Maybe mention this aspect somewhat earlier.

Table 2: Information about the radar sensitivity/noise level would be good to add.

L. 281: So that means you don't have information about particles larger than 6.4mm? Isn't that a problem for studying snow which can reach sizes of a few cm? I guess missing this large-particle-tail will impact the derived PSD and moments. I think this aspect must be discussed.

L. 340: Could you expand a bit more on this aspect of spatial consistency? First I would like to understand better what the term means. I guess it means that certain spectral features (e.g. a second spectral mode) appear to be connected throughout certain range gates? 3 range gates appear to me quite a small range if I just look at your measured spectrum in Fig. 4a.

L. 388-389, Fig. 3: Couldn't it maybe also be related to insufficient range of assumptions in the training? For example, the range of densities assumed, the simple inverse exponential PSD? Do the two panels in Fig. 3 actually have the same velocity axis? Then the X-band spectrum is from rimed particles, while the W-band is for slight updraft conditions, right? Wouldn't it make more sense to show the spectra for the same time-height of a specific event? Both spectra look quite smooth. I would be interested in seeing the original spectra.

Figure 4 looks indeed very impressive! But for me it is hard to understand how you are able to reconstruct also areas of various level of turbulence so well given that you only use one value of turbulence (EDR) in the Pamtra simulations for training.

Sect. 5.2.2: Correct me if I am wrong, but I guess you could also retrieve eddy dissipation rate, right? I think it could be nice to compare your EDR estimate with simpler methods such as presented by Borque et al., JGR, 2016 (https://agupubs.onlinelibrary.wiley.com/doi/full/10.1002/2015JD024543)

L. 495: Looks like the in-situ confirm that the inverse exponential is not an ideal PSD shape. I would be interested to know if the higher spectral moments (for example skewness) are changing if you use the in-situ PSD instead of your fitted exponential. While certainly the larger particles dominate Ze and DWR, I am not sure that they don't affect the spectral shape.

Sect. 5.3.4: I guess some of the Ar discrepancy could also result from the averaging over particle 2D projections done in common in-situ probes. I guess mentioning the study by Jiang et al., JAMC, 2017 (https://journals.ametsoc.org/view/journals/apme/56/3/jamc-d-16-0248.1.xml) could make sense here.

Fig. 12: Would be great to also have a similar plot showing the impact of absolute and relative velocity offset (I guess it will impact m-D parameters quite a bit).

Sect. 6.3: I like the discussion of the impact of the SSRGA parameters on the D0 estimate. However, I think the fact that you are sticking to inverse exponential PSD might also explain some of the D0 bias. Mason et al., AMT, 2019 (https://doi.org/10.5194/amt-12-4993-2019) demonstrate this effect very nicely and I think this aspect should be included in the discussion.

Figure 9 and D0 discussion: What I find quite puzzling is that your retrieval is never producing D0 values lower than 1mm despite very low DFRs and Ze at some time periods where you have overpasses. Couldn't it maybe be a relative bias in Ze? Can you show reproduced and measured X and W-band spectra plotted over each other once for a low DFR and once for a high DFR region? Ideally there should be no big offset in the y-direction except the different noise levels.

L. 635: Just a comment: Airborne triple-frequency data as used in Mroz et al., AMT, 2021 might not be very useful for your approach as the spectra are probably affected by the aircraft motion. There are ground-based triple-frequency datasets available which provide quite acceptable 3f-spectra (see for example as shown in another article by Mroz et al., AMT, 2021 (https://amt.copernicus.org/articles/14/511/2021/amt-14-511-2021.html) or von Terzi et al., 2022 (https://doi.org/10.5194/acp-2022-263).

**Typos:**

L. 58: "studies, however, rely"

L. 206: "We, however, believe"

Table 1: I think there is a mistake in the description of am/bm and alpha_a/beta_a. It says both "… of the mass-size power law"

L. 348: in this way

L. 360: I guess the "1" with the double line in the equation denotes a vector of ones if the condition in parenthesis is fulfilled?

Caption Fig. 5: Change DFR into DFR_X,W as in the Figure

Figure 5: If possible enlarge font size of the labels on the axis and color bars. Also information on Time should be added on the x-axis (I guess it's time in UTC); also in Fig. 7

L. 486: with, however, the

L. 513: This is not

---

## Referee Comment (RC2)

Review of "Dual-frequency spectral radar retrieval of snowfall microphysics: a physically constrained deep learning approach" by Billault-Roux et al., AMT-2022-199

This manuscript presents a novel technique to invert snowfall microphysics from dual-frequency Doppler spectra by using a deep learning approach. The approach is based on an autoencoder-like framework, which links two neural networks together – one reproducing a well-established forward model, while the other is trying to invert the first one. Besides the implementation of the framework, the manuscript also presents its application to a recent field campaign where ground-based cloud radar measurements are compared with airborne in-situ data to discuss the performance of the inversion.

I really enjoyed reading this manuscript, since it does a great job to introduce and explain its neural network approach instead of leaving the reader with a *dark box* like some other manuscripts. To my knowledge, this study is one of the first to apply the autoencoder concept to the inversion of cloud microphysics. The paper is well structured, easy to follow and engages the curiosity of the reader. Due to this fact, this paper could help to pave the way for further neural network assisted inversions of cloud microphysics. Like RC1, I was impressed by the retrieval results and the comprehensive consideration that went in the necessary assumptions for the forward problem.

My comments are only minor in nature and mainly are concerned with some aspect how these necessary assumptions were introduced and the occasionally overconfidence in the proposed method. Albeit powerful, there are certain limitations in the inversion of cloud radar measurements which even neural networks will not be able to solve. Anyway, these comments only concern the presentation and not the study itself and should therefore be easy to address.

Overall, I am convinced that the presented manuscript is a very valuable contribution to the scientific debate and will advance the application of neural networks in atmospheric science. I clearly suggest the manuscript to being published in AMT after my comments and questions have been addressed.

**General comments**:

1.) Throughout the manuscript you write that your approach can mitigate the ill-posedness of the problem, "*i.e. when several values of x may yield similar outputs y: in such cases, the [direct] retrieval may yield arbitrary outputs" (L152).* However, I am not convinced that your approach is able to do just that. The ill-posedness is determined by the number of measurements and the degrees of freedom and I do not see a way how your approach can overcome this mathematical principle. Where a "direct", e.g., lookup table-based retrieval may yield a set of possible *x* which can explain *y,* your approach only yields one unique *y = f(x)* relationship, although multiple combinations of *x* would still be physically viable. In the first case I obtain a measure of uncertainty while in the latter I am just getting "blind" for other latent variable combinations which also could explain my individual measurement. What you are actually doing during the construction of your decoder (by important sampling of *x)* and your encoder (by using specific snowfall cases) is the limitation of the ill-posed problem by well-chosen priors. That is an appropriate strategy to deal with the ill-posedness of the problem but it is not unique to your technique. Maybe your manuscript could be a little bit more modest around these sentences and acknowledge your use of priors for *x* (*the need ... to assume prior values for x ... has led us to explore a different direction, L143).*
This includes an earlier mentioning of the fact that your approach is directly trained on the data of interest and might thus not be directly suitable as a general retrieval for arbitrary X- and Ka-band spectrograms of ice clouds (first mentioned at L402ff). Moreover, while the introduction to the theory

of inverse problems (Sec. 2.2) is thorough and almost too detailed (L133-L145), I am missing a short literature review about current studies using neutral networks (e.g., Piontek, MDPI, 2021), especially auto-encoders (e.g., Behrens et al, JAMES, 2022) in climate research.

2.) An aspect that is in general a bit unclear to me is your statement in e.g., L94: *"… an important peculiarity of the encoder's architecture is its ability to leverage the spatial consistency of the radar variables, which reduces the ill-posedness of the inversion problem"* which is iterated several times throughout the manuscript. Unless I have missed it you never mention what you mean with "*spatial consistency*" in the first place and only briefly describe its implementation around L340. This was also mentioned by the other referee, and we only can guess that you refer to the observation how spatial features (e.g., updrafts or fall streaks) are connected throughout adjacent range gates. Furthermore, it is not clear to me how this "*spatial consistency*" can overcome the ill-posedness of the inversion problem in each individual gate since we simply have too few measurements compared to the natural variability of realistic ice crystals (see my point 1). In my opinion, your spatial convolution kernels (which you only use for your encoder) lead to a smoother profile of latent variables (which is desired!) but cannot overcome the underdetermination of the problem (e.g., DFR ambiguity between aspect ratio vs. $D_0$).

3.) Like RC1, I am missing a discussion how attenuation by hydrometeors (especially at W-band) could introduce biases in your inversion. In your outlook, you could briefly discuss potential approaches to this problem in the framework of neural networks. Here, it might become necessary to train the decoder on full spectrograms instead of treating each range gate independently to capture the gate-to-gate interdependence cause by attenuation effects. Moreover, you promise (in your abstract) to relax constraints on beam matching with your approach, but only seem to consider a potential doppler shift (hopefully independently between X- and W-band) caused by a beam misalignment. The more fundamental limitation caused by a nonuniform beam filling in the context of different radar beam widths (0.53° vs. 1.8°, 50 m vs. 150 m beam diameter at 5 km altitude) are not handled but should shortly be mentioned.

4.) Throughout the manuscript the used definition on the particle diameter is not entirely clear to me and needs a more precise treatment. You introduce $D_0$ as size parameter of the PSD of Straka (2009) and you call it its "*mean diameter*" in L191. According to Straka (2009), however, $D_0$ is the median volume diameter of the exponential size distribution if $D = D_{max}$. Furthermore, your fixed relation $D_{eff} = 3D_0$ seems wrong to me for ice particles regardless your definition of $D_0$. The *effective diameter* regarding solar radiation depends rather on the area size relationship. As pointed out by RC1, your strategy how you relate the aspect ratio with particle mass and size is not clear to me – it appears to be based on the common soft spheroid approximation. Regarding this approach and the discrepancies found between your retrieval and in situ measurements you should consider the remarks made e.g., by Hogan et al (2012). They show how particle shape and orientation influence the mean diameter retrieved by the DWR technique compared to in situ measurements. Likewise, I am missing a more explicit description how the diameter, the area and the terminal velocity are connected. While I am convinced that the authors are aware of these points and have put sufficient care into their implementation, their manuscript lacks the necessary diligence regarding the particle diameter.

**Specific comments:**

L37: What do you mean when you write: "*the retrieval of snow microphysics from radar variables is not explicit*"?

L47: "[DFR] *can thus be used to identify populations of snow particles with a larger size or density.*" How does density influence the DFR sensitivity? I always thought that the DFR technique is inherently insensitive to density of ice particles?

L76: "*especially when turbulent broadening is observed […], a direct computation of the dual-frequency spectral ratio is meaningless.*" This statement is too harsh in my opinion. While turbulent broadening can severely hamper the exploitation of the dual-frequency spectral ratio, you should give an estimation at which magnitude its value becomes "*meaningless*".

Fig. 1: A little bit more descriptive caption would help the reader here.

L232: On what observations or studies are these numbers based? Could the choice to give aggregates more weight during the training not also bias your retrieval towards the property of aggregates? The (40/20/20/20%) distribution is thus part of the implicit prior of your approach, correct?

Sec 3.2: Although your accompanying paper Billault-Roux et al (2022) might provide these numbers, a reader might be interested in the size (hours or number of profiles) of your training dataset.

L314: I really appreciated your clear and comprehensible description of your neural network approach. I would be delighted if you could spend 1-2 more sentences on the role of these *residual blocks.* Do we know why or how they facilitate the training process?

L318ff: You mentioned the separation of your measurements into an 80% training, 10% validation and 10% testing data set. In the following, you no longer refer to how you used the 10% validation and 10% testing data set or did I miss something?

Tab 3. Could you elaborate where the choice of 30 for the number of channels comes from? I thought that the channel dimension relates to the radar bands used (X- and Ka-band).

L377-L381: The description of the loss function *O(S, ~S)* is quite complicated and hard to understand. Explain the problem with the difference between normalized spectra and why your loss function consists of two integrals. How can you spot discrepancies in the absolute reflectivity at all when you normalize the simulated and measured spectra with each other?

Fig 4.: While impressive, I would prefer to see the profile of some latent variables like *IWC* and $D_0$ in panel b) instead of showing the same spectrogram three times. The argument that PAMTRA can be imitated reasonably well (panel c) is already demonstrated in Fig. 3.

Sec 5.3.2 Size parameter: Several times you mention "*cofluctuation*" (e.g., L500) between ground-based retrieved and airborne measured properties. Have you averaged the airborne dataset after the selection of nearby overpasses? Otherwise, I would not expect a good correlation between a ground-based and airborne platform. This could also explain the much higher variability of the RASTA retrieval.

Sec 5.3.4 Aspect ratio: While reading this section I noticed that you never mentioned the average particle orientation in your PAMTRA simulations. As this is obviously fixed, this could be a further origin for the observed bias. Furthermore, I noticed here that you limited your AR to oblate particles (see Fig. B5, last panel). In the presence of rice shaped particles, e.g., needles, bigger biases in your size and IWC retrieval may be expected and could explain the observed discrepancies. Please include this in your discussion.

**Typos and wording:**

Throughout the text you are using the saxon genitive with objects, some examples:
       L11: "*the problem's ill-posedness*" > "*the ill-posedness of the problem*"
       L13: "*the retrieval's accuracy*" … "*the accuracy of the retrieval*" or "*the retrieval accuracy*"
       L104: "*method's sensitivity*" … "*the sensitivity of the method*"
       L196: "*the radar's properties*" … "*the radar properties*"

L52: "*comforted through*" … "*confirmed through*"

L67: "*The scattering regime transition in high frequencies is in principle visible*" … "*The transition of the scattering regime at higher frequencies is visible*"

L124: "*we use as a forward model the radiative transfer code PAMTRA*" … "*we use the radiative transfer code PAMTRA as a forward model*"

L207: "*leaving to future studies the possible improvements of the forward model*" … "*leaving possible improvements of the forward model to future studies*"

Tab 1: $\alpha_a$ and $\beta_a$ should probably be the pre-factor and exponent of the area-size relationship?

L246: This sentence is awkward and hard to follow, please rephrase.

L458: "*Sect. 3.2*" is now "*Appendix A*"

---

## Author Comment (AC1)

**Dual-frequency spectral radar retrieval of snowfall microphysics: a physics-driven deep learning approach**

*amt-2022-199*

**Responses to reviewers**

A.-C. Billault-Roux, G. Ghiggi, L. Jaffeux, A. Martini, N. Viltard and A. Berne

December 20, 2022

We would like to thank S. Kneifel and one anonymous reviewer for their constructive comments, their positive feedback and useful suggestions. These helped improve the manuscript by clarifying certain aspects of the retrieval framework, and discussing more precisely and thoroughly some of its limitations.

Each section corresponds to the comments of a referee. The comments of the reviewers are reported in italic, our responses in normal font and the corresponding modifications in the manuscript in blue.

In addition to the changes made to address the comments of the reviewers, and a few minor modifications of the text, we decided to slightly modify the title of the manuscript:

Dual-frequency spectral radar retrieval of snowfall microphysics: a  **physics-driven** deep learning approach

We indeed think that the phrase "physically-constrained" may be misleading as it suggests that physical constraints are explicitly imposed in the inversion approach. In our case, physical knowledge is incorporated through the decoder network which is trained to mimic a radiative transfer model. The wording "physics-driven" seems more accurate to describe this effect.

Additionally, the figures showing retrieval results (Figs. 5 to 12) were updated after a new correction of attenuation at W-band was implemented (as detailed in the responses to the reviewer comments). Note that this did not lead to significant changes in the interpretation of these figures.

**1 RC #1 — Stefan Kneifel**

*The manuscript presents a unique approach to utilize dual-frequency radar Doppler spectra and a two-step deep learning framework to retrieve a number of snow microphysical parameters. The method is applied to spectra collected during a recent ground-based campaign where the retrieval results could be compared to in-situ data obtained by aircraft overpasses. Overall I congratulate the authors to this comprehensive approach using machine learning to extract information from multi-frequency Doppler spectra. In my opinion, this will help a lot to better exploit the information about snow microphysics in Doppler spectra. Given the complexity of the method itself and the challenge of selecting appropriate input parameters for the training dataset I find the results very impressive and encouraging. As the authors discuss properly, the method has still some shortcomings as any retrieval probably has. I found the paper very well structured and clearly written which is important in order to help non-expert readers to understand the different steps of the approach. I also appreciate that the code for training the NN is openly accessible. I think this is very important given the fact that the NN has to be trained separately for any other radar combination. My comments are mostly minor and many of them are meant as suggestions to make the description even more clear and more complete. I completely understand that certain compromises in the assumptions had to be made in order to keep the computational costs feasible. The references to previous literature could be a bit more extended; I provided some references in my specific comments.*

*I clearly suggest the manuscript to being published in AMT after my comments and questions have been addressed.*

We thank Stefan Kneifel for his appreciation of the manuscript and positive feedback on the retrieval method that we propose. In the following we address his comments point by point. To fully take into account his suggestions, in addition to inline modifications of the text, a few small structural changes were made. Specifically, we introduced a subsection in the discussion section addressing the assumptions on PSD shape, and we extended Appendix A with a more complete description of the calibration and attenuation corrections.
* * *
**General comments**

1. *Forward model assumptions, Sect. 3.1.1: I am a bit surprised to see that you used only inverse exponential PSDs for generating the synthetic Doppler spectra dataset. Several in-situ and remote sensing studies showed that a modified Gamma distribution is a much better function to fit snow PSDs. As shown by Mason et al., the PSD shape (for example mu) impacts also DWRs. My concern is that with your inverse exponential PSD, you are unable to represent for example cases of intense aggregation, where one expects the small particle number to decrease (high mu). Or imagine secondary ice processes, which can lead to super-exponential PSD by enhancing the number of small particles. If it is too much effort to implement a more general PSD in your framework I would strongly suggest to add this aspect to the discussion.*

We fully agree with the reviewer that the use of an inverse exponential PSD is one of the strong assumptions used in our framework. Studies such as Mason et al. [2019] indeed show that when considering multi-frequency measurements, the shape of the PSD should be taken into account for more accurate retrievals; furthermore, the Doppler spectrum results almost directly from the PSD and would thus very likely be affected by modifications of its shape (unless significant broadening is caused by atmospheric dynamic conditions). The choice of imposing this PSD

shape was made to avoid having a too large number of retrieved parameters, considering the underlying issue of ill-posedness (which our method partly mitigates — cf response to Specific comment #15 — but does not resolve): several combinations of microphysical descriptors can yield the same radar signature (in mathematical terms, the mapping f is not injective). Adding yet another degree of freedom could thus have deteriorated other aspects of the retrieval due to this increased ill-posedness. It would also have required the creation of a larger training dataset and consequently made the training of the decoder more challenging and computationally expensive. Additionally, while recent studies indeed show the limitations of the exponential PSDs in radar-based retrieval, this distribution still remains quite widely used in parameterizations of ice phase microphysics (for instance with the Morrison scheme [Morrison et al., 2005], implemented in WRF in e.g. Georgakaki et al. [2022]); more generally, models often rely on two-moment schemes which do not allow flexibility on the PSD shape. In view of the above considerations, we decided to approach the inversion problem with an exponential PSD, which seemed a natural baseline assumption to start with.

However, we would like to highlight that exploring the direction of PSD shape is undoubtedly a possibility for future improvements of the inversion model. For example, the shape parameter of a gamma distribution could be retrieved as an additional microphysical descriptor. In order to relieve the ill-posedness that this would cause, the addition of a third frequency —unfortunately not available with a high enough sensitivity in the ICE GENESIS dataset— would likely be very valuable.

Following this direction, we believe that potential extensions also include considering multi-modal distributions. To facilitate training, this may require a pre-selection of multi-modal spectra on which a different encoder-decoder system would be applied (here, the reconstructed bi-modal spectra would be the sum of the outputs of the decoder applied to two sets of microphysical properties).

Overall, to address this concern and state clearly the limitations of the exponential PSD, 1/ we modified the text in Sect. 3.1, 2/ we added a subsection to discuss how the shape of the PSD may affect the Doppler spectra and in turn impact the retrieved values (now subsection 6.4, see more detail in the responses to comments #19 and #22) and 3/ we extended the conclusion to clarify the possible future improvements that could be led by including more diverse PSD shapes in the training dataset.

[Sect. 3.1] The choice of an exponential shape for the PSD was made to constrain the degrees of freedom of the retrieval and keep the computational expense tractable; it is nonetheless a strong underlying hypothesis of the framework in its current version (discussed in Sect. 6.4).

[new Subsection 6.4] **Shape of the particle size distribution** (the additions to this section are detailed in the response to specific comment # 22.)

[Conclusion] The approach could potentially be extended to include other variables and further alleviate the baseline microphysical assumptions. For instance, the restrictive hypothesis of exponential PSDs, the limitations of which were discussed in Sect. 6.4, could be relaxed by considering gamma or modified gamma distributions and retrieving their additional parameters; a retrieval of the scattering coefficients themselves could also be considered. It should be kept in mind that the addition of new parameters increases the computational cost of the algorithm but also its ill-posedness, and that two-frequency Doppler spectrograms may not be sufficient to resolve it.

2. *What is general a bit unclear to me is how the retrieval is able to handle attenuation effects (especially at W-band) which can be expected especially when more super-cooled liquid water is in the ice part of the cloud or if even melting layer or rain is present. From the description I find it hard to understand how attenuation was handled. PAMTRA is able to also simulate attenuated spectra, so maybe this information could even be implemented as a variable to be retrieved in the future? (I see a small comment in L. 553 on this topic)*

The question of attenuation, especially at W-band, is indeed a crucial one for this retrieval (as for many). In its current state, the algorithm is unfortunately **not able** to handle attenuated spectrograms. In the response to this comment we 1/ provide some insights on how the framework could be adjusted to allow for retrieval of attenuation and 2/ provide more detail on the attenuation correction that we used (and how this aspect is discussed in the revised version).

**Retrieving attenuation would require a slightly more involved set-up** for the training of the algorithm. Currently, the *decoder* part of the framework operates in a "gate-to-gate" manner, as it is trained on individual spectra and not on profiles. We believe this is a key point: training the decoder on full profiles would require many additional assumptions and the creation of a huge training dataset to account for the diversity of possible atmospheric / cloud profiles, and thus does not appear easily feasible. Nonetheless, because of the cumulative nature of attenuation along the path, the full profile should somewhat be taken into account.

We believe that the framework could be adapted for this for example by 1/ adding an intermediate module computing the resulting attenuation from the retrieved microphysical properties and correcting the input and reconstructed spectrograms by adding to them the cumulated attenuation profile obtained, and 2/ by recursively applying the inversion model till convergence.

This method, illustrated in Fig. 9 below, would allow to correct for attenuation by ice hydrometeors. Still, a major source of attenuation at W-band (during snowfall) is the presence of supercooled liquid water, which is often not easily identified. Accounting for it would require identifying the presence of a secondary mode corresponding to liquid water (and this mode might not always be visible, especially in turbulent cases) and retrieving the LWC from this signature: implementing these steps in our retrieval requires some non-trivial modifications.

Another way to go would be to directly retrieve from the spectrograms an attenuation profile (which would account for gaseous, snowfall and liquid water attenuation) in parallel to the other microphysical properties and then correct the reconstructed spectrograms by adding to them the cumulated attenuation obtained, as illustrated below in Fig. 2. In this set-up, the attenuation is handled by the encoding part of the framework. Without other constraints on the attenuation, the risk is that the ill-posedness would become too strong. One could think of constraining it for instance by using additional measurements of brightness temperature (which could also be simulated with PAMTRA).

Overall, the implementation of an additional attenuation retrieval was not implemented in the first version of the algorithm because it raises shome challenges in the training of the algorithm as well as some non-trivial adjustments of the framework, although we believe some improvements/developments could surely be made along this direction.

Thus, in this manuscript we work with data that were **a-priori corrected for attenuation using more classical methods** (at W-band), although these are inevitably also error-prone. In the revised version of the manuscript, we include a more thorough correction of ROXI's calibration drift (using MXPol as a reference) and of W-band attenuation, following the approach described in Kneifel et al. [2015], using LWP values to correct for supercooled liquid water attenuation and X-band reflectivity to account for snowfall attenuation. This is now included in Appendix A, which was extended compared to the original manuscript. Note that these modifications only result in minimal changes of the results (barely visible in Fig. 8, and minor

change of the RMSE / bias values).

The challenges of incorporating attenuation as a retrieved variable in this first stage of development are what led us to discuss the effect of reflectivity biases on the retrieval outputs (in Sect. 6.1). We modified slightly this section, including its title, to highlight the importance of possible remaining attenuation effects, and how they motivate this discussion.

[Figure]

Figure 1: First example of how the framework could be adjusted to retrieve attenuation. The decoder is adapted to learn not only the Doppler spectra but also the corresponding attenuation (we include both X- and W-band in the diagram, but possibly only W-band could be considered).

[Section 3.2] As pre-processing steps, the radars are cross-calibrated and an attenuation correction is implemented at W-band (similar to Kneifel et al. [2015]), as detailed in Appendix A.

See modifications to Appendix A [Radar calibration and W-band attenuation correction]

[Modification of Sect. 6.1] Sensitivity to miscalibration and differential attenuation:
As detailed in Appendix A, the issue of attenuation was tackled by implementing a correction of W-band reflectivity based on estimates of gaseous, snowfall and liquid water attenuation profiles. This correction method is however error-prone and we cannot exclude the presence of remaining reflectivity biases in the dataset. We highlight that the presence of supercooled liquid water cloud layers or wet snow can be particulary difficult to identify and diagnose, although strongly attenuating millimeter-wavelength signal (with e.g., path-integrated attenuation up to 5 dB for liquid water paths of 500 g m$^{-2}$, Kneifel et al. [2015]). To assess the possible importance of inaccurate calibration or attenuation correction on the retrieval, we investigated its sensitivity to reflectivity offsets, both absolute and relative.

[Conclusion] One drawback of the algorithm in its current state is that it relies on attenuation-corrected data. Further improvements of the method would include the retrieval of an atten-

[Figure]

Figure 2: Second example of how the framework could be adjusted to retrieve attenuation. In this case, the encoder retrieves a cumulative attenuation profile. A constraint on total attenuation could be added to the loss term.

uation profile, computed from the retrieved microphysical properties, and used to correct the spectrograms within the pipeline itself.
* * *
3. *Another aspect is still unclear to me but maybe I misunderstood it: You mention in the beginning that it is extremely challenging to match spectra of two radars due to various effects, for example slight antenna mis-pointing. It is unclear to me how your retrieval deals with those effects. For example, a mis-alignment of the two radars would cause a different velocity shift due to different vertical and horizontal wind components in the radial velocity. This will cause "fake" sDWR simply due to the different shifting of the spectra. As such effects are probably not included in the training, I wonder what effect those artefacts might have on your retrieval?*

We thank the reviewer for raising this point, on which we realize we did not comment enough in the first version of the manuscript. This possible misalignment of the two radar beams, resulting in radial velocity offsets, is indeed a key aspect and was one of the original motivations for this framework.

The algorithm handles this by retrieving two different values of radial wind and turbulent broadening, one for W-band and one for X-band. It thus allows for possibly different radial wind shifts in the spectrograms of each radar. Similarly, spectral broadening may be larger in case of antenna mispointing, hence the independent values retrieved at X- and W-band.

[Section 3.1.3] "Then, turbulent broadening and spectrum shift due to radial wind are added, with randomly sampled values, different for X- and for W-band. Including these variables will allow the retrieval to handle possible velocity offsets in the X- and W-band spectra caused by beam misalignment [...]"

4. *General question: Maybe it could be worth commenting in the outlook a bit on how easily your method could be also applied to multi-frequency spectra obtained in rain. There has been quite some work done on that using mostly OE but I wonder how much of the "pre-selection" work that needs to be done for those OE retrievals can maybe be avoided using your approach.*

We thank the reviewer for this suggestion. Indeed, the method could be adapted to study rainfall with multi-frequency spectra. Some adjustments would likely be necessary: for instance, the implementation of an attenuation retrieval in addition to the microphysical properties would probably become necessary. The microphysical descriptors would need to be adapted as well: especially, assuming an exponential distribution would probably be even more problematic than in snowfall and thus additional parameters of the distribution should be incorporated into the retrieval. On the other hand, some descriptors could probably be simplified (e.g. the mass-size and area-size relations).

Adapting the method for the retrieval of rainfall microphysics from multi-frequency radar Doppler spectra would be feasible with a minimal number of changes in the retrieved variables — for instance, some geometrical descriptors could be simplified (e.g. mass-size power law coefficients), while more care ought to be devoted to the parameterization of the size distribution and the correction of attenuation.

**Minor comments**

1. *L. 2: I think radar polarimetry should be mentioned here as well. Admittedly, the signals are weak for snowflakes or rimed particles but overall it provides a lot of information about ice processes that are related to the generation of snow. In the future, I guess your method could also be extended to utilize also spectral polarimetry, right?*

We agree that radar polarimetry should not be omitted in this introductory sentence. Indeed, our method could also include spectral polarimetry, although this would require a more advanced forward model, since to our knowledge SSRGA does not allow the simulation of polarimetric variables. We rephrased as follows:

The use of meteorological radars to study snowfall microphysical properties and processes is well established, in particular via a few distinct techniques: the use of radar polarimetry, of multi-frequency radar measurements and of radar Doppler spectra. We propose a novel approach to retrieve snowfall properties by combining the latter two techniques, while relaxing some assumptions on e.g. beam alignment and non-turbulent atmosphere.

We also included in the conclusion the mention of spectral polarimetry as a possible extension:

Further extensions could include the use of spectral polarimetric variables, which could help retrieve more accurately geometrical properties of hydrometeors.

2. *L. 56-57: I think you should somehow indicate that there are many more studies than Tetoni et al. which explored radar polarimetry for studying ice and snow processes. There is also lots of literature on improving snowfall retrievals by using polarimetric information such as for example Bukovcic et al., 2018 (https://journals.ametsoc.org/view/journals/apme/57/1/jamc-d-17-0090.1.xml).*

We rephrased this sentence to highlight the originality of Tetoni et al. [2021] where radar polarimetry and multi-frequency are combined, and we included another sentence to highlight polarimetry-based studies on snowfall properties and processes.

In the case of scanning radars, additional polarimetric information can be included, which opens up possibilities for the geometrical description of ice-phase hydrometeors (e.g., Bukovčić et al. [2018], Matrosov et al. [2020], or Tetoni et al. [2021], Oue et al. [2021], where polarimetric and multi-frequency measurements are combined).
* * *
3. *L. 63-64: For the non-expert reader I suggest to explain more what is the underlying idea of using Doppler spectra: if the particle's terminal fall velocity differs, we can assign the backscattered signal to different populations and hence refine our retrievals. If ice, snow, rimed particles would all fall with the same velocity, the spectra would be actually quite useless.*

We expanded this part to give the reader a better intuition of what Doppler spectra bring. We prefer not to go into further detail on the subject of multi-modal spectra, since our retrieval does not handle these in its current state, and we would like to avoid any misleading statement.

On the other hand, more qualitative studies have been conducted relying not solely on radar moments (e.g. reflectivity $Z_e$ or mean Doppler velocity $MDV$) but rather on the full Doppler spectrum, which allows separating the contribution of slow-falling —typically small— vs. fast-falling —typically large or dense— particles to the total reflectivity. Indeed, the full Doppler spectrum encloses more information on microphysical properties and the particle size distribution (PSD) than scalar moments like $Z_e$ or $MDV$. By observing wider, more skewed, or even multi-modal spectra, signatures of specific microphysical processes can be identified such as riming or aggregation (e.g. Shupe et al. 2004, Kalesse et al. 2016).
* * *
4. *L. 73-76: Also differential attenuation at the two frequencies is an issue. It shifts the two spectra up and down. This effect can also be used if one knows that certain parts of the spectra must match. Li and Moisseev, JGR, 2019 (https://agupubs.onlinelibrary.wiley.com/doi/full/10.1029/2019JD030316) used this effect to estimate melting layer attenuation. I think it would make sense to mention this study here.*

We agree with the reviewer that the question of attenuation should be mentioned at this point. We included the following sentence:

Differential attenuation of the two frequencies is another significant challenge, for which some workarounds were proposed (e.g., Li et al. 2019), but are not always possible to implement.

5. *L. 114-116: As the spectra are quite essential for this work I suggest to expand a bit on it in order to enable to reader to better understand the background. I would split the sentence and explain first that the underlying microphysics defines a "perfect" microphysical spectrum which only depends on v-D relation, backscattering properties and PSD. In order to estimate the impact of the dynamics on the spectrum, I already need to know the radar parameters (e.g. beam width). Finally, the radar itself adds noise to the spectrum. In order to reduce the noise effect, one needs to average over several spectra but this leads usually to a smear-out effect of the microphysical information. A study which explores and discusses those different effects w.r.t drizzle signatures is Acquistapace et al., AMT, 2017. Maybe also a reference to a textbook or review article explaining the various influences on Doppler spectra might be good to add.*

We followed the reviewer's suggestion and included a more detailed explanation of how the different variables (microphysics, atmospheric, instrument) contribute to the Doppler spectrum. We also included additional references, including but not limited to, those suggested by the reviewer.

In the case of a vertically-pointing profiler, the shape of a Doppler spectrum in snowfall results from a combination of several factors (e.g. Doviak and Zrnic 1993, Kollias 2002, Luke and Kollias 2013, Kneifel et al. 2016). It is primarily defined by the snowfall PSD and the microphysical properties of the snow particles (e.g. bulk density, geometry, etc) which determine their backscattering cross-section and terminal velocity. In reality, this purely microphysical spectrum is affected by atmospheric dynamic conditions —turbulence, horizontal and vertical wind— in a way that depends on the settings and parameters of the radar itself —sensitivity, beam width. The actual measured spectrum is additionally perturbed by instrument noise, the effect of which is mitigated through temporal averaging of the spectra, at the risk of smearing out underlying microphysical signatures [e.g. Acquistapace et al., 2017].

6. *L. 193: It's not clear to me how you assign aspect ratio if you only have mass, size and cross sectional area. I assume you apply some ellipsoidal fitting? Please specify or provide references.*

We initially included the definition of the aspect ratio slightly further in the text, but we realize this may create confusion. We thus added the following sentence to state the definition of $A_r$ as used in the SSRGA: it differs from some usual definitions of aspect ratio, as it is simply used in the SSRGA equations as a scaling factor (effective aspect ratio). Note that aspect ratio is specified *independently* from the other parameters. While this may lead to an overly large dataset (i.e. containing non-physical combinations of parameters), we preferred this rather than biasing the dataset with pre-defined relations between parameters. See also the discussion at the end of Sect. 5.1.1 (... some expected consistent behaviors are observed in the retrieval like the apparent correlation between $b_m$ and $A_r$ [...] those variables are prescribed independently when generating the training set).

[Sect. 3.1.1] Aspect ratio $A_r$ is then specified, defined here as equal to the particle's dimension along the direction of the radar beam (here, vertical) divided by maximum dimension [Ori 2021].

7. *L. 197: If I am not wrong, you use Heymsfield+Westbrook to calculate the terminal velocity based on your assumed m-D and A-D relation, right? In the way you wrote it sounds like another independent v-D relation is used.*

Indeed, the Heymsfield&Westbrook relation used for terminal velocity uses the mass-size and area-size power laws defined before. We clarified this:

The velocity-size relation is the one proposed by Heymsfield and Westbrook [2010], and relies on the aforementioned mass-size and area-size relations.

8. *L. 195-197: Sorry if I missed it but I would like to see a table (maybe in the Appendix) which all relevant radar parameters assumed in PAMTRA such as noise level, number of spectral averages, velocity resolution, beam width, etc. I could imagine that different choices of those parameters might affect your results quite a bit. I assume you have adjusted those parameters to the radars you use later for applying the retrieval? I think making the assumptions in PAMTRA consistent with the "real" radar settings is quite key because you will get for example quite different spectral broadening if beam widths or number of spectral averages don't match. I would like to see these aspects clarified/discussed more deeply.*

We agree with the reviewer that this is an important aspect. Indeed, we used the settings of the actual X- and W-band radars also for the PAMTRA simulations. We added a few lines to Table 2, and we now refer to it in the section on the training dataset.

Individual spectra are simulated through PAMTRA for an altitude of 1000 masl, using a standard (PAMTRA default) atmospheric profile with a temperature randomly chosen in [-20 °C, 1°C]. Spectra are simulated at X- and W-band independently. The radar settings for these simulations (frequency, beamwidth, velocity resolution, velocity range, sensitivity) should have the same values as those of the radars on which the retrieval is implemented (c.f. Sect. 3.2 and Table 2).

9. *L. 198: How do you deal with attenuation? Do you use PAMTRA spectra which include attenuation?*

As detailed in the response to the reviewer's General Comment #1, our algorithm currently does not handle attenuation (it requires prior correction of attenuation on the data). We clarify it here as well:

In the current version of the algorithm, attenuation is not taken into account in the PAMTRA simulations.

10. *L. 216: Did you use SSRGA also for graupel in Pamtra?*

Yes, we used SSRGA for all types of particles: the choice of a unique scattering model was made in order to avoid possible "discontinuous" effects if different models were used. We are aware that SSRGA is not designed for the modeling of high-density sphere-like particles like graupel, and such cases might thus be misrepresented. However, Ori et al. [2021] found that SSRGA provides valid results for particles with a relatively high degree of riming, which suggests that this hypothesis is reasonable in a wide range of cases.

It should also be kept in mind that the SSRGA would fail to represent large graupel particles, although Ori et al. 2021 suggest that its validity extends to particles with a relatively high riming degree. [...] We believe that it was reasonable to use the simplest possible parameterization for the initial development of the method —and in particular, use a common SSRGA framework for all scattering calculations—, leaving to future studies possible improvements of the forward model.
* * *
11. *L. 243-244: Why did you not assume a range of Eddy dissipation rates? I think this would represent real conditions much better than a single EDR.*

We used a large range of eddy dissipation rates to generate the training dataset: for the simulation of each spectrum, a value of EDR is randomly sampled from an exponential distribution of mean $10^{-3}\, m^2 s^{-3}$.

It is computed by randomly sampling a value of atmospheric turbulence, represented by the eddy dissipation rate (sampled from a negative exponential with $10^{-3} m^2 s^{-3}$ mean, consistent with some literature standards, e.g. Sharman et al., 2014).
* * *
12. *L. 260: Similar to my previous question: Wouldn't you need to do the PAMTRA simulations separately for X- and W-Band simply because of the much larger beam width of the X-band? I mean, the broadening due to turbulence must be quite different, or? I think on L. 301 I found the answer. Maybe mention this aspect somewhat earlier.*

We include this information in two new sentences:

[Sect. 3.1.1] Spectra are simulated at X- and W-band independently. The radar settings for these simulations (frequency, beamwidth, velocity resolution, velocity range, sensitivity) should have the same values as those of the radars on which the retrieval is implemented (cf. Sect. 3.2 and Table 2).

[Sect 3.1.3] Then, turbulent broadening and spectrum shift due to radial wind are added, with randomly sampled values, different for X- and for W-band. Including these variables will allow the retrieval to handle possible velocity offsets in the X- and W-band spectra caused for example by beam mis-alignment, or differences in spectral broadening due to the different beam widths of the radars.

13. *Table 2: Information about the radar sensitivity/noise level would be good to add.*

We thank the reviewer for this suggestion and included this information in the table as a new line.

| Radar properties | ROXI | WProf | | |
| --- | --- | --- | --- | --- |
| | | *chirp 0* | *chirp 1* | *chirp 2* |
| Sensitivity (dBZ) [at range (km)] | -19 [2] | -45 [0.5] | -41 [2] | -39 [5] |

14. *L. 281: So that means you don't have information about particles larger than 6.4mm? Isn't that a problem for studying snow which can reach sizes of a few cm? I guess missing this large-particle-tail will impact the derived PSD and moments. I think this aspect must be discussed.*

This is indeed an important point. Because of this significant size limitation, we decided not to compare our retrieval of size parameter $D_0$ to the mean size of the distribution (or to a moment derived from it), but rather to the $D_0$ from an exponential fit of the PSD. Small particles are excluded because of the non-exponential behavior typically observed in this range, which can be due to (for example) the presence of new ice crystals or of liquid droplets. Note that we monitor the quality of this "exponential tail" assumption with the correlation coefficient of the fit, as shown further on in Fig. 9.
To clarify this already here, we include the following sentence in Sect. 3.3.2.

In order to estimate the $D_0$ parameter, an exponential distribution is fit to the PSD (leaving out small particles with $D_{max} < 800 \ \mu$m since it was empirically noted that these did not follow this exponential behavior); instances when the assumption of an exponential tail is invalid can be identified by filtering on the correlation coefficient of this fit. This approach was chosen rather than computing moments from the in-situ PSDs, as those could potentially be affected by the size cut-off at 6.4 mm, while the slope of the distribution is expected to be a more robust indicator.

15. *L. 340: Could you expand a bit more on this aspect of spatial consistency? First I would like to understand better what the term means. I guess it means that certain spectral features (e.g. a second spectral mode) appear to be connected throughout certain range gates? 3 range gates appear to me quite a small range if I just look at your measured spectrum in Fig. 4a*

We thank the reviewer for raising this point which we believe is important. Reviewer #2 raised a similar question on what we meant with this expression, therefore we answer in the same way:

We propose a toy example to illustrate why we think our method helps reduce the ill-posedness by using the full spectrograms, instead of a gate-to-gate approach. Let us consider that for a given profile of length 10, at each range gate, there exists a given number (e.g. 3) of sets of microphysical properties that would translate into the same spectrum observed at this gate. Then, there would be potentially $3^{10}$ profiles of microphysical properties that would translate

into the same observed full spectrograms. We can reduce this number by imposing explicit constraints on smoothness (e.g. regularizing the first or second-order derivatives), or (which we do here) by using convolution kernels that pick up on the structure of the profile in the *measurements* themselves and allow to propagate this information to the profile of microphysical variables. Here, by "structure" (or "spatial consistency" in the manuscript) we refer to the fact that the spectrogram (or reflectivity profile) might be continuous, smooth (i.e. spectra at nearby ranges are similar), or on the contrary have some abrupt changes (e.g. in the case of high shear, where neighboring spectra might be very different).

In both cases (explicit or implicit constraint), by using the full profile instead of a gate-to-gate retrieval, we reduce the number of degrees of freedom and thus restrain the number of possible solution profiles to a number smaller than the initial $3^{10}$ in our toy example.

Undoubtedly, we cannot say that we can completely resolve the ill-posedness with this approach: there might still be several microphysical solutions that satisfy the "structure" or "spatial consistency" constraints. One way through which we mitigate (or at least quantify) this further is by using an ensemble approach (see Sect. 4.3, or Fig. 8 with the error bars indicating the spread), i.e., by training several times the neural networks with different random initializations in order to obtain not a single retrieved profile but several of them, on which we can compute usual statistics (mean, median, standard deviation, inter-quantile ranges, etc.). The spread of these values gives an idea of the uncertainty of the retrieval, and illustrates the remaining intrinsic ill-posedness.

To convey this more clearly, we extended the explanation of why/how the use of the full profile is an important aspect of the retrieval (see below).

Regarding the comment on the number of range gates used for these kernels: the number of 3 was found during the tuning of the algorithm, i.e., several values were tried and the one that provided the best results (total loss) was kept. Please note that the convolutions are sliding, meaning that the first convolution will look at range gates 1,2,3, the second one at range gates 2,3,4, etc. Additionally, each layer of the model contains convolution kernels (see Fig. 2): this series of convolution progressively increases the field of view. Ultimately, the output of the neural network at a given range gate is influenced not only by the closest neighbors but also by range gates which are further away. Therefore, it is difficult to interpret the size of the convolution kernel as a physical quantity. Finally, we should keep in mind that the dataset includes a large variety of spectrograms, some of which are much less smooth than the one shown in 4a); for instance, some of them may contain only a few shallow cloud layers so the spatial correlation is not very high. We realize that this statement in the original manuscript, intended to give an intuition on the role of convolution kernels, was misleading and a too strong statement. We rephrased this sentence to convey this better.

[Sect. 2] The architecture of the decoder and encoder will be detailed further on (Sect. 4), but one key property should already be underlined. The retrieval operates on the full dual-frequency Doppler spectrograms at once, rather than on each gate independently: the idea is to synergistically make use of the spatial structure of the measurements to reduce the ill-posedness. By "spatial structure", or "spatial consistency", we refer to the fact that the spectrogram might be continuous, smooth (i.e. spectra at nearby ranges are similar), or on the contrary have some abrupt changes (e.g. in the case of high shear, where neighboring spectra might be very different). By constraining the retrieval to output a profile of microphysical variables with a similar spatial structure, we restrain the number of degrees of freedom.

In practice, this is handled by the architecture of the encoder network, which contains convolution kernels: thanks to this feature, the model can capture the spatial structure of the Doppler

spectrograms, and propagate this information in a way that the output profiles are themselves spatially consistent. Note that while the issue of ill-posedness is mitigated, it is not entirely resolved, as there may remain some intrinsic under-determination and thus uncertainties.

[Sect. 4.2.1]  The size of the convolution kernels along the range dimension gives a sense of the scale at which we can expect meaningful spatial correlation —both in the latent space and in the measured spectrograms. It is, however, not directly interpretable as a physical quantity: as the NN contains stacks of convolution layers, the field of view progressively increases with the model's depth; the output of the NN at a given range gate is influenced not only by its closest neighbors but also by range gates which are further away.
* * *
16. *L. 388-389, Fig. 3: Couldn't it maybe also be related to insufficient range of assumptions in the training? For example, the range of densities assumed, the simple inverse exponential PSD? Do the two panels in Fig. 3 actually have the same velocity axis? Then the X-band spectrum is from rimed particles, while the W-band is for slight updraft conditions, right? Wouldn't it make more sense to show the spectra for the same time-height of a specific event? Both spectra look quite smooth. I would be interested in seeing the original spectra.*

The purpose of Fig. 3 is to show the performance of the decoder on the synthetic testing dataset (i.e., the 10% of the synthetic data set that was set aside before training/ tuning the algorithm). It does not show the output of the decoder for real spectra, when implemented in the full pipeline (shown later in Fig. 4). This evaluation is basically a safety check that the decoder is able to correctly emulate the forward model (i.e. PAMTRA with all the assumptions on scattering, microphysics, etc. detailed before) on a set of microphysical properties. Therefore, it does not tell us anything about the validity of those assumptions but merely informs us of the quality of our decoder to replicate the results of the physical forward model (by performing a high-dimensional non-linear regression of the synthetic data set it was trained on).

We realize that this was not clearly conveyed in the manuscript and especially in the caption of Fig. 3. We made the following modifications to clarify this point:

Examples of model outputs on the synthetic testing set are shown in Fig. 3.

[Fig.3 caption] Examples of results on the synthetic testing set of the decoders, showing the decoder output (dashed red) and the target PAMTRA-generated spectrum (black line) at a) W- and b) X-band. The examples were chosen to reflect some of the typical behaviors and possible artifacts that were observed; the X- and W-band examples do not correspond to the same microphysical properties.
* * *
17. *Figure 4 looks indeed very impressive! But for me it is hard to understand how you are able to reconstruct also areas of various levels of turbulence so well given that you only use one value of turbulence (EDR) in the PAMTRA simulations for training.*

[Figure]

Figure 3: W-band (left) and X-band (right) turbulent broadening retrieved with the algorithm compared to estimations following Shupe et al. [2008], with wind data from COSMO-1.

As mentioned in the response to Specific comment #11, multiple values of turbulent eddy dissipation rate are used in the PAMTRA simulations. This is indeed crucial to allow for the reconstruction of regions with different levels of turbulence.

It is computed by randomly sampling a value of atmospheric turbulence, represented by the eddy dissipation rate (sampled from a negative exponential with $10^{-3} m^2 s^{-3}$ mean, consistent with some literature standards, e.g. Sharman et al. 2014).
* * *
18. *Sect. 5.2.2: Correct me if I am wrong, but I guess you could also retrieve eddy dissipation rate, right? I think it could be nice to compare your EDR estimate with simpler methods such as those presented by Borque et al., JGR, 2016 (https://agupubs.onlinelibrary.wiley.com/doi/full/10.1002/ 2015JD024543).*

We thank the reviewer for raising this comment and suggestion, and providing this reference. What we retrieve is a global broadening parameter, which includes the effect of horizontal and radial wind, shear of radial wind, and indeed turbulence: $\sigma^2 = \sigma_B^2 + \sigma_S^2 + \sigma_T^2$ following the notations of Shupe et al. [2008], where the effect and calculation of each of these terms are nicely detailed.

Following Borque et al. [2016] and others [Bouniol et al., 2003, Shupe et al., 2008], we can estimate the same broadening kernel directly, using the variance of the Doppler velocity and wind values from model data (here, COSMO-1 is used).

In the figure below (Fig. 3), we can see that broadening values from our retrieval are reasonably similar to the estimates from literature methods, although the retrieved values are often larger than the direct estimates. The bias seems larger for lower values of turbulence / broadening. One possible cause would be the underestimation of the radial wind shear in model data (where the radial wind is vertical and has virtually no shear). Additionally, Borque et al. [2016] underline that errors in estimates of horizontal wind lead to a larger relative bias for small wind speeds; such an effect may also contribute to the bias we observe on the low-turbulent side.

A reasonable agreement is found (not shown) when comparing these values to broadening estimates derived through classical methods [Borque et al., 2016, Shupe et al., 2008], which rely on the variability of mean Doppler velocity and on wind profiles.
* * *
19. *L. 495: Looks like the in-situ confirms that the inverse exponential is not an ideal PSD shape. I would be interested to know if the higher spectral moments (for example skewness) are changing*

[Figure]

Figure 4: Left panels: measured PSD (black dots) with exponential (blue) and gamma (red) fit. The exponential fit uses only the larger particles (maximum dimension greater than 800 $\mu m$). Right panels: corresponding PAMTRA simulations of W-band spectra, with other microphysical descriptors unchanged.

*if you use the in-situ PSD instead of your fitted exponential. While certainly the larger particles dominate Ze and DWR, I am not sure that they don't affect the spectral shape.*

We fully agree with the reviewer that the shape of the PSD is likely to affect that of the spectra, especially in conditions with relatively low turbulence. As discussed in the response to General Comment #1, the arguable choice of the exponential distribution was made to restrict the number of parameters to retrieve and keep the computational costs reasonable.

To check this, we conducted the following analysis for each aircraft overpass. In addition to the fit of the exponential tail described in the manuscript, we fit a gamma distribution to the full PSD. We then simulate the PAMTRA spectra in both cases (keeping the other microphysical parameters unchanged), and compute the corresponding moments. We did not directly use the aircraft PSD for the main reason that it often contains few of the larger particles (the size limit is officially at 6.4 mm, but large particles might be undersampled even below this threshold), and this would affect artificially the spectrum's shape — while what we are mostly interested in here is the role of the smaller particles.

In Fig. 4 below, two examples are shown (with W-band); in the top example, the non-exponential behavior is restricted to very small particles and the spectra are almost identical. The lower panels correspond to a time step where the PSD is clearly closer to a gamma distribution (super-exponential) than an exponential one, and the spectra are indeed different.

To get a more general sense of this effect, we show in Fig. 5 the scatter plots of the first spectral moments (up to skewness) computed when a gamma fit or an exponential fit of the in-situ distribution.

As detailed in the response to comment #22, we included a new subsection in the revised manuscript (now Sect. 6.4) to discuss the role of PSD shape, where we specifically consider the changes in skewness caused by non-exponential behavior.

New subsection 6.4 **Shape of the particle size distribution**

[Figure]

Figure 5: Moments of the spectra simulated with PAMTRA using an exponential or a gamma fit of in-situ PSD, for all overpasses.

20. *Sect. 5.3.4: I guess some of the $A_r$ discrepancies could also result from the averaging over particle 2D projections done in common in-situ probes. I guess mentioning the study by Jiang et al., JAMC, 2017 (https://journals.ametsoc.org/view/journals/apme/56/3/jamc-d-16-0248.1.xml) could make sense here.*

Computing aspect ratio from 2-dimensional images (i.e. what is done here with the in-situ probes) can indeed create a certain bias. However, as pointed out by Jiang et al., 2017, this bias is different from what we observe here (i.e., estimates of $A_r$ from 2D are higher than actual 3D values for oblate spheroids). Hence, we believe that this is not the primary reason for the observed discrepancy; as explained in the text, we rather suspect the random orientation, together with the definition of aspect ratio in the SSRGA computations, to be the dominant effect.

Another element to consider is that aspect ratio is here assumed to be the same across the particle size distribution, which may well be an oversimplification. In particular, smaller particles may have smaller aspect ratios which would affect the aircraft-derived quantity differently than the retrieved value. This effect could be particularly strong in the case of super-exponential PSDs with e.g. columnar crystals dominating the smaller sizes.

Note that the aspect ratio values derived from the in-situ images are themselves prone to some bias, as discussed in Jiang et al., 2017, due to the projection of 3-dimensional particles in a 2-dimensional space. This bias would however be opposite to what is observed here, and is likely not dominant in our study. Another element to consider is that aspect ratio is here assumed to be the same across the particle size distribution, which may well be an oversimplification; in particular, smaller particles may have smaller aspect ratios which would affect the aircraft-derived quantity differently than the radar-based estimate.

21. *Fig. 12: Would be great to also have a similar plot showing the impact of absolute and relative*

[Figure]

[Figure]

Figure 6: Left: effect on the interquartile range of $b$ values when shifting the spectra with a constant velocity offset (same in X- and W-band). Right: effect on the retrieved variable $wind_W$ when shifting the W-band spectrum with a velocity offset(similar figure regardless of the velocity offset added to X-band).

*velocity offset (I guess it will impact m-D parameters quite a bit).*

When adding a constant or relative velocity offset, the retrieved mass-size coefficients are only minimally affected. This is due to the fact that a shift of the spectrum **(without change in shape)** is mostly "interpreted" internally by the algorithm as a radial wind shift (through the corresponding variable, see right panel of Figure 6 below) and corrected as such. When adding a wind shift within $\pm 0.5$ m s$^{-1}$, the root mean square of the change in $b$ values on the entire time series is within 0.05. Other microphysical variables are also barely affected.

One global effect can however be noted when adding significant constant velocity offsets: the spread of $b$ values reduces slightly. This seems to suggest that the algorithm is less able to pick up on changes in microphysical mass-size properties when the downward motion is dominated by external drivers. The amplitude of this effect is however not very strong, and we therefore decided not to include the detail of this discussion in the revised manuscript.

We included a comment on the role of velocity offsets:

[Sect. 6.1] We note (not shown) that shifting the spectra by constant or relative velocity offsets, to mimic one of the effects of radar mispointing, only minimally affects the retrieval of microphysical properties, but mostly translates in the radial wind variables.
* * *
22. *Sect. 6.3: I like the discussion of the impact of the SSRGA parameters on the D0 estimate. However, I think the fact that you are sticking to inverse exponential PSD might also explain some of the D0 bias. Mason et al., AMT, 2019 (https://doi.org/10.5194/amt-12-4993-2019) demonstrate this effect very nicely and I think this aspect should be included in the discussion.*

Following the reviewer's recommendation, we investigated further the sensitivity of the retrieval to PSD shape. We used a similar approach as the one we implemented to analyze the importance of SSRGA parameters, but focusing this time on the shape of the spectrum. A multi-dimensional sensitivity analysis is unfortunately quite difficult to conduct (i.e. ideally, we would like to investigate how the shape of the PSD influences the dual-frequency spectra as a whole: but this cannot be directly summarized through a single scalar quantity). We therefore decided to use

one shape metric of the spectra (skewness at W-band) and investigate how sensitive this parameter is to 1/ changes in $D_0$ if an exponential PSD is assumed and 2/ changes in $\mu$ if a Gamma PSD is assumed ($N(D) = N_0 D^\mu exp(-D/D_1)$) while keeping the effective diameter constant equal to $D_{eff} = 3 D_0$. We thus simulated relative changes in W-band skewness caused by $D_0$ offsets, on the one hand, caused by varying $\mu$ from -2 to +5 [Mason et al., 2019]. Overall, we obtain a similar figure as Fig. 14 of the original manuscript, as shown below (Fig. 7). The resulting intuition is that, as expected by the reviewer, the spectrum shape is quite sensitive to the underlying particle size distribution, and this may also play a role in the observed $D_0$ bias.

[New subsection 6.4] **Shape of the particle size distribution**
Another underlying hypothesis that was made when designing the pipeline and defining the set of microphysical descriptors was to consider only exponential particle size distributions. This choice was made to keep a minimal number of retrieved parameters at this stage. It is however known that multi-frequency signatures, on the one hand, and Doppler spectra, on the other hand, are both affected by PSD shape (e.g., resp. Mason et al. [2019], Barrett et al. [2019]). In this subsection, we conduct a similar analysis as that on the sensitivity to SSRGA coefficients: this time, as changes in the PSD shape mostly influence the shape of the spectrum, we focused on the skewness of the W-band spectrum (instead of the DFR) as a metric to understand how changing the shape of the PSD could influence the retrieval. A gamma distribution was assumed ($N(D) = N_0 D^\mu \exp(-D/D_1)$, e.g. Petty and Huang [2011]), constraining $D_1$ by keeping the effective diameter constant ($D_{eff} = 3D_0$), and varying the shape parameter $\mu$ in the range [-2, +5] as observed in snowfall (e.g., Mason et al. [2019]). Figure 7 illustrates that changes in the PSD shape may have a similar effect on W-band spectrum shape as varying $D_0$ of approximately 1 mm (-0.9 to +1.5 mm). Here again, this observation is mainly qualitative and cannot be directly used to quantify the influence of PSD shape on our retrieval, but it does underline that considering more complex distributions would be necessary to further refine the framework, and that the assumption of an exponential behavior may also have a role in the observed $D_0$ bias.

[Figure]

Figure 7: Colored lines with scattered points: relative change in W-band skewness ($\Delta \gamma_W/\gamma_W$) caused by adding a diameter offset $\Delta D_0$ on microphysical descriptors of selected (time, range) gates, if assuming an exponential PSD. Horizontal lines: for each of these (time, range) gates, maximum relative change in W-band skewness caused by a modification of the PSD shape (assumed a gamma distribution, $\mu$ in the range [-2, +5], c.f. text.). For each selected (time, range) gate, the intersection of the horizontal and colored lines gives a $\Delta D_0$ value which causes the same relative change in skewness as a change in PSD shape (worst case). Dashed vertical lines show the mean of these $\Delta D_0$ values.

[Figure]

Figure 8: Examples of measured and reconstructed spectra corresponding to different DFR values. The time, range and DFR are indicated in the titles. Note that the y-axis differs between the plots to allow a proper visualization.

23. *Figure 9 and D0 discussion: What I find quite puzzling is that your retrieval is never producing D0 values lower than 1mm despite very low DFRs and Ze at some time periods where you have overpasses. Couldn't it maybe be a relative bias in Ze? Can you show reproduced and measured X and W-band spectra plotted over each other once for a low DFR and once for a high DFR region? Ideally there should be no big offset in the y-direction except the different noise levels.*

Like the reviewer, we were puzzled by the systematically larger $D_0$ values output by the retrieval. Note that the 1mm apparent limit is not strictly true, as visible for instance in the timeseries of Fig. 5 where lower values are observed. We do not think this bias is caused by a relative bias in Ze: in the figure below, we follow the reviewer's suggestion and plot three examples of measured and retrieved spectra corresponding to different DFRs (indicated in the titles of the panels). As expected, the spectra are quite similar for small DFR values (apart from the different noise levels and the larger spectrum width at X-band) while the change is quite visible in the center and right plot as the DFR becomes larger.

As it does not seem related to a systematic bias in the data, we believe that the error in size is caused by excessive approximations in our training dataset: in particular (as discussed in Sect. 6), we identify as possible causes the under-representation of small particles in the MASC dataset, the assumption of an exponential distribution (as also underlined by the reviewer), and the sensitivity to SSRGA coefficients.

24. *L. 635: Just a comment: Airborne triple-frequency data as used in Mroz et al., AMT, 2021 might not be very useful for your approach as the spectra are probably affected by the aircraft motion. There are ground-based triple-frequency datasets available which provide quite acceptable 3f-spectra (see for example as shown in another article by Mroz et al., AMT, 2021 (https://amt.copernicus.org/articles/14/511/2021/amt-14-511-2021.html) or von Terzi et al., 2022 (https://doi.org/10.5194/acp-2022-263).*

We thank the reviewer for suggesting these additional references; we initially only mentioned Mroz et al. [2021] since it focused on a retrieval approach. We clarified that they do not use Doppler spectra, and we included the other references:

Given the convincing results obtained recently with triple-frequency data (e.g. the retrieval

of snowfall properties from triple-frequency radar moments proposed in Mroz et al. [2021], or studies of Mróz et al. [2021] and von Terzi et al. [2022] where triple-frequency spectra are used to study respectively the melting and dendritic growth layers), it is likely that our method would gain in robustness and precision with the inclusion of an additional frequency.
* * *
**Typos**

- L. 58: "studies, however, rely": Fixed.

- L. 206: "We, however, believe": Fixed.

- Table 1: I think there is a mistake in the description of $am/bm$ and $alpha_a/beta_a$. It says both "... of the mass-size power law": Fixed.

- L. 348: in this way: Fixed.

- L. 360: I guess the "1" with the double line in the equation denotes a vector of ones if the condition in parenthesis is fulfilled? : We added a clarification Here $1_{(x<x_{min})}$ denotes the function which is equal to 1 when $x < x_{min}$ and to 0 otherwise.

- Caption Fig. 5: Change DFR into $DFR_{X,W}$ as in the Figure: Fixed.

- Figure 5: If possible enlarge font size of the labels on the axis and color bars: Fixed.

- Also information on time should be added on the x-axis (I guess it's time in UTC); also in Fig. 7: Fixed.

- L. 486: with, however, the: Fixed.

- L. 513: This is not: Fixed.

**2    RC #2 - Anonymous referee**

*This manuscript presents a novel technique to invert snowfall microphysics from dual-frequency Doppler spectra by using a deep learning approach. The approach is based on an autoencoder-like framework, which links two neural networks together – one reproducing a well-established forward model, while the other is trying to invert the first one. Besides the implementation of the framework, the manuscript also presents its application to a recent field campaign where ground-based cloud radar measurements are compared with airborne in-situ data to discuss the performance of the inversion. I really enjoyed reading this manuscript, since it does a great job to introduce and explain its neural network approach instead of leaving the reader with a dark box like some other manuscripts. To my knowledge, this study is one of the first to apply the autoencoder concept to the inversion of cloud microphysics. The paper is well structured, easy to follow and engages the curiosity of the reader. Due to this fact, this paper could help to pave the way for further neural network assisted inversions of cloud microphysics. Like RC1, I was impressed by the retrieval results and the comprehensive consideration that went in the necessary assumptions for the forward problem. My comments are only minor in nature and mainly are concerned with some aspects of how these necessary assumptions were introduced and the occasionally overconfidence in the proposed method. Albeit powerful, there are certain limitations in the inversion of cloud radar measurements which even neural networks will not be able to solve. Anyway, these comments only concern the presentation and not the study itself*

*and should therefore be easy to address. Overall, I am convinced that the presented manuscript is a very valuable contribution to the scientific debate and will advance the application of neural networks in atmospheric science. I clearly suggest the manuscript to being published in AMT after my comments and questions have been addressed.*

We thank the reviewer for their positive feedback on the retrieval method and on the manuscript, and for their valuable suggestions. We fully agree that the proposed approach would not be able to overcome some fundamental challenges, and we tried to stress this point in the revised version of the manuscript.
* * *
**General comments**

1. *Throughout the manuscript you write that your approach can mitigate the ill-posedness of the problem, "i.e. when several values of x may yield similar outputs y: in such cases, the [direct] retrieval may yield arbitrary outputs" (L152). However, I am not convinced that your approach is able to do just that. The ill-posedness is determined by the number of measurements and the degrees of freedom and I do not see a way how your approach can overcome this mathematical principle. Where a "direct", e.g., lookup table-based retrieval may yield a set of possible x which can explain y, your approach only yields one unique y = f(x) relationship, although multiple combinations of x would still be physically viable. In the first case I obtain a measure of uncertainty while in the latter I am just getting "blind" for other latent variable combinations which also could explain my individual measurement. What you are actually doing during the construction of your decoder (by important sampling of x) and your encoder (by using specific snowfall cases) is the limitation of the ill-posed problem by well-chosen priors. That is an appropriate strategy to deal with the ill-posedness of the problem but it is not unique to your technique. Maybe your manuscript could be a little bit more modest around these sentences and acknowledge your use of priors for x (the need … to assume prior values for x … has led us to explore a different direction, L143).*

We thank the reviewer for raising this point which we believe is important. Reviewer 1 raised a similar question, which we answer in the same way:

We propose a toy example to illustrate why we think our method helps reduce the ill-posedness by using the full spectrograms, instead of a gate-to-gate approach. Let us consider that for a given profile of length 10, at each range gate, their exists a given number (e.g. 3) of sets of microphysical properties that would translate into the spectrum observed at this gate. Then, there would be potentially $3^{10}$ profiles of microphysical properties that would translate into the observed full spectrograms. We can reduce this number by imposing explicit constraints on smoothness (e.g. regularizing the first or second-order derivatives), or (which we do here) by using convolution kernels which pick up on the structure of the profile in the *measurements* themselves and allow to propagate this information to the profile of microphysical variables. Here, by "structure" (or "spatial consistency" in the manuscript) we refer to the fact that the spectrogram (or reflectivity profile) might be continuous, smooth, or on the contrary have some abrupt changes (e.g. in the case of high shear). In both cases (explicit or implicit constraint), by using the full profile instead of a gate-to-gate retrieval, we reduce the number of degrees of freedom and thus restrain the number of possible solution profiles to a number smaller than the initial $3^{10}$ in our toy example.

Undoubtedly, we cannot say that we can completely resolve the ill-posedness with this approach: there might still be several microphysical solutions that satisfy the "structure" or "spatial consistency" constraints. One way through which we mitigate (or at least quantify) this further is

by using an ensemble approach (see Sect. 4.3), i.e., by training several times the neural networks with different random weights initializations in order to obtain not a single retrieved profile but several of them, on which we can compute usual statistics (mean, median, standard deviation, interquantile ranges, etc.). The spread of these values gives an idea of the uncertainty of the retrieval, and illustrates the remaining intrinsic ill-posedness. In this way, we characterize the uncertainty component associated to the *model* itself; uncertainty associated to the input data can be assessed through sensitivity analyses (e.g. what is done in Fig. 12).

To convey this more clearly, we extended the explanation of why/how the use of the full profile is an important aspect of the retrieval (see below), and we detail slightly more how the ensemble approach helps monitor the remaining ill-posedness.

Regarding the reviewer's comment on the use of priors: one key difference between classical Bayesian methods and machine-learning-based techniques is that the former requires to *explicitly* define a prior value —which serves as a starting point "$x_0$" for the retrieval algorithm— while the latter uses *implicit* priors —defined by the training dataset. If the training dataset is well-balanced and essentially covers the entire range of possible values, then this is quite a light prior, which is unlikely to cause systematic biases.

We surely do not want to convey a misleading message to the reader; while we are convinced that leveraging the spatial consistency is a key point, and that the framework overall allows reducing certain hypotheses compared to classical methods, we are also well aware of its limitations (e.g. fundamental ill-posedness, still a few strong assumptions on microphysics and scattering, computational constraints, not directly usable, ...). We hope that our discussion/outlook sections cover in some detail at least a few of these points. Following the reviewer's suggestion, we additionally toned down a few sentences in the introductory sections to make this visible already at this stage.

[Previously l. 170–174] The architecture of the decoder and encoder will be detailed further on (Sect. 4), but one key property should already be underlined. The retrieval operates on the full Doppler spectrogram at once, rather than on each gate independently: the idea is to make use of the spatial structure of the measurements to reduce the ill-posedness. By "spatial structure", or "spatial consistency", we refer to the fact that the spectrogram might be continuous, smooth (i.e. spectra at nearby ranges are similar), or on the contrary have some abrupt changes (e.g. in the case of high shear, where neighboring spectra might be very different). By constraining the retrieval to output a profile of microphysical variables with a similar spatial structure, we restrain the number of degrees of freedom.
In practice, this is handled by the architecture of the encoder network, which contains convolution kernels: thanks to this feature, it can capture the vertical structure of the Doppler spectrograms, and propagate this information in a way that the output profiles are themselves spatially consistent. Note that while the issue of ill-posedness is mitigated, it is not entirely resolved, as there remain some fundamentally under-determined features and thus uncertainties.

[Previously l. 142–147] This "classical Bayesian" approach faces some limitations, which include but are not limited to, the need to linearize the forward operator in order to compute its Jacobian, or to explicitly assume prior values for $x$.
Alternatively, machine learning techniques offer the possibility to tackle inverse problems with a statistical [...]

[Sect. 4.3 **Ensemble approach for uncertainty quantification**] This is especially relevant given the under-determination of the problem: with this ensemble approach, we can illustrate the uncertainty related to the remaining intrinsic ill-posedness of the model.

*This includes an earlier mentioning of the fact that your approach is directly trained on the data of interest and might thus not be directly suitable as a general retrieval for arbitrary X- and Ka-band spectrograms of ice clouds (first mentioned at L402ff).*

We agree that this is an essential aspect, and we include it at the end of Sect. 2:

One intrinsic limitation which should be highlighted is that the method is trained directly on the data of interest and cannot be directly used on any given measurements.
* * *
*Moreover, while the introduction to the theory of inverse problems (Sec. 2.2) is thorough and almost too detailed (L133-L145), I am missing a short literature review about current studies using neural networks (e.g., Piontek, MDPI, 2021), especially auto-encoders (e.g., Behrens et al, JAMES, 2022) in climate research.*

We thank the reviewer for these suggestions of references which, albeit not directly related to cloud or precipitation microphysics, are more than relevant in our introduction and very useful to put our method into perspective. We slightly expanded Sect. 2.2 to include an overview of this topic, with the proposed references and a few additional ones.

Ultimately, this produces a gate-to-gate inversion of the problem which can be implemented on real data. This approach has been successfully used for atmospheric retrievals, for example by Piontek et al. [2021] to detect volcanic ash clouds, Vogl et al. [2022] to estimate riming occurrences from radar measurements, or Chase et al. [2021] to retrieve snowfall properties from airborne or satellite radars.

One major limitation of this "direct" gate-to-gate method is when the problem itself is ill-posed, i.e. when several values of $x$ may yield similar outputs $y$: in such cases, the retrieval may yield arbitrary outputs.

The proposed approach, illustrated in Fig. 1, can mitigate this issue. It is inspired from auto-encoder architectures [Kramer, 1991, Hinton and Salakhutdinov, 2006], which use neural networks to perform powerful non-linear dimension reduction: an *encoder* network maps a high-dimension signal to a low-dimension latent space, while the *decoder* network learns to reconstruct the original high-dimensional signal from the latent space. Such tools are relevant for atmospheric sciences and in particular in the context of climate studies, which handle complex, high-dimensional signals [Behrens et al., 2022]. In our case, the aim is to constrain the dimensions of the latent space to contain microphysical descriptors: the originality of the approach presented here is thus that it incorporates physical knowledge by using a physics-informed decoder.
* * *
2. *An aspect that is in general a bit unclear to me is your statement in e.g., L94: "... an important peculiarity of the encoder's architecture is its ability to leverage the spatial consistency of the radar variables, which reduces the ill-posedness of the inversion problem" which is iterated several times throughout the manuscript. Unless I have missed it you never mention what you mean with "spatial consistency" in the first place and only briefly describe its implementation around L340. This was also mentioned by the other referee, and we only can guess that you refer to the observation how spatial features (e.g., updrafts or fall streaks) are connected throughout adjacent range gates. Furthermore, it is not clear to me how this "spatial consistency" can overcome the ill-posedness of the inversion problem in each individual gate since we simply have*

*too few measurements compared to the natural variability of realistic ice crystals (see my point 1). In my opinion, your spatial convolution kernels (which you only use for your encoder) lead to a smoother profile of latent variables (which is desired!) but cannot overcome the underdetermination of the problem (e.g., DFR ambiguity between aspect ratio vs. D 0 )..*

We believe that our response to the comment #1 addresses the points raised here. As mentioned, by "selecting" the solutions of the inverse problems which are spatially consistent (see the definition provided above and included in the revised manuscript), we reduce the number of degrees of freedom —only a subset of all possible solutions will satisfy this constraint (which is implicitly imposed by the CNN). Nonetheless, we agree that a fundamental ill-posedness can remain, as we now highlight in the revised manuscript. Note that the ensemble approach that we additionally implement by training multiple models helps characterize some of this residual uncertainty.

By "spatial structure", or "spatial consistency", we refer to the fact that the spectrogram might be continuous, smooth (i.e. spectra at nearby ranges are similar), or on the contrary have some abrupt changes (e.g. in the case of high shear, where neighboring spectra might be very different). By constraining the retrieval to output a profile of microphysical variables with a similar spatial structure, we restrain the number of degrees of freedom.

Note that while the issue of ill-posedness is mitigated, it is not entirely resolved, as there may remain some fundamentally, physically under-determined features and thus uncertainties.
* * *
3. *Like RC1, I am missing a discussion how attenuation by hydrometeors (especially at W-band) could introduce biases in your inversion. In your outlook, you could briefly discuss potential approaches to this problem in the framework of neural networks. Here, it might become necessary to train the decoder on full spectrograms instead of treating each range gate independently to capture the gate-to-gate interdependence caused by attenuation effects.*

We fully agree with the reviewer that the question of attenuation, especially at W-band, is indeed a crucial one for this retrieval (as for many). As this point was also raised by RC#1, we respond in a similar way to highlight how 1/ the current version requires attenuation-corrected data and 2/ possible ways through which attenuation could be incorporated in future versions of the retrieval.

In its current state, the algorithm is unfortunately **not able** to handle attenuated spectrograms. Retrieving attenuation would require a slightly more involved set-up for the training of the algorithm: currently the *decoder* part of the framework operates in a "gate-to-gate" manner, as it is trained on individual spectra and not on profiles. We believe this is a key point: if we were to train the decoded on full profiles, the size of the training dataset needed would be extremely large to account for the diversity of possible atmospheric / cloud profiles. However, retrieving values of attenuation requires to take into account the full profile in the decoding part of the framework (since attenuation is cumulative).

Nonetheless, we believe that the framework could be adapted for this for example by 1/ adding an intermediate module computing the resulting attenuation from the retrieved microphysical properties and correcting the input and reconstructed spectrograms by adding to them the cumulated attenuation profile obtained, and 2/ by recursively applying the inversion model till convergence. This method, illustrated in Fig. 9 below, would allow to correct for attenuation by ice hydrometeors. Still, a major source of attenuation at W-band is the presence of supercooled liquid water, which is often not easily identified. Accounting for it would require to identify the

presence of a secondary mode corresponding to liquid water (and this mode might not always be visible, especially in turbulent cases) and retrieve the LWC from this signature: implementing these steps to our retrieval requires some non-trivial modifications.

Another way to go would be to directly retrieve an attenuation profile (which would account for gaseous, snow and liquid water attenuation) in parallel to the other microphysical properties and correct the reconstructed spectrograms by adding to them the cumulated attenuation obtained, as illustrated below in Fig. 10 Without other constraints on the attenuation, the risk is that the ill-posedness would become too strong. One could think of constraining it using additional measurements of brightness temperature (which could also be simulated with PAMTRA).

Overall, the implementation of an additional attenuation retrieval was not implemented in the first version of the algorithm because it raises shome challenges in the training of the algorithm as well as some non-trivial adjustments of the framework, although we believe some improvements/developments could surely be made along this direction.

Thus, in this manuscript we work with data that were **a-priori corrected for attenuation using more classical methods**, although these are inevitably also error-prone. In the revised version of the manuscript, we include a more thorough correction of ROXI's calibration drift (using MXPol as a reference) and of W-band attenuation, following the approach described in Kneifel et al. 2015, using LWP values to correct for supercooled liquid water attenuation and X-band reflectivity to account for snowfall attenuation. This is included in Appendix A, which was extended compared to the original manuscript. Note that these modifications only result in minimal changes on the results (barely visible in Fig. 8, and minor change of the RMSE / bias values).

The challenges of incorporating attenuation as a retrieved variable in this first stage of development is what lead us to discuss the effect of reflectivity biases on the retrieval outputs (in Sect. 6.1). We modified slightly this section, including its title, to highlight the importance of possible remaining attenuation effects, and how they motivate this discussion.

[Section 3.2] As pre-processing steps, the radars are cross-calibrated and an attenuation correction is implemented at W-band (similar to Kneifel et al., 2015), as detailed in Appendix A.

See modifications to Appendix A [Radar calibration and W-band attenuation correction]

[Modification of Sect. 6.1] Sensitivity to miscalibration and differential attenuation:
One limitation of our framework is that it requires a good calibration of the radars —both absolute and relative— as well as an independent correction of attenuation. As detailed in Appendix A, the issue of attenuation was tackled by implementing a correction of W-band reflectivity based on estimates of gaseous, snowfall and liquid water attenuation. This correction method is however error-prone and we cannot exclude that reflectivity biases are present in the measurement dataset. We highlight that the presence of supercooled liquid water cloud layers or wet snow can be particulary difficult to identify and diagnose, while attenuating strongly millimeter-wavelength signal (with e.g., path-integrated attenuation up to 5 dB for liquid water paths of 500 g m$^{-2}$, Kneifel et al. 2015). To assess the possible importance of inaccurate calibration or attenuation correction on the retrieval, we investigate its sensitivity to reflectivity offsets, both absolute and relative.

[Conclusion] One drawback of the algorithm in its current state is that it relies on attenuation-corrected data. Further improvements of the method could include the retrieval of an attenuation profile, used to correct the spectrograms within the pipeline itself in a recursive way.
* * *
*Moreover, you promise (in your abstract) to relax constraints on beam matching with your approach, but only seem to consider a potential doppler shift (hopefully independently between*

[Figure]

Figure 9: First example of how the framework could be adjusted to retrieve attenuation. The decoder is adapted to learn not only the Doppler spectra but also the corresponding attenuation (we include both X- and W-band in the diagram, but possibly only W-band could be considered).

[Figure]

Figure 10: Second example of how the framework could be adjusted to retrieve attenuation. In this case, the encoder retrieves a cumulative attenuation profile. A constraint on total attenuation could be added to the loss term.

*X- and W-band) caused by a beam misalignment. The more fundamental limitation caused by a nonuniform beam filling in the context of different radar beam widths (0.53° vs. 1.8°, 50 m vs. 150 m beam diameter at 5 km altitude) is not handled but should shortly be mentioned.*

We thank the reviewer for noting this mistake. We are indeed focusing on the issue of beam misalignment rather than general mismatching; our algorithm does not handle the limitation of non-uniform beam filling.

In the abstract and introduction, we replace the wording "beam (mis)matching" with "beam (mis)alignment".

Additionally, we comment on the issue of non-uniform beam filling in the corresponding paragraph:

[Formerly l. 72–76] Difficulties related to imperfect measurements are substantial: not only should the different radars be well cross-calibrated in reflectivity, they should also be well aligned vertically to avoid contamination by the horizontal wind. The additional issue of non-uniform beam filling is all the more problematic when the radars have different beam widths or range resolutions: this would hinder the retrieval, especially when turbulent broadening is observed, or when the particle populations in the sampled volumes are too heterogeneous.
* * *
4. *Throughout the manuscript the used definition of the particle diameter is not entirely clear to me and needs a more precise treatment. You introduce D 0 as size parameter of the PSD of Straka (2009) and you call it its "mean diameter" in L191. According to Straka (2009), however, D 0 is the median volume diameter of the exponential size distribution if D = D max . Furthermore, your fixed relation D eff = 3D 0 seems wrong to me for ice particles regardless your definition of D 0 . The effective diameter regarding solar radiation depends rather on the area size relationship. As pointed out by RC1, your strategy how you relate the aspect ratio with particle mass and size is not clear to me – it appears to be based on the common soft spheroid approximation. Regarding this approach and the discrepancies found between your retrieval and in situ measurements you should consider the remarks made e.g., by Hogan et al (2012). They show how particle shape and orientation influence the mean diameter retrieved by the DWR technique compared to in situ measurements. Likewise, I am missing a more explicit description how the diameter, the area and the terminal velocity are connected. While I am convinced that the authors are aware of these points and have put sufficient care into their implementation, their manuscript lacks the necessary diligence regarding the particle diameter.*

In general, we define the size/diameter of a single particle ($D$) as its maximum dimension, as stated in Sect. 3.1.1. If the particle size distribution is an inverse exponential ($N(D) = N_0 \exp(D/D_0)$) then $D_0$ is the mathematical mean of this distribution (which we can obtain by integrating $\int_0^{+\infty} DN(D)dD = D_0$), i.e. it is the number-concentration-weighted mean diameter. This is also detailed in Straka 2009, Chapter 2.5.3 (Eq. 2.56). Therefore, we sometimes refer to $D_0$ as the mean diameter.

Regarding the effective diameter, we realize that this requires to be defined. In certain radiative transfer models (such as PAMTRA) the effective diameter used is defined as the ratio of the third to second moment of the PSD: $D_{eff} = \frac{\int_0^{+\infty} D^3 N(D)dD}{\int_0^{+\infty} D^2 N(D)dD}$. In this case, we can calculate the values of both integrals with $N(D) = N_0 \exp(D/D_0)$ and it ultimately comes that $D_{eff} = 3D_0$.

This can also be found in Straka [2009], Chapter 2.5.6 (Eq. 2.70).

We attempted to make the definitions of particle size and of the various size-related parameters as clear as possible by rephrasing this paragraph in Sect. 3.1.1:

The PSD is assumed to be a negative exponential ($N(D) = N_0 \exp(-D/D_0)$), e.g. Straka [2009], whose size parameter $D_0$ is prescribed; here and further, the size or diameter of a particle is defined as its maximum dimension: $D = D_{max}$. For an exponential PSD, $D_0$ is equal to the number-concentration-weighted mean diameter (shortened as "mean diameter"); the effective diameter (ratio of the third to second moment of the PSD), often relevant for radiative transfer models, is $D_{eff} = 3D_0$.

The aspect ratio is used in the forward model to scale the diameter in the SSRGA calculations. The corresponding equations can be found in Hogan et al. [2017] and Ori et al. [2021](see for example Eq. 6 and 7 in the latter). We realize that the definition of the aspect ratio was not properly included in the first version of the manuscript (it only appeared in further sections). We therefore included its definition in Sect. 3.1.1.

Aspect ratio $A_r$ is then specified, defined here as equal to the particle's dimension along the direction of the radar beam (here, vertical) divided by maximum dimension (Ori et al. 2021).
* * *
**Specific comments**

1. *L37: What do you mean when you write: "the retrieval of snow microphysics from radar variables is not explicit"?*

   The word "explicit" was meant in a mathematical way, i.e., it would be explicit if we had an analytical expression of a function $f$ such that $snowfall\_microphysics = f(radar\_variables)$. However, this wording can cause confusion and is not necessary, so we rephrased this sentence.

   Yet, such measurements are indirect and there is no known analytical expression to derive snowfall microphysical descriptors from radar measurements.
* * *
2. *L47: "[DFR] can thus be used to identify populations of snow particles with a larger size or density." How does density influence the DFR sensitivity? I always thought that the DFR technique is inherently insensitive to density of ice particles?*

   Here our wording was not rigorous enough: by density, we referred to the bulk density of the particles and not their concentration in a volume of air. It is likely more accurate to refer to the particle's fractal dimension or to their degree of riming, since the effect of enhanced density is not easily separate from the resulting change in shape. Overall, these variables influence DFR less than particle size, but they is still non-negligible: this is notably a motivation for the use of triple-frequency measurements to diagnose riming, see e.g. Kneifel et al. [2011] and Kneifel

et al. [2015], or more recently Tridon et al. [2022], and a thorough explanation of those effects in Battaglia et al. [2020]. To remove any ambiguity, we rephrased as follows:

[DFR] can thus be used to identify populations of snow particles with a larger size, or with a higher degree of riming (e.g. Matrosov 1992, Matrosov 1998, Szyrmer 2014, Liao 2016, Battaglia 2020).
* * *
3. *L76: "especially when turbulent broadening is observed [...], a direct computation of the dual-frequency spectral ratio is meaningless." This statement is too harsh in my opinion. While turbulent broadening can severely hamper the exploitation of the dual-frequency spectral ratio, you should give an estimation at which magnitude its value becomes "meaningless"*

Our goal was to convey that the combined effect of beam misalignment (resulting in velocity shift) combined with differential spectral broadening makes the dual-frequency spectral ratio extremely difficult to interpret and even to correct (it might be negative in certain velocity ranges, etc.). But we acknowledge that this was a too harsh statement. We tempered it:

In such cases, a direct computation of the dual-frequency spectral ratio is difficult to interpret and may be dominated by these artifacts.
* * *
4. *Fig. 1: A little bit more descriptive caption would help the reader here.*

We followed the reviewer's suggestion and we expanded the caption to make the figure hopefully more stand-alone.

Schematic illustration of the method. The notations are those of Sect. 2.2. The upper right box illustrates that the decoder NN is trained to emulate a forward radiative transfer model. The lower box shows the full pipeline where the pretrained decoder is used to reconstruct full spectrograms based on the microphysical properties output by the encoder NN.
* * *
5. *L232: On what observations or studies are these numbers based? Could the choice to give aggregates more weight during the training not also bias your retrieval towards the property of aggregates? The (40/20/20/20%) distribution is thus part of the implicit prior of your approach, correct?*

These numbers were initially based on the statistics of the MASCDB. However, in those, aggregates were even more represented ($> 50\%$) while for instance only a few planar crystals were observed. This is in part due to the fact that the classification of planar crystals is quite challenging (because of the different appearance a planar crystal may have depending on its viewing angle), hence some of them may be classified in MASCDB as "small particles", a category corresponding to particles which are not easy to categorize.

We therefore decided to reduce the preponderance of aggregates to 40% and set the other particle types at 20%.

Nonetheless, we were reluctant to split evenly the particles because the range of values of the microphysical descriptors was often larger in the case of aggregates and it seemed relevant to keep this variability in our dataset (in particular, $D_0$ values can span a larger range of values than for the other particle classes). Aggregates can be extremely diverse (e.g. small dry dendritic aggregates, vs. big rimed aggregates, ...) and we wanted our synthetic dataset to keep some of this diversity.

In short, we agree with the reviewer that we may bias slightly our retrieval toward aggregates by using these numbers. However, with an even split of particle types, we would have risked missing out on some of the variability that exists within the aggregate class.

We modified slightly this sentence to indicate where the bias comes from (MASCDB observations):

Given the large variety that exists within the aggregate category, as observed in MASCDB, it is given more weight in the sampling procedure (aggregates: 40% - planar crystals: 20% - graupel: 20% - columnar crystals: 20 %).
* * *
6. *Sec 3.2: Although your accompanying paper Billault-Roux et al (2022) might provide these numbers, a reader might be interested in the size (hours or number of profiles) of your training dataset.*

We agree with the reviewer that this information should be included, and we added it :

Only time frames with precipitation are used in this study, i.e. with detectable signal in both frequencies, leading to a total of $\sim 9000$ profiles corresponding to around 50 hours of measurements, collected between January 16th and January 28th.
* * *
7. *L314: I really appreciated your clear and comprehensible description of your neural network approach. I would be delighted if you could spend 1-2 more sentences on the role of these residual blocks. Do we know why or how they facilitate the training process?*

Residual blocks are indeed quite valuable tools in deep learning. Historically, when deep learning was developed by "stacking" together simple layers of neural networks, some issues were encountered during the back-propagation step (i.e. during this step of the learning algorithm, we compute the gradient of the final loss with respect to each coefficient of the model; then, each of these coefficients is updated so as to reduce the loss). It was observed that this gradient was close to 0 in the early layers of the model. This means that the early layers are virtually useless in the model, and this effect of "vanishing gradients" makes the training inefficient. One effect of the residual blocks, with their *skip connections*, is to mitigate this effect by propagating the information of the early layers to further stages of the NN, which in turn allows sustaining the amplitude of the gradients throughout the depth of the NN. Note that the reasons why residual blocks are so efficient are not completely understood and are still a subject of debate [e.g. He

et al., 2016, Veit et al., 2016, Balduzzi et al., 2017]: this is why we prefer not to go into a too detailed discussion of these techniques.

We hope to convey some of the intuition of this effect by including a few additional sentences in this paragraph.

In a nutshell, these techniques help mitigate issues caused by the depth of the model: they do not per se improve the expressiveness of the neural network, but they strongly facilitate the training process. For instance, the skip connections allow to propagate information from earlier layers to further stages of the neural network, and this reduces the risk of gradients vanishing to zero during training [Balduzzi et al., 2017].
* * *
8.  *L318ff: You mentioned the separation of your measurements into an 80% training, 10% validation and 10% testing data set. In the following, you no longer refer to how you used the 10% validation and 10% testing data set or did I miss something?*

We clarified the use of the training/validation/testing sets by including another sentence at this point. Additionally, we modified the description of Fig. 3 (both in the text and in the caption) to highlight that it is based on the synthetic testing set.

The NN is trained on the synthetic dataset described in Sect. 3.1, which is split into training, validation and testing sets (80% - 10% - 10%). The NN is trained on the training set, while the validation set is used to tune the architecture of the NN, and the testing set for a final assessment of its performance (Sect. 5.1.1).

Examples of model outputs on the synthetic testing set are shown in Fig. 3

[Fig. 3 caption] Examples of results on the synthetic testing set of the decoders, showing the decoder output (dashed red) and the target PAMTRA-generated spectrum (black line) at a) W- and b) X-band.
* * *
9.  *Tab 3. Could you elaborate where the choice of 30 for the number of channels comes from? I thought that the channel dimension relates to the radar bands used (X- and Ka-band).*

The number of channels is one aspect of the convolutional neural network's complexity. In the *"inner" layers* of the model, the number of channels essentially corresponds to the number of different convolution kernels which are learned at each layer. The number of 30 was found during the tuning of the model's architecture, i.e., we increased the number of channels until no further improvement was noted. However, the reviewer is right that the X- and W-band spectrograms constitute the two *input* channels of the encoder. We tried to make this distinction by including a new line in the table:

| Hyperparameter | Decoder | Encoder |
|---|---|---|
| Number of input channels | 1 | 2 |
| Number of inner channels | 30 | 30 |

10. *L377-L381: The description of the loss function O(S, S) is quite complicated and hard to understand. Explain the problem with the difference between normalized spectra and why your loss function consists of two integrals. How can you spot discrepancies in the absolute reflectivity at all when you normalize the simulated and measured spectra with each other?*

The overlap metric is introduced to have a more easily-interpretable quantity to monitor the quality of the decoder, as the MSE itself (which is the loss computed for training) can be difficult to interpret. We recall the definition:

$$O(S, \tilde{S}) = 0.5[\frac{\int min(S^*, \tilde{S}^*)}{\int S^*} + \frac{\int min(S^*, \tilde{S}^*)}{\int \tilde{S}^*}]$$

To explain the definition of this overlap, the figure below can be useful (note that the spectra are not real ones, they were drawn for a purely illustrative purpose).

(a) $S$ is the reference spectrum (target), and $\tilde{S}$ is the model output (whose quality we want to assess).

(b) $S$ and $\tilde{S}$ are normalized as $S^* = S - min(S)$ and $\tilde{S}^* = \tilde{S} - min(S)$, i.e. we substract the noise level of $S$ to both $S$ and $\tilde{S}$. $S^*$ and $\tilde{S}^*$ are introduced to bring the base level of the target spectrum to 0, otherwise, the integrals would be dominated by the noise rather than the signal, as logarithmic values are used. Note that both spectra are normalized with the same value (rigorously speaking, $min(S)$). In that way, we are able to spot discrepancies in absolute reflectivity (this answers part of the reviewer's question).

(c) The first term of the sum in $O(S, \tilde{S})$ is the hatched area divided by the blue area; the second term is the hatched area divided by the pink area. Both terms are needed to account for cases when $\tilde{S}$ would be "broader" than $S$ (i.e. when $\tilde{S}$ would overlap $S$ completely), **and** when $\tilde{S}$ would be "narrower" than $S$ (i.e. when $\tilde{S}$ would completely overlapped by $S$).

(d) In this example, the value of the overlap metric is 0.65 (65 %).

We included this description and the figure as an Appendix in the revised manuscript.

[Figure]

Figure 11: Illustration of the overlap metric.

New Appendix section "Overlap metric".
* * *
11. *Fig 4.: While impressive, I would prefer to see the profile of some latent variables like IWC and D 0 in panel b) instead of showing the same spectrogram three times. The argument that PAMTRA can be imitated reasonably well (panel c) is already demonstrated in Fig. 3.*

The purpose of this figure was to illustrate the quality of the reconstruction obtained with the pipeline, while the following figure (Figure 5) focuses on the retrieved latent variables. We believe that including the PAMTRA output is quite important: in the earlier figures, only elements from the synthetic dataset were shown, but this time we run PAMTRA directly on the retrieved variables. This sanity check verifies that the decoder matches PAMTRA, not only on *unseen synthetic* data (i.e. testing synthetic dataset) but also on real data that was generated independently from our synthetic sampling procedure.

This being said, we understand that the reader expects at this stage already some insight on the retrieved latent variables, and we therefore follow the reviewer's suggestion and include the profile of $D_0$ and IWC corresponding to this spectrogram.

[Modification of Fig. 4 and caption] Panel d): *IWC* and $D_0$ profiles retrieved from these spectrograms.
* * *
12. *Sec 5.3.2 Size parameter: Several times you mention "cofluctuation" (e.g., L500) between ground-based retrieved and airborne measured properties. Have you averaged the airborne dataset after the selection of nearby overpasses? Otherwise, I would not expect a good correlation between a ground-based and airborne platform. This could also explain the much higher variability of the RASTA retrieval.*

The airborne data were averaged with a 5-second running mean. Assuming the platform has a speed of $\sim 100 \ ms^{-1}$, the aircraft data are thus averaged on a horizontal distance of around 500 m, which is already significant (i.e., there may already be some signs of spatial variability at this scale). Given the small volume of the in-situ probes, averaging over shorter time steps would result in even noisier measurements, hence the choice of this averaging window as a trade-off. We included this information:

All aircraft-based microphysical descriptors are computed using 5-second running averages of the measurements.
* * *
13. *Sec 5.3.4 Aspect ratio: While reading this section I noticed that you never mentioned the average particle orientation in your PAMTRA simulations. As this is obviously fixed, this could be a further origin for the observed bias. Furthermore, I noticed here that you limited your AR to oblate particles (see Fig. B5, last panel). In the presence of rice-shaped particles, e.g., needles, bigger biases in your size and IWC retrieval may be expected and could explain the observed*

*discrepancies. Please include this in your discussion.*

We agree with the reviewer that particle orientation is an important aspect, that our retrieval does not handle. In the PAMTRA simulations, particles were considered to be oriented with their maximum dimension in the horizontal plane. We included a sentence in Sect. 3.1.1 with this information, and a reminder of this important assumption in the discussion.

Regarding the reviewer's other comment on aspect ratio, we note that the definition of aspect ratio for SSRGA calculations (i.e. the definition which we used in this work) differs from the one typically used to describe oblate vs. prolate. We refer for example to Ori et al. [2021], where this difference is highlighted. The effective aspect ratio in SSRGA calculations is the ratio of particle dimension in the direction of propagation, divided by its maximum dimension, and is thus always smaller than 1.

The aspect ratio $A_r$ is then specified, defined here as equal to the particle's dimension along the direction of radar beam (here, vertical) divided by maximum dimension [Ori et al., 2021], which implies $A_r \leq 1$. The particles are considered to be oriented with their maximum dimension in the horizontal plane.

[Discussion, Sect. 6.4] In addition to these important hypotheses —SSRGA scattering model and assumption of exponential size distributions—, we recall that other modeling choices were made during the design of the synthetic dataset and the underlying physical framework, e.g. assumptions on particle orientation, velocity-size relation, etc. (cf Sect. 3.1) which are inevitably a simplification of the physical reality and may thus also influence the retrieval.
* * *
**Typos and wording**

- *Throughout the text you are using the saxon genitive with objects, some examples:* Fixed.
  - *L11: "the problem's ill-posedness"* → *"the ill-posedness of the problem"*
  - *L13: "the retrieval's accuracy" … "the accuracy of the retrieval" or "the retrieval accuracy "*
  - *L104: "method's sensitivity" … "the sensitivity of the method"*
  - *L196: "the radar's properties" … "the radar properties"*

- *L52: "comforted through" … "confirmed through":* Fixed.

- *L67: "The scattering regime transition in high frequencies is in principle visible" … "The transition of the scattering regime at higher frequencies is visible":* Fixed.

- *L124: "we use as a forward model the radiative transfer code PAMTRA" … "we use the radiative transfer code PAMTRA as a forward model":* Fixed.

- *L207: "leaving to future studies the possible improvements of the forward model" … "leaving possible improvements of the forward model to future studies":* Fixed.

- *Tab 1: $\alpha$ a and $\beta$ a should probably be the pre-factor and exponent of the area-size relationship?:* Fixed.

- *L246: This sentence is awkward and hard to follow, please rephrase.* We rephrased the last two items: 4. [...] The resulting broadening is derived following Shupe et al. [2008]; the radar settings (e.g. beam width) used in these equations are those of the W- and X-band radars used in this study described in Section 3.2. 5. Finally, for computational reasons, X- and W-band spectra are both reduced to 256 points through bin averaging.

- *L458: "Sect. 3.2" is now "Appendix A":* Fixed.

**References**

Shannon Mason, Robin Hogan, Christopher Westbrook, Stefan Kneifel, and Dmitri Moisseev. The importance of particle size distribution shape for triple-frequency radar retrievals of the morphology of snow. *Atmospheric Measurement Techniques Discussions*, pages 1–30, 2019. doi: 10.5194/amt-2019-100.

Hugh Morrison, J. A. Curry, and V. I. Khvorostyanov. A new double-moment microphysics parameterization for application in cloud and climate models. Part I: Description. *Journal of the Atmospheric Sciences*, 62(6):1665–1677, 2005. ISSN 00224928. doi: 10.1175/JAS3446.1.

P Georgakaki, G Sotiropoulou, É Vignon, A.-C. Billault-Roux, A Berne, and A Nenes. Secondary ice production processes in wintertime alpine mixed-phase clouds. *Atmospheric Chemistry and Physics*, 22(3):1965–1988, 2022. doi: 10.5194/acp-22-1965-2022. URL https://acp.copernicus.org/articles/22/1965/2022/.

Stefan Kneifel, Annakaisa Von Lerber, Jussi Tiira, Dmitri Moisseev, Pavlos Kollias, and Jussi Leinonen. Observed relations between snowfall microphysics and triple-frequency radar measurements. *Journal of Geophysical Research*, 120(12):6034–6055, 2015. ISSN 21562202. doi: 10.1002/2015JD023156.

Eleni Tetoni, Florian Ewald, Martin Hagen, Gregor Köcher, Tobias Zinner, and Silke Groß. Retrievals of ice microphysics using dual-wavelength polarimetric radar observations during stratiform precipitation events. *Atmospheric Measurement Techniques Discussions*, (October), 2021. doi: 10.5194/amt-2021-216.

Petar Bukovčić, Alexander Ryzhkov, Dusan Zrnić, and Guifu Zhang. Polarimetric radar relations for quantification of snow based on disdrometer data. *Journal of Applied Meteorology and Climatology*, 57(1):103–120, 2018. ISSN 15588432. doi: 10.1175/JAMC-D-17-0090.1.

Sergey Y. Matrosov, Alexander V. Ryzhkov, Maximilian Maahn, and G. I.J.S. De BOER. Hydrometeor shape variability in snowfall as retrieved from polarimetric radar measurements. *Journal of Applied Meteorology and Climatology*, 59(9):1503–1517, 2020. ISSN 15588432. doi: 10.1175/JAMC-D-20-0052.1.

Mariko Oue, Pavlos Kollias, Sergey Y. Matrosov, Alessandro Battaglia, and Alexander V. Ryzhkov. Analysis of the microphysical properties of snowfall using scanning polarimetric and vertically pointing multi-frequency Doppler radars. *Atmospheric Measurement Techniques*, 14(7):4893–4913, 2021. ISSN 18678548. doi: 10.5194/amt-14-4893-2021.

Claudia Acquistapace, Stefan Kneifel, Ulrich Löhnert, Pavlos Kollias, Maximilian Maahn, and Matthias Bauer-Pfundstein. Optimizing observations of drizzle onset with millimeter-wavelength radars. *Atmospheric Measurement Techniques*, 10(5):1783–1802, 2017. ISSN 18678548. doi: 10.5194/amt-10-1783-2017.

A. J. Heymsfield and C. D. Westbrook. Advances in the estimation of ice particle fall speeds using laboratory and field measurements. *Journal of the Atmospheric Sciences*, 67(8):2469–2482, 2010. ISSN 00224928. doi: 10.1175/2010JAS3379.1.

Davide Ori, Leonie von Terzi, Markus Karrer, and Stefan Kneifel. snowScatt 1.0: consistent model of microphysical and scattering properties of rimed and unrimed snowflakes based on the self-similar Rayleigh–Gans approximation. *Geoscientific Model Development*, 14(3):1511–1531, mar 2021. ISSN 1991-9603. doi: 10.5194/gmd-14-1511-2021. URL https://gmd.copernicus.org/articles/14/1511/2021/.

Robert D. Sharman, L. B. Cornman, G. Meymaris, J. Pearson, and T. Farrar. Description and derived climatologies of automated in situ eddy-dissipation-rate reports of atmospheric turbulence. *Journal of Applied Meteorology and Climatology*, 53(6):1416–1432, 2014. ISSN 15588432. doi: 10.1175/JAMC-D-13-0329.1.

Matthew D. Shupe, Pavlos Kollias, Michael Poellot, and Edwin Eloranta. On deriving vertical air motions from cloud radar doppler spectra. *Journal of Atmospheric and Oceanic Technology*, 25(4):547–557, 2008. ISSN 07390572. doi: 10.1175/2007JTECHA1007.1.

Paloma Borque, Edward Luke, and Pavlos Kollias. On the unified estimation of turbulence eddy dissipation rate using Doppler cloud radars and lidars. *Journal of Geophysical Research: Atmospheres*, 120:5972–5989, 2016. ISSN 00280836. doi: 10.1038/175238c0.

Dominique Bouniol, Anthony J Illingworth, and Robin J Hogan. Deriving turbulent kinetic energy dissipation rate within clouds using ground based 94 GHz radar. *31st Conference on Radar Meteorology*, pages 193–196, 2003. URL http://ams.confex.com/ams/pdfpapers/63826.pdf.

Andrew Barrett, Christopher Westbrook, John Nicol, and Thorwald Stein. Rapid ice aggregation process revealed through triple-wavelength Doppler spectra radar analysis. *Rapid ice aggregation process revealed through triple-wavelength Doppler spectra radar analysis*, 19(8):5753–5769, 2019. ISSN 1680-7375. doi: 10.5194/acp-2018-836.

Grant W. Petty and Wei Huang. The modified gamma size distribution applied to inhomogeneous and nonspherical particles: Key relationships and conversions. *Journal of the Atmospheric Sciences*, 68(7):1460–1473, 2011. ISSN 00224928. doi: 10.1175/2011JAS3645.1.

Kamil Mroz, Alessandro Battaglia, Cuong Nguyen, Andrew Heymsfield, Alain Protat, and Mengistu Wolde. Triple-frequency radar retrieval of microphysical properties of snow. *Atmospheric Measurement Techniques*, 14(11):7243–7254, 2021. ISSN 18678548. doi: 10.5194/amt-14-7243-2021.

Kamil Mróz, Alessandro Battaglia, Stefan Kneifel, Leonie Von Terzi, Markus Karrer, and Davide Ori. Linking rain into ice microphysics across the melting layer in stratiform rain: A closure study. *Atmospheric Measurement Techniques*, 14(1):511–529, 2021. ISSN 18678548. doi: 10.5194/amt-14-511-2021.

Leonie von Terzi, José Dias Neto, Davide Ori, Alexander Myagkov, and Stefan Kneifel. Ice microphysical processes in the dendritic growth layer: a statistical analysis combining multi-frequency and polarimetric Doppler cloud radar observations. *Atmospheric Chemistry and Physics*, 22(17):11795–11821, 2022. doi: 10.5194/acp-22-11795-2022.

Dennis Piontek, Luca Bugliaro, Marius Schmidl, Daniel K. Zhou, and Christiane Voigt. The new volcanic ash satellite retrieval vacos using msg/seviri and artificial neural networks: 1. development. *Remote Sensing*, 13(16):1–29, 2021. ISSN 20724292. doi: 10.3390/rs13163112.

Teresa Vogl, Maximilian Maahn, Stefan Kneifel, Willi Schimmel, Dmitri Moisseev, and Heike Kalesse-Los. Using artificial neural networks to predict riming from Doppler cloud radar observations. *Atmospheric Measurement Techniques*, 15(2):365–381, jan 2022. ISSN 1867-8548. doi: 10.5194/amt-15-365-2022. URL `https://amt.copernicus.org/articles/15/365/2022/`.

Randy J. Chase, Stephen W. Nesbitt, and Greg M. McFarquhar. A dual-frequency radar retrieval of two parameters of the snowfall particle size distribution using a neural network. *Journal of Applied Meteorology and Climatology*, 60(3):341–359, 2021. ISSN 15588432. doi: 10.1175/JAMC-D-20-0177.1.

Mark A. Kramer. Nonlinear principal component analysis using autoassociative neural networks. *AIChE Journal*, 37(2), 1991. ISSN 15475905. doi: 10.1002/aic.690370209.

G. E. Hinton and R. R. Salakhutdinov. Reducing the dimensionality of data with neural networks. *Science*, 313(5786), 2006. ISSN 00368075. doi: 10.1126/science.1127647.

Gunnar Behrens, Tom Beucler, Pierre Gentine, Fernando Iglesias-Suarez, Michael Pritchard, and Veronika Eyring. Non-Linear Dimensionality Reduction With a Variational Encoder Decoder to Understand Convective Processes in Climate Models. *Journal of Advances in Modeling Earth Systems*, 14(8):1–23, 2022. ISSN 19422466. doi: 10.1029/2022MS003130.

Jerry M Straka. *Cloud and precipitation microphysics: principles and parameterizations*. Cambridge University Press, 2009. ISBN 9780511581168. doi: https://doi.org/10.1017/CBO9780511581168.

Robin J. Hogan, Ryan Honeyager, Jani Tyynelä, and Stefan Kneifel. Calculating the millimetre-wave scattering phase function of snowflakes using the self-similar Rayleigh-Gans Approximation. *Quarterly Journal of the Royal Meteorological Society*, 143(703):834–844, jan 2017. ISSN 00359009. doi: 10.1002/qj.2968. URL `https://onlinelibrary.wiley.com/doi/10.1002/qj.2968`.

S. Kneifel, M. S. Kulie, and R. Bennartz. A triple-frequency approach to retrieve microphysical snowfall parameters. *Journal of Geophysical Research Atmospheres*, 116(11):1–15, 2011. ISSN 01480227. doi: 10.1029/2010JD015430.

Frederic Tridon, Israel Silber, Alessandro Battaglia, Stefan Kneifel, Ann Fridlind, Petros Kalogeras, and Ranvir Dhillon. Highly supercooled riming and unusual triple-frequency radar signatures over McMurdo Station, Antarctica. *Atmospheric Chemistry and Physics*, 22(18): 12467–12491, 2022. ISSN 16807324. doi: 10.5194/acp-22-12467-2022.

Alessandro Battaglia, Simone Tanelli, Frederic Tridon, Stefan Kneifel, Jussi Leinonen, and Pavlos Kollias. *Triple-Frequency Radar Retrievals*, pages 211–229. Springer International Publishing, Cham, 2020. ISBN 978-3-030-24568-9. doi: 10.1007/978-3-030-24568-9_13. URL `https://doi.org/10.1007/978-3-030-24568-9_13.KaimingHe,XiangyuZhang,ShaoqingRen,a`

Andreas Veit, Michael Wilber, and Serge Belongie. Residual networks behave like ensembles of relatively shallow networks. *Advances in Neural Information Processing Systems*, (May 2016): 550–558, 2016. ISSN 10495258.

David Balduzzi, Marcus Frean, Lennox Leary, Jp Lewis, Kurt Wan-Duo Ma, and Brian Mcwilliams. Shattered Gradients. *Proceedings of the 34th International Conference on Machine Learning*, pages 342–350, 2017.

---

## Author Response (AR2)

**Dual-frequency spectral radar retrieval of snowfall microphysics: a physics-driven deep learning approach**

**amt-2022-199**

**Responses to reviewers**

A.-C. Billault-Roux, G. Ghiggi, L. Jaffeux, A. Martini, N. Viltard and A. Berne

February 2, 2023

We thank S. Kneifel and one anonymous reviewer for their positive feedback on our manuscript.

We took into account the technical corrections suggested by S. Kneifel, listed below. In addition to these changes, we modified the y-axis of Fig. 15 in which the units were incorrect.
* * *
*It would be nice if Fig. 14 and Fig. 15 would include a legend or description for the different line colors. While it is understood that for Fig. 14 the different parameters have been randomly chosen, the parameter mu is varied linearly for Fig. 15 and could thus be easily included in a legend. This would help to understand the sign of the d_mu - d_Skew relationship and enhance the reproducibility.*

Each colored line corresponds to a different (time, range) gate, for which we vary $D_0$. The change in skewness caused by a modification of PSD shape at this same (time, range) gate is indicated with a horizontal lines (min and max change when varying $mu$ from -2 to 5). As it turns out, the maximum increase in skewness always corresponds to $\mu = 5$, and the maximum reduction of skewness corresponds to $\mu = -2$. We now clarify this point in the caption:

The caption of Fig. 15 now reads:

> Colored lines with scattered points: relative change in W-band skewness $\gamma_W$ ($\Delta\,\gamma_W/\gamma_W$) caused by adding a diameter offset $\Delta D_0$ on microphysical descriptors of selected (time, range) gates, if assuming an exponential PSD. Horizontal lines: for each of these (time,range) gates, maximum relative change in skewness caused by a modification of the PSD shape (assumed a gamma distribution, $\mu$ in the range [-2, +5]); the maximum increase (resp. reduction) in skewness, in full line (resp. dashed) is consistently obtained for $\mu = 5$ (resp. $\mu = -2$). For each selected (time,range) gate, the intersection of the horizontal and colored lines gives a $\Delta D_0$ value which causes the same relative change in skewness as a change in PSD shape (worst case). Dashed vertical lines show the mean of these $\Delta D_0$ values.
* * *
*L634 (marked-up version): "while attenuating strongly millimeter-wavelength signal" -¿ "while it can strongly attenuated the millimeter-wavelength signal"*

Fixed.